# A Historical Review of Military Medical Strategies for Fighting Infectious Diseases: From Battlefields to Global Health

**DOI:** 10.3390/biomedicines10082050

**Published:** 2022-08-22

**Authors:** Roberto Biselli, Roberto Nisini, Florigio Lista, Alberto Autore, Marco Lastilla, Giuseppe De Lorenzo, Mario Stefano Peragallo, Tommaso Stroffolini, Raffaele D’Amelio

**Affiliations:** 1Ispettorato Generale della Sanità Militare, Stato Maggiore della Difesa, Via S. Stefano Rotondo 4, 00184 Roma, Italy; 2Dipartimento di Malattie Infettive, Istituto Superiore di Sanità, Viale Regina Elena 299, 00161 Roma, Italy; 3Dipartimento Scientifico, Policlinico Militare, Comando Logistico dell’Esercito, Via S. Stefano Rotondo 4, 00184 Roma, Italy; 4Osservatorio Epidemiologico della Difesa, Ispettorato Generale della Sanità Militare, Stato Maggiore della Difesa, Via S. Stefano Rotondo 4, 00184 Roma, Italy; 5Istituto di Medicina Aerospaziale, Comando Logistico dell’Aeronautica Militare, Viale Piero Gobetti 2, 00185 Roma, Italy; 6Comando Generale dell’Arma dei Carabinieri, Dipartimento per l’Organizzazione Sanitaria e Veterinaria, Viale Romania 45, 00197 Roma, Italy; 7Centro Studi e Ricerche di Sanità e Veterinaria, Comando Logistico dell’Esercito, Via S. Stefano Rotondo 4, 00184 Roma, Italy; 8Dipartimento di Malattie Infettive e Tropicali, Policlinico Umberto I, 00161 Roma, Italy; 9Dipartimento di Medicina Clinica e Molecolare, Sapienza Università di Roma, Via di Grottarossa 1035-1039, 00189 Roma, Italy

**Keywords:** the military, infectious diseases, passive immunization, vaccines, antibodies, active immunization, biological agents, war

## Abstract

The environmental conditions generated by war and characterized by poverty, undernutrition, stress, difficult access to safe water and food as well as lack of environmental and personal hygiene favor the spread of many infectious diseases. Epidemic typhus, plague, malaria, cholera, typhoid fever, hepatitis, tetanus, and smallpox have nearly constantly accompanied wars, frequently deeply conditioning the outcome of battles/wars more than weapons and military strategy. At the end of the nineteenth century, with the birth of bacteriology, military medical researchers in Germany, the United Kingdom, and France were active in discovering the etiological agents of some diseases and in developing preventive vaccines. Emil von Behring, Ronald Ross and Charles Laveran, who were or served as military physicians, won the first, the second, and the seventh Nobel Prize for Physiology or Medicine for discovering passive anti-diphtheria/tetanus immunotherapy and for identifying mosquito Anopheline as a malaria vector and plasmodium as its etiological agent, respectively. Meanwhile, Major Walter Reed in the United States of America discovered the mosquito vector of yellow fever, thus paving the way for its prevention by vector control. In this work, the military relevance of some vaccine-preventable and non-vaccine-preventable infectious diseases, as well as of biological weapons, and the military contributions to their control will be described. Currently, the civil–military medical collaboration is getting closer and becoming interdependent, from research and development for the prevention of infectious diseases to disasters and emergencies management, as recently demonstrated in Ebola and Zika outbreaks and the COVID-19 pandemic, even with the high biocontainment aeromedical evacuation, in a sort of global health diplomacy.

## 1. Introduction

The military worldwide have always been challenged with the issue of infectious diseases, which may deeply influence the outcome of battles/wars. The military are particularly exposed to the risk of infectious diseases for a series of reasons, including the community life, often in precarious environmental conditions regarding the hygiene of water and food supply, sanitation, the traumatism with contaminated wounds, and the possibility to be exposed to extreme temperatures and to diseases unknown in their country of origin, for which no natural immunization has, therefore, been developed [1,2]. In 431 BCE, the outcome of the Peloponnesian war between the Athens of Pericles and Sparta was determined by the so-called “plague of Athens”, a terrible epidemic responsible for the death of approximately one-third of the Athens’ population, of Pericles and two of his sons, which seems to have been due to an outbreak of *Salmonella typhi*, as recently reported [3,4]. More recently, Napoleon lost 90% of his army deployed to Haiti, 27,000/30,000 soldiers including the commander, who was Napoleon’s brother-in-law, as a consequence of yellow fever, which was endemic in Haiti, but unknown to the French troops, which were, therefore, highly vulnerable. This situation induced Napoleon to retire from the New World and leave Louisiana for the then-nascent United States of America (USA) to concentrate his efforts in Europe [3]. It has been estimated that among the 600,000 French soldiers who lost their lives in war during the eighteenth century, over 50% were due to disease. During the war campaign in Madagascar, 1895–1896, 30% of French soldiers lost their lives, approximately one hundred to combat wounds and 4500 to infectious diseases (malaria, typhoid, dysentery) [5]. In the USA troops, the ratio of death rate for disease/death rate for combat wounds was 7:1 during the Mexican war (1846–1848) and 5:1 during the Spanish war in 1898. Conversely, among the Germans during the Franco-Prussian War of 1870 and among the Japanese and the Russians in the Russo-Japanese War of 1904, the number of wounded was higher than the number of sick soldiers [6].

Consequently, the issue of infectious diseases has been faced by the military health services often earlier than the civilian counterpart, and the contribution provided by the military scientists to the birth of passive immunization and the development of active immunization was relevant starting from the end of the nineteenth century. Moreover, many vaccines have been developed and often tested in the military, considering that pre-enrollment screening, easy follow-up and a standardized way of life make the military an ideal population for studying the safety and efficacy of a drug/vaccine. A survey carried out by the World Health Organization (WHO) in 1998 showed that 47 out of 52 participating countries (90%) had a compulsory vaccination program for the military [7]. The lethal and/or incapacitating power of certain infectious diseases has also been exploited to fight enemies, and armies have developed strategies to use pathogens or toxins as biological weapons.

In this paper, a historical approach to the military fight against infectious diseases is reviewed by describing the military involvement in (i) vaccine- and (ii) non-vaccine-preventable diseases; (iii) acute respiratory and (iv) diarrheal syndromes, (v) the study of major agents developed for biological warfare and (vi) the high biocontainment aeromedical evacuation. Although the military are particularly exposed to some specific infectious diseases, which are widespread and often burdened by high mortality (Table 1), they may be also at higher risk for other infectious diseases, whose spreading is favored by the specific environmental conditions that characterize the military life. Thus, we extended our review to cover all the main infectious threats to the military and the role of military health services and scientists in their containment (Table 2).

This is not the first report on historical military medicine of infectious diseases, but it is the result of a large examination of the available literature from the USA, Australia, and western Europe military medical science that offers a systematic and global review that is unique in the field.

## 2. Vaccine-Preventable Infectious Diseases

### 2.1. Smallpox

Smallpox was a feared infectious disease caused by one of two virus variants, *Variola major* and *Variola minor*, belonging to the genus *Orthopoxvirus*. Smallpox was characterized by an incubation period of 10–14 days, a transmission mainly through respiratory route [26], and symptomatology with fever, general malaise and a vesicular, and then pustular, skin rash. Historically it manifested with periodical epidemics. The disease was declared eradicated in 1980, following an aggressive vaccination campaign driven by the WHO [27]. It was highly contagious, with an average lethality of 15–30%, ranging from 1% in the case of variola minor up to more than 97% in the case of hemorrhagic smallpox. In the attempt to reduce the consequences of smallpox, the practice of variolation was reported starting from the tenth century in China, and probably in India. Variolation is the inoculation of the secretion of a smallpox lesion taken from a mild case in a susceptible subject to protect him/her from natural smallpox. This practice was in use for a long-time, even though burdened by relatively high mortality, of the order of 0.33%, but up to 3%, which was, however, markedly lower than the average lethality of the natural infection, approximately 16% [28]. The relevance of smallpox for the outcome of wars was demonstrated during the independence war of the American colonies against the British Army (1775–1783) and later in Europe during the Franco-Prussian war (1870–1871). In the former war in 1777, General George Washington ordered the variolation of the Continental Army after losing many soldiers because of smallpox, 1800 out of 7000 only in the last 2 weeks of May 1776 [29]. In 1796, Edward Jenner, an English physician, based on the observation that the milkmaids were protected from smallpox because of hand lesions contracted during the milking of cows affected by cowpox, decided to take the secretion of a cowpox lesion and inoculate it into a boy, who was later challenged with smallpox, to which he was protected.

This same experiment was repeated in 22 other volunteers, always with successful results, which were published in 1798 [30]. At the beginning of the nineteenth century in most European countries and the USA, vaccination was adopted, to become compulsory for infants in Bavaria (1807), Denmark (1810), Norway (1811), Bohemia and Russia (1812), Sweden (1816), Hanover (1821), and Great Britain (1853) [27]. Similar to most other infectious diseases, precarious hygienic conditions, as observed in war periods, facilitate the spreading and severity of infection; thus, the military all over the world immediately applied prophylaxis for this dreadful disease, in some countries well before the application to the general population. This was the case for Italy, where the smallpox vaccine became compulsory for the military nearly 30 years earlier than for the general population, for which smallpox vaccination was made compulsory for all newborns in 1888 [31]. Despite that the vaccination for the military in Prussia was mandatory since 1831 [26], it was only offered to the general population in Germany, whereas it was only made mandatory following the Franco-Prussian war in 1870–1871. However, the Prussian army was well protected during the smallpox outbreak; smallpox was nearly non-influent for the Prussian soldiers (only 278 soldiers died), whereas the French soldiers, for whom vaccination was compulsory but revaccinations were not systematically carried out [32], had 23,400 deaths because of smallpox; thus, the smallpox epidemic was one relevant factor for deciding the outcome of the war [33]. In 1811, Napoleon introduced the smallpox vaccination for the army recruits [3], whereas for the general population, the vaccination was promoted, offered for free to indigent individuals, but never made mandatory by law [34]. In 1806, the ruler of Lucca and Piombino, Elisa Bonaparte, Napoleon’s sister, mandated the vaccination of newborns and adults. In 1853, the Compulsory Vaccination Act introduced the mandatory smallpox vaccination for infants in England and Wales. However, due to the military organization, the relevance for the military to fight infectious diseases, and the lower relevance in the military of the no-vax opinion groups, the application of vaccination in the military was generally earlier and better implemented than in the corresponding civilian population.

In 1980, the WHO declared smallpox eradicated, after the last case of natural smallpox occurring in Somalia in 1977, and recommended vaccination interruption, considering that the risk of adverse events was higher than the risk of smallpox infection. However, in some countries, the military continued to be immunized, as a prevention for the possible use of smallpox as a biological agent on a population that was not protected anymore. The fear of the possible use of smallpox as a biological weapon became more pressing after the episode of anthrax sent by mail; thus, USA President Bush ordered that the health workers and the military were compulsorily immunized. However, vaccination was interrupted after having vaccinated approximately 500,000 military subjects and 40,000 health workers, for the relatively high frequency of adverse events [35]. Nonetheless, the last North Atlantic Treaty Organization (NATO) document on the vaccinations for the military in the 30 NATO countries reports that 3/25 countries that have reported their military vaccination schedule, still maintain smallpox vaccine for selected categories of personnel [36]. New, less reactogenic, tissue-culture-based live attenuated, and subunit smallpox vaccine formulations are studied for the risk that smallpox may be used as a biological weapon [37], or for protection against naturally occurring monkeypox. Moreover, by collecting the blood of the immunized people, it was possible to produce specific polyclonal Ig, which were protective and could be used in emergencies, with significantly lower adverse events than the vaccine [35]. Thus, despite smallpox having been eradicated since 1980, the interest for the military is still great, in light of its possible use as a biological weapon, of category A. The military contribution is the worldwide early military vaccination, which may have contributed to its eradication.

### 2.2. Typhoid Fever

Typhoid fever is a serious infection caused by *Salmonella typhi*, a Gram-negative bacterial microorganism, which may infect through ingestion of contaminated water or food. The disease is characterized by high fever, headache, gastralgia, diarrhea or constipation, hepato-splenomegaly and possible complications, such as intestinal perforation. In the pre-antibiotic era, mortality was as high as 20%. Salmonella has three antigens, the O and H antigens, thermostable and thermolabile, respectively, and a third antigen Vi, for virulence. The diagnosis may be carried out by stool culture, blood culture and serologically, by the search for specific anti-O and anti-H antibodies. Some people may become chronic carriers of *S. typhi*, continuing to release bacteria in their stools, thus spreading the disease. Typhoid fever is a classic example of an infectious disease spreading in unfavorable hygienic conditions, with lack of access to safe water and food, as may be observed during the war. This, joined with the severe clinical picture and the relatively high lethality, makes the disease of great interest to the military. In addition to the environmental prophylaxis, the search for an effective vaccine has registered the activity of military researchers from Germany, Great Britain, France, Italy, and the USA. The development of the typhoid vaccine has been traditionally attributed to Almroth Wright, Professor of Pathology at the British Army Medical School at Netley, even though documents prove that Wright, appointed by the Director of the Army Medical Service to develop a typhoid vaccine and worried to be unable to comply, was reassured after knowing the results obtained by Prof. Richard Pfeiffer in Germany about the development of a typhoid vaccine [38,39]. Pfeiffer, a military doctor of the German Army, was seconded to the Laboratory of Robert Koch at the University of Berlin, and applied with success to bacteriology and immunology, by observing that a heated *S. typhi* culture, inoculated subcutaneously in man, could induce antibody-mediated agglutination. These data were described by Pfeiffer and Kolle in 1897 [40]. In 1896, Wright published a paper that was not focused on typhoid vaccination [41], while his paper on typhoid vaccination was contemporaneous to the paper of Pfeiffer and Kolle in 1897 [42]. However, independently of who was the first, this activity witnesses the interest of the military in preventing this dreadful disease. The first chance to test the vaccine’s effectiveness was the Anglo-Boer War in southern Africa in 1899, during which the British Army used early forms of the typhoid vaccine. Among 14,626 immunized British soldiers, 1417 contracted typhoid fever and 163 died, with an attack rate of 9.7% and a case fatality rate of 11.5%. Conversely, 48,754 cases of typhoid fever occurred among 313,618 unimmunized soldiers, and 6991 died, with an attack rate of 15.5% (*p* < 0.0000001 vs. the attack rate of immunized soldiers), and a case fatality rate of 14.34% (*p* = 0.002965 vs. lethality of immunized soldiers) [43]. However, the golden test for proving the vaccine effectiveness of the typhoid vaccine in preventing typhoid fever was World War I (WWI) trench warfare during which all the infectious diseases favored by poor hygiene, such as typhoid fever, could easily spread. The British Army decided, therefore, that troops should be vaccinated, but, contrarily to Germany, France, and Italy, which imposed the compulsory vaccination, the British Army, for the strength of anti-vaccine movements that had obtained exemption from the smallpox vaccine, could not decide for the mandatory typhoid vaccination, but only for a warm vaccine recommendation. Nonetheless, the percentage of vaccinated soldiers was 94%, and the vaccine effectiveness was clearly demonstrated; among the 604,420 vaccinated, 570 typhoid cases and 34 deaths were observed versus 295 cases and 89 deaths among the 38,580 non-vaccinated (Table 3) [44]. The incidence of typhoid in the British Army during the Anglo-Boer War was 285 per 1000, while in WWI, it fell to <1 per 1000 [45]. Moreover, the British Army, under the guidance of Col. David Harvey, could develop a trivalent vaccine against *S. typhi* and *S. paratyphi* A and B (TAB), able to better protect against enteric infections during the war in France. This vaccine was introduced in 1916 [46]. In the same period, the Italian microbiologist Aldo Castellani, Director of Government Clinic for Tropical Diseases, Colombo, Ceylon, later full Professor of Tropical and Sub-Tropical Diseases, University of Rome, and physician in the Italian Navy with the rank of General, developed and successfully experimented the combined killed TAB vaccine [47], by promoting its use on the military during WWI. The TAB vaccine, combined with the tetanus toxoid (TABTe), was used in the Italian military until the second half of the 1980s, when it was replaced by a new live oral vaccine [48], following a comparative study between the two vaccines, which showed lower reactogenicity of the oral vaccine associated with good immunogenicity, even at mucosal level [49].

A few years later, in France and the USA, military researchers prepared inactivated typhoid vaccines, which have been largely and successfully used during WWI. In France, the military medical doctor Hyacinthe Vincent, in collaboration with André Chantemesse, a medical researcher of the Pasteur Institute, developed a typhoid vaccine, able to control the typhoid epidemic, which had provoked more than 65,000 cases among the French troops in the period ranging between September 1914 and May 1915 [50,51]. Meanwhile in the USA, Major Frederick Russell, who had visited the laboratories of Wright and Pfeiffer, developed a whole cell typhoid vaccine, heat and chemically inactivated, similar to the vaccine of Wright and Pfeiffer, which became compulsory for the US Army and Navy in 1911. By using this vaccine for the 4,100,000 USA military during WWI, only approximately 2000 cases of typhoid fever have been observed with 227 deaths [52]. The US Army had an incidence of typhoid fever of 142 per 1000 in 1898, which fell to <1 per 1000 during WWI [53].

In conclusion, the first typhoid vaccines, all developed by military researchers, even though reactogenic and incompletely protective, showed a satisfying protection in the unfavorable hygienic conditions of the trench warfare such as of the one of WWI. During the 1970s, a new live attenuated oral typhoid vaccine was developed from a wild-type *S. typhi* strain Ty2 made defective from the galactose-epimerase gene and Vi antigen by chemical mutagenesis [54]; it was approved in Europe in 1983 and in the USA in 1989. A Vi polysaccharide injectable typhoid vaccine was developed in the 1970s and is used in many world countries. Moreover, in the second half of the 1980s, the Vi polysaccharide–protein conjugate was also developed [55]. The conjugate vaccine, in which the Vi polysaccharide is linked to a protein matrix, which may be represented by tetanus toxoid, or diphtheria toxoid, or CRM197 (a recombinant, avirulent analogous of diphtheria toxin) or recombinant exotoxin A of *Pseudomonas aeruginosa*, compared to the plain polysaccharide vaccine, allows a T-independent antigen to be transformed into a T-dependent one, thus eliciting memory cells. However, despite that it represents a more effective vaccine than the plain polysaccharide, the conjugation process is complex and expensive; thus, it has currently only been approved in endemic countries, such as India and Nepal [56].

Currently, with the improvement of hygienic conditions, typhoid fever has virtually disappeared in developed countries; thus, in the military of many developed countries, such as Italy, typhoid vaccination is only compulsory for troops deployed abroad, in developing countries with unfavorable epidemiological situations. The WHO estimates an annual incidence of 11–20 million typhoid cases and annual deaths of 128,000–161,000, mainly occurring in developing areas of Africa, the Americas, Southeast Asia and Western Pacific regions [20]. Typhoid vaccination is present in all the 25 NATO countries out of 30, which have reported the vaccination schedule for the military. In 18 countries, the used vaccine is the inactivated one, whereas in the remaining seven, it is the live attenuated one. In none of these 25 countries vaccination is it addressed to the whole military personnel, but analogous with Italy, it is addressed to selected categories only [36]. The first vaccine development was uniquely carried out by the military, and it was crucial in disease containment. *S. typhi* has been included among the biological agents, category B [57].

### 2.3. Tetanus

Tetanus is a potentially lethal disease caused by the anaerobic microorganism *Clostridium tetani*, which produces a neurotoxin toxin (tetanospasmin). The severe symptomatology of the disease is characterized by spastic palsy, due to the inhibition of the inhibitory neurotransmitters of nerve terminals of lower motor neurons, the nerves activating voluntary muscles [58]. The spores of *C. tetani* are resistant in the soil; thus, the wounds with necrotic parts contaminated by topsoil are at particular risk of developing the infection. In absence of therapy, the disease is virtually always lethal. Emil Adolf von Behring, a German military physician expert in disinfection, joined the Robert Koch’s Institute of Hygiene in 1890, after leaving the Army. In that period, in France with Louis Pasteur and in Germany with Robert Koch, microbiology and immunology were emerging. In particular, the Koch’s Laboratory collected many scientists around, including Behring, Richard Pfeiffer who with Kolle and Wright in Great Britain, will share credit for developing the typhoid vaccine, Paul Ehrlich, bacteriologist, and immunologist, Shibasaburo Kitasato, who isolated the *C. tetani*. Behring and Kitasato, in December 1890, published one paper describing that the inoculation of sterilized cultures of tetanus in rabbits induced the appearance of antitoxins in the blood, as proven by the inoculation of this immune blood in mice that resulted protected by a challenge with tetanus [59]. A week later, Behring published another paper to extend this observation to diphtheria as well [60]. Based on these premises, Behring inoculated the serum of a previously immunized animal to diphtheria toxin in an eight-year-old boy with severe diphtheria, who later had a full recovery. The lethality rate of diphtheria in the following 10 years decreased from 50% to 13% [61]. This represented the birth of passive immunization, which has later been applied to different clinical contexts, including the recently set up of monoclonal antibodies to severe acute respiratory syndrome coronavirus (SaRS-CoV-2). For the relevance of this discovery, Emil von Behring was awarded the first Nobel Prize for Physiology or Medicine in 1901, “*for his work on serum therapy*, *especially its application against diphtheria*, *by which he has opened a new road in the domain of medical science and thereby placed in the hands of the physician a victorious weapon against illness and death*” [62]. The test case for demonstrating the effectiveness of anti-tetanus hyper-immune animal serum was WWI, during which contaminated wounds were frequently complicated by tetanus. Considering that the vaccine was not developed yet, the only protective weapon, in addition to a thorough wound toilet, was the hyper-immune serum, which appeared more effective in prophylaxis than in therapy, as reported in the UK experience. Among 2,032,142 wounded British soldiers, 2385 were tetanus cases, with an incidence of 1.17:1000 [63]. The case-fatality rate among the 1437 cases of tetanus occurring in England was 34.8%, ranging from over 70% to less than 20% according to the tetanus severity. The case-fatality rate was higher in the British troops stationed in France (71.3%) than in England [63]. The use of hyper-immune serum as therapy could poorly influence the outcome; instead, the prophylactic use was probably responsible for the reduction of incidence from 9 per 1000 in September to 1.4 per 1000 in December and for the reduction of the case-fatality rate from 85%, which was the average pre-serum observed case-fatality rate, to 47%, which resulted from joining together the British cases of tetanus observed in England and the French war theater [63] (Table 4). In 1914, the lethality rate for tetanus in the German Army ranged from 75% to 100% [64]. In the Italian Army, tetanus was negligible, with an incidence of 0.5 per 1000 (it was 10 per 1000 in the Russo-Japanese War) and a mortality rate of 1:33,000 [65].

In 1923, a veterinarian of the Pasteur Institute, Gaston Ramon, by exposing tetanus and diphtheria toxins to 0.5% formaldehyde and heat, was able to eliminate their virulence, while maintaining their antigen power, thus paving the way for the respective vaccines to be prepared. The transformed toxins were denominated by Ramon anatoxins, and, in 1926, profiting from the close collaboration between the Institute Pasteur Network and the French Military Medical Service, the military medical doctor Christian Zoeller collaborated with Ramon to improve the vaccines for tetanus and diphtheria. These vaccines were studied in the military population, and a few years later, they became compulsory for the military, diphtheria in 1931 and tetanus in 1936 [5]. The tetanus vaccine became compulsory in the military of other countries before the start of WWII, in Italy in 1938, and in the USA in 1940 [52]. As WWI was the test case to analyze the effectiveness of anti-tetanus hyper-immune serum, WWII was the test case to analyze the effectiveness of the tetanus toxoid vaccine. In WWI, the incidence of tetanus in the US Army was 13.4 per 100,000 wounded and injured versus 0.44 per 100,000 in WWII [66], over thirty-fold lower (Table 4); thus, definitively demonstrating the high effectiveness of the tetanus toxoid vaccine.

Although nearly 100 years have elapsed since the tetanus vaccine development by Ramon, no substantial modifications have been introduced in this vaccine preparation, which has remained the same. A certain degree of reactogenicity observed in the 1940s has been attributed to some peptones formed during the process of toxoid preparation, which have been removed [66]. In developed countries, the disease has virtually disappeared; however, in most military vaccination programs, tetanus vaccine is present [7]. The tetanus vaccine is included in the military vaccination program of all the 25 NATO countries, which have reported the respective vaccination schedule out of the 30 ones, 23 of which for the whole personnel and in another two for selected categories [36]. In Italy, the tetanus toxoid vaccine was included in the vaccination program for infants only in 1968, thirty years later than for the military. The relevance of military vaccination as a public health measure for tetanus prevention was witnessed in Italy and France, until the conscription was present in both countries, by the unbalanced epidemiological situation of the few cases of tetanus annually reported, which were characterized by a marked preponderance of old females, who were not covered by vaccination because it was not administered during the military service, which was only compulsory for males, nor in infancy, because it was not introduced in the infant vaccination schedule yet [5]. Currently, an open issue is the durability of vaccine-induced antibodies and thus the right timing for booster administration to maintain the protective antibody levels without risking hyper-immunization [67,68,69]. The military contributed to the discovery of passive immunotherapy and to the collaboration to vaccine development.

### 2.4. Diphtheria

Diphtheria is an infectious disease caused by the toxigenic strains of the Gram-positive *Corynebacterium diphtheriae*, of which three main biotypes exist: gravis, intermedius, and mitis. The infection is localized in the high airways, where the toxin causes rhinitis, pharyngitis, and laryngitis. The toxin may induce myocarditis and polyneuropathy; the disease is generally observed in <15-year-old boys and the case-fatality rate is 5–17% [70]. Diphtheria has been described in the sixteenth and seventeenth centuries in Spain, with recurrent epidemics in the eighteenth century in the USA, in the nineteenth and twentieth in Europe and more recently even in Asia and Africa. The etiologic agent was identified by Edwin Klebs in 1883 and was cultured by Friedrich Löffler, who demonstrated the toxin as well, whereas the progress in passive and active immunization is parallel to the one of tetanus, and it has been reported above in the paragraph of tetanus.

Considering that in non-vaccinated subjects, the disease is generally observed in <15-year-old boys, diphtheria is not apparently of military interest. However, the military must travel to different world countries, and if they are exposed to the etiologic agent in conditions of insufficient immune protection, they may be infected and become carriers, thus spreading the infection. This appears to have been the case for the start of a diphtheria epidemic occurring in the period 1990–1995 in the newly independent states of the former Soviet Union, where 47,808 cumulative cases of diphtheria occurred, 1746 of which were fatal [71]. A cluster of diphtheria infection was described in the members of a military construction battalion in Moscow in 1990. It must be considered that the Soviet troops had been present, in the period 1980–1989, in Afghanistan, which reported to the WHO 13,628 cases of diphtheria in the same period. Considering that the notification system for infectious diseases in the former Soviet Union was completely separated between military and civilian populations, civilian health authorities were not immediately aware of these diphtheria cases occurring in the military; thus, the actions for limiting the infection spreading were late and largely ineffective [71]. However, the causes for the spreading of the infection were largely unknown, but a high rate of unimmunized children and waning immunity in adults was certainly present; thus, even in the armies of developed countries, where diphtheria has been eradicated, particular attention to maintaining the antibody levels above the threshold for protection has become mandatory. In Italy, diphtheria booster was added to the compulsory vaccination schedule for the military after demonstration of the relatively low percentage of recruits with protective antibody levels [72]. However, even though in the military much attention has been paid to the need to maintain protective antibody levels for diphtheria, a survey made up among the military medical services of 52 world countries showed that the tetanus vaccine was present in the compulsory vaccine program for the military in 45/52 (87%), whereas diphtheria was only present in 30/52 (58%) [7]. Currently, the diphtheria vaccine is included in the military vaccination program of all 25 NATO countries, which have reported the respective vaccination schedule out of the 30, 22 of which are for the whole personnel and the other three for selected categories [36]. The outbreak of diphtheria in the newly independent states of the former Soviet Union in the 1990s is a clear example of how the military may become involuntary carriers of disease; thus, the military health authorities should not only combat infectious diseases for assuring the operational readiness but even closely collaborate with civilian health authorities in order to prevent possible military-mediated outbreaks. The complete separation of civilian and military notification systems for infectious diseases in the former Soviet Union was instead an example of a flawed organization, which has allowed the happening of such a dramatic event.

### 2.5. Pertussis

Pertussis is a highly contagious infectious disease, known for many centuries, caused by the Gram-negative coccobacillus *Bordetella pertussis*, which was isolated and cultivated by Jules Bordet and Octave Gengou in 1906 [73]. The most relevant symptom is whooping cough, which may be accompanied by inflammation of the high airways and may be complicated by apnea, pneumonia, rib fractures, insomnia, hospitalization, and rarely death [74]. The disease was generally observed in infancy, but in the last 20–30 years, it has even been observed in adults [74], thus acquiring an interest for the military, considering that in many countries, limited outbreaks in the military have been described [75,76,77,78,79]. In the Italian military, a study carried out in the 1990s showed that more than 90% of subjects had specific cell-mediated and antibody immunity to *B. pertussis* and that symptoms suggestive of pertussis were absent in the military [80].

Two types of vaccines are available, the first one is whole-cell, older, and inactivated, whereas the second vaccine, developed in the 1970s, but practically available since the 1990s, is acellular, recombinant and may only contain one, two, or three of the main virulence factors of the microorganism, represented by the pertussis toxin, the pertactin, and the filamentous hemagglutinin. The whole-cell vaccine is more reactogenic, however, it seems quite more effective and able to provide more durable protection. Both vaccines are combined with tetanus and diphtheria, in a trivalent diphtheria/tetanus/pertussis (DTP) or diphtheria/tetanus/acellular pertussis (DtaP). Pertussis is now, in both the USA and Europe, particularly present in adults, who represent the major reservoir for the infection [81]. Currently, 21/25 countries reporting the vaccination military program among the 30 countries considered in the document of the NATO standardization agreement for vaccination of 2021 declare having pertussis included in the program, in 18 countries for all the military personnel, in two out of the remaining three countries for selected categories (deployable, alert, risk personnel) and in one country for recommendation only [36]. The use of the trivalent DTP/DtaP vaccine in the military is a relevant measure of public health, particularly in the countries with conscription because maintaining a high level of immunity reduces the microorganisms’ circulation.

### 2.6. Tuberculosis (TB)

TB is a severe disease, whose infectious nature was demonstrated by the French military physician Jean-Antoine Villemin in 1865, and which was published in 1868 [82], through inoculation of material from infected humans to laboratory rabbits. TB is caused by *Mycobacterium tuberculosis*, discovered in 1882 by Robert Koch, who was awarded the Nobel Prize for Physiology or Medicine in 1905 [83]. The microorganism is transmitted through airways and may induce, after an average period of 3–9 months up to two years [84], either a latent or active disease, generally at lung level, but, more rarely, everywhere in the body. It is estimated that one-third of the world population is infected, the large majority with a latent infection and a minority, which in 2011 was represented by 8.7 million cases, with active infection, and 1.4 million deaths [8]. In 1895, a French military physician, Albert Calmette, who founded the Pasteur Institute in Saigon and later directed the Pasteur Institute in Lille, started his studies on tuberculosis and, together with the veterinarian Camille Guérin, developed a live attenuated vaccine for TB, which was successfully tested for the first time in 1921 [85]. This vaccine uses attenuated *Mycobacterium bovis* and is known as Bacillus Calmette–Guérin (BCG), after the names of its discoverers.

Similar to many other infectious diseases, TB spread increases in unfavorable environmental conditions, such as insane housing, overcrowding or hypo-nutrition, that characterize poverty and occur during wars, but community life may also favor TB spreading [86]. In the US military, the epidemiology of TB has been analyzed since the Civil War (1861–1865) up to the last wars in Iraq and Afghanistan. TB was more frequent in the military up to WWI and lower than in the civilian population in the following years [87]. During the American Civil War, the morbidity rate for TB was 924/100,000 and the mortality rate 261/100,000, whereas during the Spanish–American War (1898) the mortality rate was slightly reduced to 197/100,000, and during WWI, the morbidity rate increased up to 1168/100,000, with a higher prevalence of cases among soldiers who had remained in the USA compared with those who were deployed to Europe [86]. With WWII, lung X-ray was extensively employed to improve the screening before enrollment, thus preventing new cases coming from the contagion with infected comrades. From WWII, the influence of war on the epidemiology of TB seems inapparent, even when the wars occurred in countries endemic to TB, such as Korea, Iraq, and Afghanistan; the epidemiological curve of incidence continued to descend, until 0.4/100,000 in 2012, which represents a value eight-fold lower than in the USA civilian population [87]. However, a crucial point for reducing the cases of active TB in the military is to identify with the highest possible precision the subjects with latent TB among the applicants for military service during the pre-enrollment medical screening, for these cases to be adequately treated before enrollment, thus preventing the possible development of active TB as a consequence of the stress of the military life [88]. In the UK, the situation is quite different, considering that TB still presents a morbidity rate of 12.3/100,000 in the general population, mainly due to immigrants from high-endemicity countries (70/100,000 among immigrants versus 4/100,000 of UK-born people), but even dependent on risk factors, such as smoking, alcohol consumption, immunosuppression, and concomitant diseases, such as HIV infection and diabetes. The situation is similar in the UK military, in which historically at the end of the nineteenth century, TB represented the first cause of medical discharge from active service (300/100,000 in 1891). In the first half of the twentieth century, the situation improved by showing a reduction of approximately 50% (an average of 150/100,000), a behavior that was observed during WWII and even afterward, up to half a century. In the second half of the century, a series of initiatives were taken, including pre-enrollment screening, the diagnosis and treatment of latent TB infections and the offer of BCG to skin-negative subjects who had not received BCG in infancy. Based on a careful study, it emerged that the risk of TB was higher in older veterans who entered the Army before the implementation of preventive measures [89]. In Italy, a study carried out on over 2000 soldiers in the 1990s found a prevalence of latent infections (tuberculin-positive, asymptomatic subjects) of over 6% [90], a percentage not dissimilar from the percentage of the US Army in the same period [91]. Based on this result, in 2001, the norm of article 10 of Act 1088/1970 requiring that all skin-negative soldiers would have been vaccinated with BCG at enrollment was cancelled. The reactogenicity and the uncertain protection induced by BCG in adults [92] did not justify its administration in the presence of a relatively reduced prevalence. Moreover, in 2005, the conscription in Italy was abolished, thus deeply modifying the socio-epidemiology in the military. In addition to a numerical reduction of the military personnel, even the community life was reduced and only maintained during the training and operational periods, thus reducing the occasions for infections spreading. The lung TB in the period 1986–1997 in the Italian military had an annual incidence ranging from 8 to 13.5/100,000, higher than that observed in the age- and sex-matched civilian population, with an average annual incidence of 10.4/100,000 [93], whereas in the period 2008–2018, the annual incidence was always lower than 1/100,000, except for in 2013 and 2017, when it was 1.68/100,000 and 2.1/100,000, respectively, with an annual average incidence of 0.675/100,000 and a reduction of 15.4-fold (Table 5). Only six NATO countries maintain the BCG for the military; however, in only two countries, it is administered to the whole military personnel; in one, it is only recommended, and in the last three, it is administered to selected categories of personnel, who are exposed as a consequence of occupational risk or deployed in high-risk areas [36]. A rising problem is the multi-drug-resistant TB (MDR-TB), caused by isoniazid- and rifampin-resistant *Mycobacterium tuberculosis*; this issue has been considered of awareness for the military, not only because of the difficulties in the treatment of patients with MDR-TB but also because drug-resistant *Mycobacterium tuberculosis* is a pathogen included among the biological agents, category C in the Centers for Disease Control and Prevention (CDC) classification, and studies aimed at counteracting its infection are of strategic interest. In conclusion, even for TB, the role of military physicians in the demonstration of the infectious nature of the disease and the prophylactic vaccine, as well as in its epidemiology, especially in wartime, witnesses the interest of the military and the contribution the military provided.

### 2.7. Meningococcal Meningitis

Meningococcal meningitis is a serious, potentially lethal, and invalidating disease, caused by the Gram-negative microorganism *Neisseria meningitidis*, which is transmitted through airborne droplets and was identified by Weichselbaum in 1887 [94]. Based on the chemical characteristics of the polysaccharide capsule of the microorganism, 13 serogroups are known, six of which may induce invasive meningococcal disease (IMD) in humans, A, B, C, W_135_, Y, and X. It is estimated that the annual global cases of IMD are at least 1,200,000 and the annual global deaths 135,000 [10] (Table 1). In the pre-vaccine period, the highest disease prevalence was observed in infants and people living in communities, particularly in the first days of community life, such as college students and the military. Hence, the particular interest of the military in this dreadful disease and the successful efforts in identifying the protective role of specific antibodies, the type of immune response, and a vaccine, by the researchers of the Department of Bacteriology of the Walter Reed Army Institute of Research (WRAIR) in the 1960s [95,96,97,98,99]. Meningococcal meningitis has been described as a severe disease in the military since the nineteenth century both in peacetime and wartime. It struck the Prussian Army in 1806–1807, the French Army in Algeria in 1840, different European countries and the USA, which were particularly hit during the American Civil War (1861–1865), and, since 1875, it has spread worldwide [100]. During WWI and WWII, meningococcal meningitis was a relevant problem for all armies. In the first year of WWI, 150/100,000 meningococcal meningitis cases occurred in the US Army, with a case-fatality rate of 39%, whereas during WWII 14,000, cases were described in the US Army; however, the case-fatality rate was reduced to 4%, as a consequence of early diagnosis and the availability of anti-bacterial drugs [101]. During WWII, the only available treatment was sulfa drugs, discovered in 1937, but by the first half of the 1960s, most meningococci were resistant to the sulfa drugs [102]. Thus, the search for an effective vaccine was pushed by the awareness that the most effective preventive tool was active immunization. The C polysaccharide vaccine, introduced in 1972 in the US Army, provided 87% of protection [103]; this vaccine was in 1979 replaced by the bivalent A + C, and in 1983 by the tetravalent (A, C, W_135_, and Y). Compared with the pre-vaccine era, the vaccine introduction reduced morbidity by over 90%, whereas the case-fatality rate did not result to be significantly modified, always remaining around 7% [101]. In Italy, the burden of meningococcal meningitis in the military became particularly relevant during the 1980s (in 1985, an incidence of 17/100,000 cases, 92% serogroup C, and in 1986, an incidence of 7/100,000, 95% serogroup C, were observed, compared with an incidence of 0.8/100,000 in the general population [104,105]); thus, the bivalent A + C vaccination was introduced since 1 January 1987. Vaccination was effective in reducing the burden of meningococcal meningitis A and C, showing an effectiveness of 91.2% [104,106], an immunogenicity of 84% and 91% of protective seroconversion for polysaccharides A and C, respectively, with the appearance of mainly oligoclonal specific antibodies, and safety [107]. In 1991, the tetravalent polysaccharide ACW_135_Y vaccine was introduced, recently largely replaced by the protein–conjugate formulation. However, the tetravalent polysaccharide vaccine maintains its validity because of its good immunogenicity and the long durability of induced antibody response, which were recently examined at 9 months [108] and 5 years [109]. In the French military, the vaccine was introduced in 1996, and 2 years later, its protective effectiveness was calculated to be 100% [110]. Currently, the tetravalent vaccine ACW_135_Y is included in the vaccination program of 24/25 NATO countries which have replied out of the 30, in 10 countries for the whole military personnel and in the other fourteen for selected categories [36].

A vaccine based on polysaccharide antigen could not be pursued only for meningococcal polysaccharide B, considering that polysaccharide B has a chemical structure close to human brain polysaccharide, thus resulting in being poorly immunogenic or, even worse, auto immunogenic [111]. Therefore, the approach for obtaining an effective anti-B vaccine was long, laborious, and not based on the use of polysaccharides as antigens; rather, through the innovative approach of reverse vaccinology, a recombinant protein vaccine was achieved only in 2005 [112]. This vaccine proved to be mildly moderately reactogenic in infants, particularly when administered in association with other vaccines; however, it was proven that the concurrent administration of paracetamol significantly reduced reactogenicity without interfering with the immune response [113]. Due to the relative rarity of IMD, not many significative studies on efficacy in the pre-registration phase have been carried out, however, the vaccine has been approved based on its immunogenicity [113]. The effectiveness in preventing IMD has been demonstrated in the real-world [114]. Vaccination with meningococcal B vaccine has been included in the national immunization program (NIP) of the UK, Ireland, and Italy; 12 European Member States have made an assessment to include the vaccine in the NIP, three are recommending the vaccine without reimbursement, whereas five are not recommending, as of March 2015 [115]. Only 5/25 NATO countries, which had reported the respective military vaccination program, declare having meningococcal B vaccine included in their vaccination program for the military; in two cases the vaccine is only recommended, whereas in the remaining three, it is compulsory for selected categories of military personnel [36]. Even though sporadic cases are still observed, the vaccine introduction induced a substantial reduction of IMD in both civilians and militaries [116]. In Italy, the anti-meningococcal polysaccharide vaccine has been introduced in the compulsory vaccination program for the military thirty years before its availability for free in infants; however, the meningococcal B vaccine has been freely offered to infants since 2017, and it has not been included in the vaccination schedule for the military yet.

### 2.8. Hepatitis A

Hepatitis A is a disease caused by an RNA virus (HAV), transmitted via the fecal–oral route, by contaminated water and food, that easily spreads in poor hygienic conditions and overcrowding. It was so largely widespread in the military, both in peacetime and mainly in wartime, that it was even known as “camp jaundice” [117]. In 2019, the global annual infections were estimated to be 158,944,000, an increase of nearly 14% compared with 1990, and the annual deaths were 39,280 [24] (Table 1). Poor hygienic conditions and overcrowding as risk factors for the military were present in the literature up to 1990, whereas in more recent times, the major risk factor for the military has been the deployment to countries of high endemicity [118]. One epidemiological study in Italy in the decade 1987–1997 revealed a similar incidence in the military and the age- and sex-matched civilian population [93]. Moreover, a study carried out in the Italian military in 2003 documented that Italy passed from a prevalence of 66.3% positive subjects for anti-HAV antibodies in 1981 to 5.3% in 2003, thus from high to low HAV endemicity in 20 years, with the military reflecting the epidemiology of the general population [119]. A similar behavior of anti-HAV seroprevalence was even observed in the French military [120]. In the Italian military, the annual incidence in the period 1986–1997 ranged from 5 to 60/100,000, with an average annual incidence of 17.5/100,000 [93], whereas in the period 2008–2018, it ranged from 0.35 to 0.66/100,000, an average annual incidence of 0.5/100,000, a reduction of 35-fold (Table 5). In 1953, for the first time, the definition of hepatitis A and hepatitis B, to identify the infectious (shorter incubation time, fecal-oral transmission, better prognosis) versus the serum-transmitted (higher incubation time, serum transmission, worse prognosis) hepatitis, respectively, was reported by an expert committee of the WHO [121]. However, until 1942, when an outbreak of acute viral hepatitis involving nearly 50,000 US Army personnel following yellow fever vaccination [122], no clear idea that at least two types of hepatitis could occur was still present: only the study of this outbreak, and the clarification that the outbreak was not due to a side effect of yellow fever vaccine, but to the preparation of the vaccine with human serum contaminated with virus hepatitis, has allowed a better comprehension of acute hepatitis to be achieved.

A disease with the characteristics of epidemic or infectious jaundice was described during the British–American War of 1812, but especially during the American Civil War, when 87,326 cases of jaundice were recorded by the Medical Corps of the Union Army [123]. In WWI, epidemic jaundice represented a relevant problem for the French, British, and German armies, whereas this was not the case for the US Army, and in WWII, the US Army registered over 180,000 cases of infectious jaundice, with a case-fatality rate of 0.3% [124]. However, following the occupation of Italy and Germany, where infectious jaundice was endemic, the US military registered an increase in cases, which in Italy reached the incidence of 37/1000 and in Germany continued even after the end of the war [116]. This observation allowed the first epidemiological studies to be carried out by US researchers in a newly established hepatitis center in Bavaria [115]. During the Korean War in 1950–1951, in a country of high endemicity, the cases of jaundice in the troops hospitalized or isolated were over 4000 [124].

During WWII, US military researchers demonstrated the protective role of the pooled gamma-globulin plasma fraction against epidemic jaundice [125]. During the Korean War, a randomized double-blind study driven by US military researchers on intramuscular IgG administration to soldiers could establish that the passively immunized subjects resulted protected from hepatitis A, B, and non-A non-B for 6 months [126]. Even though passive immunization has been used for a long time for the protection of travelers and military personnel, more recently an inactivated vaccine was developed by US researchers of WRAIR in collaboration first with Robert Purcell at the National Institute of Health [NIH] and later with SmithKline Beecham (SKB), now GlaxoSmithKline. This vaccine proved to be safe, immunogenic, and highly protective (94% after two doses) in a large phase III study in Thailand on approximately 20,000 individuals and 20,000 controls who had received hepatitis B vaccine [127]; based on these results, the vaccine was approved by the Food and Drug Administration (FDA) in 1995 [116]. The vaccine, administered in two doses 6 months apart not only demonstrated to be highly immunogenic but even effective, by inducing a long, probably a life-long, protection. The persistence of anti-hepatitis A antibodies following vaccination is so long that in a recent study, the durability of vaccine-induced antibodies could not even be calculated because the curve representing mean antibody titers was slightly ascending in joining the levels found at 9 months and 5 years post-vaccination [109]. HAV vaccine has been introduced in the military vaccination program of all 25/30 NATO countries, which have reported the vaccination program for the military: in 12/25 countries, the vaccination is indicated for all the military personnel and in the remaining 13 countries for selected categories [36]. Currently, HAV infection, which has historically represented a real obstacle to the operational readiness of the military, does not represent a problem for the military anymore, not even when deployed to high endemicity countries. The military contribution in the fight against hepatitis A has been crucial for epidemiology, the demonstration of protection by passive immunization with human immunoglobulins, and vaccine development.

### 2.9. Hepatitis B

Hepatitis B is a disease caused by a DNA virus (HBV), which may cause either acute or chronic disease. Chronic disease may eventually induce liver cirrhosis and/or hepato-carcinoma. The disease is highly contagious and may be transmitted by contaminated blood and blood derivatives, sexual route and perinatally. The diagnosis may be made by identifying: the surface antigen (HBsAg) of the virus released in biological fluids; the antibody response to viral antigens (serum antibodies to the viral core (HBcAb), surface (HBsAb), and/or envelope (HBeAb) antigens); or by amplification of viral genes by polymerase chain reaction (PCR) at serum and hepatic levels. It is estimated that, worldwide, approximately 2 billion people have come in contact with HBV [128] and the WHO estimates that in 2019, 296 million people were living with chronic HBV infection; each year 1,500,000 new infections and 820,000 deaths occur, the majority from severe sequelae of hepatitis B, such as cirrhosis and hepatocarcinoma [12] (Table 1). This blood-borne disease is of interest for the military, considering that wounds may be a source of contagion and whole blood transfusions are used as resuscitation tools, consequently, the need that the soldiers are “walking blood banks”, thus free of blood-borne viruses, such as HBV, hepatitis C virus (HCV), human immunodeficiency virus (HIV) types I and II, and human T-cell lymphotropic virus (HTLV) types I and II, is imperative [129].

A pre-enrollment screening for blood-borne viral infections to prevent admission to the military seems, therefore, the best preventive measure. However, in a survey carried out by the WHO in 1998 in over half of the countries reporting to the WHO (107/193), only 76 replied; of these, 53 declared having a central military laboratory to perform the screening of the recruits, 27/53 (51%) for HIV, 17/53 (32%) for HBV and 7/53 (13%) for HCV [7]. Currently, the situation is probably improved, even in consideration that in 1991 in different world countries, the compulsory HBV vaccination for infants was introduced; thus, in the last decade, the applicants for military service had generally been vaccinated in infancy. The vaccine, which was made available as plasma-derived in the first half of the 1980s, and, since 1986 as recombinant, is effective and, after having completed the whole vaccination cycle (three administrations), provides a long, probably life-long, protection [130]. Moreover, in 24/25 NATO countries hepatitis B vaccine is present, in 15 for the whole personnel and in 9 for selected categories [36]. However, in some NATO countries, in which the access to HBV vaccination in infant age has been delayed, the prevalence of serum HBV infection markers was still quite high in the first decade of this century [131,132].

The combined influence of entry screening, awareness of the risk of infection due to sexual activity as a consequence of the HIV infection prevention programs, and vaccination has determined a rate of infection slightly lower in the US military (0.23%) than in the corresponding civilian population (0.3–0.5%) [133]. The influence of vaccines may be inferred by the significant difference between the rate observed in the older cohort, born before 1979, generally not vaccinated, and the rate observed in the younger cohort, born in or after 1979, generally vaccinated, 0.39% vs. 0.13%, respectively (*p* = 0.016, Yates corrected, two tails, χ^2^). Conversely, the influence of social factors and fear of HIV infection may be observed in the dramatic decline, in less than a decade, of HBV markers in two Italian military populations of approximately 5000 individuals each, the first from the Italian Navy analyzed in 1981 and the second from the Italian Air Force analyzed in 1990. HbsAg and HbcAb were 3.4% and 16.8%, respectively, in 1981, whereas they declined to 1.6% and 5.8%, respectively, in 1990 [134]. Even a study of incidence in the same period on approximately 1300 Italian students at a military school located in the Italian region with the highest prevalence of HBV infection, followed-up for eight months, showed seroconversion to HBV markers of only two subjects (0.24/100 person-years of exposure), thus witnessing a low spreading of HBV markers among the Italian recruits [135]. In the Italian military in the period 1986–1997, the annual incidence of HB ranged from 7 to 33/100,000, with an average annual incidence of 19/100,000 [92], whereas in the period 2008–2018, only four cases have been reported, an annual incidence ranging from 0.32 to 0.65/100,000 cases, an average of 0.44/100,000, a 43-fold reduction (Table 5). This epidemiological situation and the consideration that currently the cohorts of recruits have been previously vaccinated when entering the military life are at the basis of the decision of the Italian military health authorities to eliminate the HBV vaccine from the military vaccination schedule, thus avoiding an expensive, useless, and unjustified booster.

### 2.10. Poliomyelitis

Poliomyelitis is a severe disease caused by an enterovirus, of which three types, 1, 2, and 3, are known. The disease may be transmitted through the nasopharynx, through an oral–oral way, by feces, or through a fecal–oral way, and after infection, the virus enters the bloodstream. This virus is highly contagious, and up to 100% of households may be infected, but in 95%, the infection runs asymptomatically or pauci-symptomatically, whereas in the remaining 5%, the symptoms are characterized by fever, headache, fatigue, nausea, vomiting, and neck stiffness, for meningitis. In some subjects, the virus, which has a marked neurotropism, localizes at the spinal level, most frequently in the anterior horn cells of the cord, thus eventually determining an asymmetric flaccid paralysis, particularly in the arms. More rarely, the virus may localize at the bulbar level, thus compromising vital functions, such as circulation and respiration, with consequent high mortality [136]. The case-fatality rate of paralytic cases was 2–5% for children and 15–30% for adults [137]. The disease in the pre-vaccine era was largely widespread worldwide; in 1956, the inactivated trivalent vaccine developed by Jonas Salk was introduced, whereas in 1962, it was largely replaced by the oral, living, vaccine developed by Albert Sabin [136]. The use of vaccines has allowed the disease spread to be dramatically reduced; however, in 1988, the WHO decided to start an eradication campaign with the objective to eliminate the disease by the year 2000. Despite that the eradication campaign could not achieve eradication by 2000, the 350,000 estimated cases in 1988 were reduced to 3000 in the year 2000 [138]. Currently, the viral types 2 and 3 have been declared eradicated; thus, the wild virus is only type 1, which is still present in Afghanistan and Pakistan, where in the last years, it has even increased [139], and sporadic cases are reemerging in other politically unstable countries and sometimes sites of conflicts, such as Syria, Iraq, Cameroon, Equatorial Guinea, Ethiopia, Kenia, Nigeria, and Somalia [140]. In the process of eradication, in addition to the difficulties created by war and political instability, a further complication derives from the fecal elimination of a vaccine virus in countries where the oral, living vaccine is, or was, used. The live attenuated vaccine virus may revert to virulence; thus, being able of induce paralytic polio in vaccine recipients, particularly in those with immunodepression [139]. All these difficulties may delay the date of eradication; consequently, vaccination should be maintained at least until eradication.

Although the disease has been known for a long time, with the first evidence identified approximately 1500 years BCE, poliomyelitis did not induce outbreaks until the end of the nineteenth century, when outbreaks of infantile paralysis occurred in Scandinavia and the USA [141]. The disease was not considered relevant for the military, because it scarcely occurred in adults, and even during WWI, no outbreaks were described, despite the poor hygienic conditions and sanitation. However, in the interwar period, the cases of poliomyelitis in adults increased, and in the course of WWII, the US military registered 1023 cases with over 20% of deaths [142]. Out of the 1023 cases, 446 occurred in the troops who remained in the USA, whereas 577 occurred in the troops deployed overseas, in particular in Egypt, Italy, and the Philippines. Although these figures do not appear so high if compared with another severe, “military”, infectious disease such as meningococcal meningitis, for which 14,000 cases were described during WWII with a case-fatality rate of 4%. Polio had over 20% of mortality, 42% of discharge for disability and was the infectious disease with the highest number of lost days, with only 34% of infected military returning to duty, figures not comparable with other infectious diseases [142]. Nonetheless, polio has never been considered a “military” infectious disease, and vaccination is maintained only to make the military ready to be deployed everywhere, even in countries such as Afghanistan, where wild poliovirus is still circulating, and yearly cases due to poliovirus type 1 are notified. Out of the 25/30 NATO countries reporting the vaccination program for the military, all maintain an inactivated polio vaccine, 16 for all the military personnel and nine for selected categories [36]. The vaccine-induced antibodies are well stimulated by inactivated vaccine even though the priming is carried out with oral vaccine [139], and their durability above the threshold for protection has been calculated in 10–20 years for anti-type 1 and 3 antibodies [109], data in line with the literature [143]. Maintaining the anti-polio inactivated booster for the military creates ulterior protection to prevent the possibility that soldiers returning from a mission to endemic areas become involuntary carriers of wild poliovirus; moreover, it is a relevant measure of public health, because it reduces the viral circulation, thus contributing to the eradication campaign of the Global Polio Eradication Initiative.

### 2.11. Measles

Measles is a disease caused by a virus derived from the agent of the cattle rinderpest, which adapted to humans 5000–10,000 years ago [144]. It is air-borne transmitted and is highly contagious (one infected person may infect on average 9–18 susceptible individuals, more than the smallpox virus, which may infect 5–7 susceptible individuals, and influenza, which may infect 2–3 susceptible subjects). The disease is characterized by fever, cough, coryza, maculopapular rash, and conjunctivitis; however, the virus is carried by lymphocytes and may localize in the lymphoid tissue and everywhere in the body, with possible severe complications, such as pneumonitis, keratoconjunctivitis, and encephalitis. The infection of lymphocytes causes a transitory immunodepression, and the measles virus was the first infectious agent for which induced immunodepression was demonstrated. The Nobel Laureate John Franklin Enders developed the first live attenuated vaccine in 1960 [92]. Measles was responsible for over 2 million deaths annually in the pre-vaccine era, but even now, it is still responsible for over 100,000 deaths per year. In 2015, the global annual cases were estimated to be over 9,700,000 (only 245,928 cases reported), and the global annual deaths were 134,200 [11] (Table 1). Despite the RNA genome being generally characterized by a high rate of mutations, both the wild virus and vaccine strains are stable, making it not necessary to update the vaccine to a newly circulating mutated virus, as required for the influenza vaccine. The inclusion of this live attenuated vaccine in the Expanded Program of Immunization (EPI) in 1980 contributed to the reduction of measles morbidity and mortality, particularly in areas such as Sub-Saharan Africa, with the highest morbidity and case-fatality rate [145]. Measles eradication by a global immunization program is in theory possible, as the vaccine is effective and no animal reservoir is known. However, the deadline of 2010 for its eradication set by WHO in the European region was not respected, and to date, measles prevalence is still quite high (11%) in this area, whereas in some countries, an increase was reported after 2010 [145,146]. Various causes can be hypothesized for this failure, including the vaccine hesitancy caused by the publication and diffusion on mass media of the false association of measles/mumps/rubella (MMR) vaccination with autism, which led many parents to not vaccinate their children [92].

Measles has represented a relevant problem, even for the military, particularly up to the twentieth century. For example, in the American Revolutionary War and the American Civil War, measles was one of the main causes of death among the soldiers [52]. During the whole Civil War, measles caused 67,763 cases and 4246 deaths (case-fatality rate of 6.27%) in the Union Army [147]. The case-fatality rate was 6% and 11% for white and black soldiers, respectively [148]. A reduction in the impact of measles on the US military in the war was observed in the following years. The morbidity in the Union Army in the first 2 years of war (1861–1862) was 56/1000, with a case-fatality rate of 2/1000 [149]. Morbidity (42/1000) and mortality (0.45/1000) caused by measles decreased in the first 2 years (1898–1899) of the Spanish–American War. In 2 years of WWI (1917–1918), the reported morbidity was 28/1000 and the mortality 0.7/1000. The progressive reduction of morbidity and mortality was confirmed in WWII when over 300,000 admissions to hospitals were registered for measles, mumps, rubella, and varicella [52], but a limited number of US soldiers died of measles. Finally, during the Vietnam War, no death to measles was registered among US soldiers. The progressive reduction of cases and deaths for measles cannot be explained by medical progress, considering that no immunoglobulins or antibiotics for the possible bacterial super-infections, nor vaccines, were still available in the first phase of observed reduction. A possible explanation that has been proposed for this phenomenon is the epidemiological isolation of recruits. In the nineteenth and first years of the twentieth century, the majority of soldiers were enrolled from rural, isolated areas, where the possibility to acquire measles and natural immunization in infancy was scarce. Overcrowded barracks, tents and battle camps forced young men coming from different areas of the country to live in close contact, creating the best conditions for viral spread among susceptible individuals upon the emergence of new cases [148]. Measles, in the first part of the last century, was mainly complicated by bacterial pneumonia, more often caused by *Streptococcus haemolyticus*, currently known as *Streptococcus pyogenes*, largely present in apparently healthy carriers, and able to induce pneumonia, and sometimes empyema, in a respiratory tree already damaged by measles virus [150]. In 1915, the Highland Division of the British Army suffered a measles outbreak associated with scarlet fever; out of 529 soldiers with measles, 65 died, a case-fatality rate of 12%, greater than that observed during the American Civil War [151]. In 1917, measles and pneumonia were responsible for 30% of all USA deaths in the troops [52]. This same paradigm of a bacterial super-infection on a viral disease was repeated in 1918 with the Spanish influenza pandemic, whose high mortality was largely dependent on the bacterial super-infection, with severe cases of pneumonia, which were frequently lethal in the pre-antibiotic era. Conversely, prior to the twentieth century, measles-associated deaths were mainly due to lethal gastrointestinal complications and a hemorrhagic illness known as black measles [152].

Even in the post-vaccine era, the military, due to the high contagiousness of the disease and the community life, which is characteristic of the military population, seem to be more exposed to measles than the general population, as observed in Italy in the period 1986–1997 [93], and France in 2011 [153]. This observation pushed the military medical authorities in Italy and in France to introduce the compulsory measles vaccination in the trivalent formulation MMR, which was developed by Dr. Maurice Hilleman in Merck, after leaving WRAIR (in Italy, it was introduced in the military vaccination program in 1998 [48], whereas only in 1999 was vaccination offered for free to infants, to become compulsory only at the end of 2017, following a large measles outbreak in January of the same year [154]). In Italy, the effectiveness was found to be 95% [154], and even the immunogenicity was good, considering that 96% of vaccinees showed post-vaccine protective antibody levels [140]. However, the high prevalence of pre-vaccine antibody positivity, probably due to natural immunization [140], induces to believe that pre-vaccine screening may be the best policy to adopt, such as in the USA [52]. In Italy, in the period 1986–1997, measles annual incidence ranged from 70 to 1300 cases per 100,000, with an average annual incidence of 671/100,000 versus an annual incidence ranging from 0.33 to 4.2/100,000 in the period 2008–2018, an average annual incidence of 1.31/100,000, and a 512-fold reduction (Table 5). This epidemiological situation probably reflects not only the effectiveness of MMR, which was introduced in 1998, but even the socio-environmental transformation due to the passage, in 2005, from the mandatory conscription to the professional army. This resulted in a reduction of occasions of disease transmission, consequent to the reduction of the number of military personnel, but mainly to the marked reduction of the requirement for the soldiers to live in barracks, a rule that has remained limited to training or operational military personnel. Currently, 23/25 NATO countries reporting the military vaccination program maintain the measles vaccination, in 18 countries for all the military personnel and in five for selected categories of personnel [36]; in all these countries, the administered vaccine is the trivalent MMR. Despite that measles responds well to the vaccine, such that it does not represent a severe risk for public health in most countries anymore, the eradication process is quite hard to reach, even in some European countries [155]; thus, guard must remain high, even because there is the awareness that the disease-induced protection is lifelong, whereas the vaccination-induced protection is not, and currently, there is an open discussion on how many boosters are needed, in addition to the two already accepted vaccine administrations [156], for maintaining protection in the different environmental conditions [140]. The military are particularly exposed because of being a close community and because of operational activity, which may put them in contact with under-vaccinated populations where the virus is still highly circulating; thus, it should be desirable that the military is always updated with this vaccination, even by periodical checks, which may verify the state of immunization [157]. However, a measles outbreak has been recently reported even in a highly vaccinated population [158].

### 2.12. Mumps

Mumps is a disease caused by the Mumps virus, a member of the *Paramyxoviridae* family in the genus *Rubulavirus* that naturally infects only humans. Mumps generally has mild clinical course, characterized by swelling of salivary glands, especially parotid, accompanied by fever, headache, and malaise, but complications such as aseptic meningitis in up to 10%, orchitis in approximately 25% of post-pubertal male subjects, pancreatitis, deafness in approximately 4% of subjects, and rarely encephalitis, which may induce permanent disabilities or even death, may occur [159]. The infection is transmitted with moderate-high effectiveness by respiratory route, is only observed in humans, and has an incubation time of 2–4 weeks with a clinical course of 1–2 weeks [160]. A live attenuated vaccine has been developed in the 1960s [161]; it may contain different viral strains, with major or minor reactogenicity/efficacy, and it is generally administered in a combined formulation, similar to MMR. One mostly used strain, because of its safety and efficacy, is named Jeryl Lynn, after the daughter of Dr. Maurice Hilleman, who isolated the virus from her throat and prepared the attenuated vaccine strain.

In the eighteenth century, mumps was known and occurred worldwide, particularly in crowded environments such as in schools, colleges, prisons, and military barracks [162], with an annual incidence of >100/100,000 [159]. However, in the military, an even higher incidence of 6000/100,000 was observed [163]. In the first year of WWI in the USA, mumps spread explosively when recruits coming from rural areas or cities of the USA were assigned together in military barracks [164,165]. Recruits from rural areas perhaps had fewer probabilities compared to recruits from cities to come in contact with infectious agents and acquire natural immunization at infancy; thus, they were more susceptible to this and other viral infections. The epidemics followed a periodical trend, with a period of approximately 3 years and a higher peak during WWII [166]. In the USA, mumps vaccine was initially made available in 1967 to specific categories; then, from 1968 to 1977, it was gradually extended to all children of 12 months of age. The annual mumps incidence from 88/100,000 in 1968 decreased to 2.5/100,000 in 1982 with a net reduction of 97% [166]. Despite two periods of mumps resurgence in the decade 1983–1992 and in the 15-year 1993–2008 period, generally occurring in schools and colleges of rural USA populations, no resurgence was observed in the military, probably for the vaccination policy of the military with MMR since 1991. A crucial point is the choice of the vaccine, considering that some vaccine strains are effective but poorly attenuated, such as Urabe Am9, which was responsible in the Italian military for a post-vaccine outbreak due to the vaccine strain, as molecularly demonstrated [167]. The vaccine-induced mumps for scarce vaccine strain attenuation may possibly be one of the reasons for finding 70% of mumps vaccine efficacy, compared with 95% of vaccine efficacy for measles and rubella [168]. However, the mumps vaccine effectiveness is quoted ranging from 69% to 88% [169], and a mumps outbreak has been reported in a French military Parachuting Unit in 2013, in the majority vaccinated with two MMR doses, characterized by a high attack rate, ranging from 21.6% to 25% [170]. The mumps occurrence in highly vaccinated populations is a well-known phenomenon even in other countries [171,172], and different hypotheses have been proposed for its interpretation, including early waning of immunity or antigenic variance that may reduce the efficacy of the vaccine against new circulating strains, as frequently observed with influenza vaccine [172]. Another crucial point is the number of boosters that should be administered for maintaining antibody levels above the threshold for protection. Currently, it has been established that in countries where two vaccine doses at approximately five years of distance are administered in infancy, immunization is protective with an effectiveness of over 99% of disease reduction, a percentage higher than that observed in the countries where vaccination schedule is based on only one vaccine administration [159]. However, the need for further booster(s) is still a matter of discussion and has not been established yet. Only one mumps case has been reported in the Italian military in the decade 2008–2018, an incidence of 0.32/100,000; thus, it has virtually disappeared, whereas, in the period 1986–1997, when the MMR was lacking in the compulsory vaccination schedule for the Italian military, it ranged from 25 to 65/100,000 cases, an average annual incidence of 45.5/100,000 [93]. The ratio of reduction is over 142-fold (Table 5); however, for this dramatic reduction, the same considerations spent for measles on the passage in Italy from mandatory conscription to professional army in 2005 are valid. Considering that the administered vaccine is MMR, among the 25 NATO countries reporting the respective military vaccination schedule, the mumps vaccine, similar to measles, is administered in 23 countries, in 18 of them for the whole military and in five for selected categories [36].

### 2.13. Rubella

Rubella is a viral disease caused by *Rubivirus rubellae*, a member of the genus *Rubivirus*, with a generally mild clinical course, rash and lymphadenopathy, mainly at nuchal level. The major complication of rubella is fetus infection, which may provoke miscarriage or congenital rubella syndrome (CRS), a severe condition characterized by congenital ocular, hearing, heart, brain, or endocrine disabilities [173]. Despite that an effective live attenuated vaccine has been developed by Dr. Maurice Hilleman in the 1960s [174], CRS is still present with approximately 100,000 cases per year [173]. The disease has no animal reservoir, has an effective vaccine and has been eradicated in the Americas since 2009 and in Great Britain; thus, it is an optimal candidate for global eradication, even in consideration of its lower transmissibility compared with measles, provided that a suitable percentage of herd immunity, which may range from <70% to >90% according to the different world areas, is achieved and maintained [173].

The interest of rubella for the military is not only witnessed by the outbreaks observed in wartime and peacetime, but even and especially by the fact that the virus was first identified and isolated by military researchers of the WRAIR in the US Army recruits in 1961 [175]. The rubella vaccine was adopted in the US Army in 1972 [52]. The effect of vaccine introduction in reducing rubella cases was dramatic. In the three years before vaccine introduction, the number of rubella cases notified in the USA was 47,745, whereas in 2005, the CDC announced that endemic rubella was eradicated in the USA [124]. In the Italian military, rubella showed an over four-fold incidence increase in the period 1991–1995 compared with the period 1976–1980 (1150/100,000 vs. 280/100,000, respectively), with an annual incidence ranging from 50/100,000 to 2300/100,000 in the period 1986–1997 and an average annual incidence of 936/100,000 [93], whereas in the period 2008–2018, only 11 cases have been registered, 10 of which were in 2008, an incidence of 3.32/100,000, and one in 2013, an incidence of 0.33/100,000, and an average annual incidence of 1.825/100,000, a reduction of 512-fold (Table 5). However, for this dramatic reduction, the considerations spent for measles and mumps on the passage in Italy from mandatory conscription to professional army in 2005 should be taken into account. Even for rubella, the situation in the NATO countries is identical to the situation reported for measles and mumps, with 18 countries using MMR for the whole personnel and five for selected categories of military personnel [36]. MMR vaccination in the military is a relevant measure of public health even in countries where MMR is provided in infancy, where, acting as a booster, it contributes to reducing viral circulation.

### 2.14. Varicella

Varicella or chickenpox is a disease caused by a DNA herpesvirus that generally induces a mild disease, characterized by fever, malaise, and vesicular erythema. The disease has high transmissibility, with an R0 estimated at around 10–12 [176], by airborne route of the virus coming from skin vesicles [177]. Varicella only occurs in humans and is present at global level, with an average annual incidence of 13–16/1000, but greater than 100/1000 in the <9-year-old children [177]. However, this epidemiological pattern is generally observed in temperate areas, because in tropical areas, the adult age is more frequently represented [177]. The clinical course tends to be self-limiting in children, whereas it may be complicated in adults in pregnancy, in which varicella may cause fetal malformations (congenital varicella syndrome) in approximately 1% if infection occurs in the first two trimesters, and in immunosuppressed people, in whom it may be responsible for death in up to 15–18% [177,178]. In 1974, Takahashi developed a live attenuated vaccine [179], which has been shown to be safe and effective. Its systematic use with two doses has deeply modified the disease epidemiology, with a reduction of over 95% of incidence, hospitalizations, and deaths in children in the USA [177].

Varicella is highly contagious; thus, it has represented a problem for the military in the pre-vaccine era, in analogy to measles, rubella, mumps, and pertussis. Even though most recruits are protected when they join the military, nonetheless, some dozen cases occur each year, as in Israel [180], considering that vaccine-induced seroprotection seems to be lower than disease-induced protection [181]. In some countries, a marked increase in varicella infection has been observed in the military between the 1970s and the 1980s [182] or between the 1970s and 1990s [93]. In the Italian military in the period 1986–1997, the annual incidence ranged from 800 to 1900/100,000 cases with an average annual incidence of 1300/100,000 [93], whereas it ranged from 2.4 to 12.6/100,000 in the period 2008–2018 with an average annual incidence of 7.29/100,000, a reduction of 178-fold (Table 5). This seems more a probable expression of the transformation of the military service in Italy than of the effect of vaccination, considering that in 2005, compulsory conscription was substituted with the professional army. In Italy, similar to many other countries, even though vaccination is mandatory, in practice, it is only administered to those who do not refer having suffered the disease or carried out vaccination in infancy, a method that does not appear as reliable, particularly in the presence of negative history [183]. Moreover, vaccination is applied in only 10/25 NATO countries reporting the military vaccination schedule, in half of which is either compulsory for all the military personnel or compulsory/recommended for selected categories of personnel [36]. Finally, in some countries, the percentage of susceptible recruits to varicella is quite high, of the order of 50% in the current period [178]. In conclusion, despite the availability of a safe and effective tool for varicella prevention, it appears that the vaccine is not as largely used in the military and, even when it is used, the policy to limit vaccine administration to those lacking documentation of infant vaccination or disease may reduce its impact on disease prevention. This may probably explain why the reduction rate of varicella is lower than the reduction rate of measles and rubella in the Italian military.

### 2.15. Influenza

Influenza is an acute respiratory disease that is transmitted by respiratory route, characterized by fever, cough, myalgias, and a generally benign clinical course of approximately 2–8 days. However, sometimes, particularly in children less than 5 years of age, older adults, subjects with underlying diseases, and in pregnant women, influenza may be complicated, mainly by pneumonia and even by multi-organ failure, with possible hospitalization and death [184]. The WHO estimates that annually approximately 1 billion people become infected with seasonal influenza, with approximately 3–5 million severe influenza and 300,000–500,000 deaths [9] (Table 1). The etiological agent is a highly mutant RNA virus, of which four types are known, A, B, C, and D, the first being responsible for epidemics and pandemics, and the most severe clinical forms [185]. All four types may be found in humans and other animal species, such as swine, horses, dogs, seals, bats, and the largest reservoir, represented by wild aquatic birds [186]. The virus A expresses in its surface two proteins, hemagglutinin (H), responsible for the infection, through the attachment to the corresponding receptors on respiratory cells, of which 18 subtypes are currently known, and neuraminidase (N), responsible for detachment from cell to infect other cells, of which 11 subtypes are known [186]. Type B, which may be responsible for epidemics, C, which has been associated with mild symptoms, and D, which has not been associated with pathology in humans, may be found in animals and humans. Currently, two A strains are circulating, H1N1 and H3N2, and two B subtypes, B/Yamagata and B/Victoria [185]. Hemagglutinin and neuraminidase, as first observed by Dr. Maurice Hilleman at WRAIR in 1957 [29], undergo annual slight antigenic modifications, defined “drifts”, and periodic marked antigenic transformations, defined “shifts”, which are responsible for pandemics since the immune system does not recognize the brand-new antigen. In 1918, a terrible influenza pandemic, called “Spanish flu” started inside the USA military, at the training camps of recruits of the American Expeditionary Force (AEF), due to the strain H1N1, which was responsible for an estimated infection of one-third of humankind and death of approximately 50 million subjects, with a case-fatality rate of over 2.5% vs. 0.1% observed in other pandemics [187]. In 1957, a new pandemic, due to the strain H2N2, defined as “Asian flu”, was responsible for approximately 1.5 million deaths, followed in 1968 by a new pandemic, due to the strain H3N2, defined as “Hong Kong flu”, which was responsible for approximately 1 million deaths. Finally, in 2009, a new pandemic, due to a swine strain H1N1, was responsible for an estimated 300,000 deaths [188].

The “Spanish” influenza pandemic was the worst. It deeply hit the military, at the beginning the US military, and afterward the military and the civilian populations of other countries, including different European countries, Africa, India and Asia, Australia and New Zealand [189]. However, the rate of infection was always higher in the military than in the corresponding civilian population [190]. This pandemic developed along three successive waves, starting in spring 1918 with a relatively mild disease and then proceeding to fall and winter–spring 1919 with two highly lethal waves. The high lethality was observed not only in the extreme life’s ages, as in other influenza epidemics or pandemics, but also in young adults. This wide distribution of lethality had a dramatic demographic and economic impact on the working and productive sectors of the interested population, higher than the war itself [187]. The virulence of the influenza virus was unique, unprecedented, and never observed afterward [191], but many other causes may have contributed to the extraordinary severity of the pandemics in wartime, including overcrowding, undernutrition, and stress due to the war, thus making the disease spread and the bacterial super-infection with consequent pneumonia easier. The high case-fatality rate, in general, and for young adults in particular, remains without an answer, despite several, careful studies [191]. In 1918, two months before the armistice of November, a peculiar event occurred that will never be repeated: the simultaneous outbreak of influenza and malaria in the Egyptian Expeditionary Force in Palestine, in which out of 315,000 soldiers, 773 died from malaria and 934 from influenza–pneumonia. Disease victims outnumbered those due to combat by over 37 to 1. Moreover, out of 40,000 men of the Desert Mounted Corps, 19,652 sick soldiers were evacuated due to malaria from *Plasmodium falciparum*, a condition that caused the interruption of combat operations [192]. However, the US military tolerated a high influenza pandemic burden in 1918–1919, such that their engagement in studying and preventing influenza was witnessed by establishing, in 1941, the Board for the Investigation and Control of Influenza and Other Epidemic Diseases in the Army, which evolved into the Army Epidemiological Board in 1944 and the Armed Forces Epidemiological Board (AFEB) in 1949 [125]. This structure supported the studies for the development of the influenza vaccine [193,194], which was tested on the military. Starting in 1943, army personnel were immunized against virus A, prior to the licensure to Parke Davis, in order to prevent possible influenza outbreaks during troop mobilization [125]. Moreover, AFEB supported real-world studies of vaccine effectiveness in the military [195,196,197,198,199,200]. Influenza virus is highly mutant, and the immunization success is closely dependent on the matching between the circulating and the vaccine viral strains; thus, the WHO has organized a network of collaborating laboratories, in order to early identify the circulating strain and give precise indications to the industry for the seasonal vaccine preparation [201]. The US military participates in such a network with the Armed Forces Health Surveillance Center, Division of Global Emerging Infections Surveillance and Response System (AFHSC-GEIS), which supports at least 52 national influenza centers and other country-specific influenza, regional and US-based, emerging infectious disease reference laboratories (44 civilian, 8 military) in 46 countries worldwide for surveillance and response [202]. Even the French military has implemented a surveillance system for influenza, called the military influenza surveillance system (MISS), as further evidence of the relevance of influenza to the military [203]. Finally, even the Italian Armed Forces have organized an Influenza Surveillance System in coordination with the civilian Influenza Surveillance Network (Influnet), driven by the Italian National Institute of Health. All these activities aim to contrast a fearsome infectious disease, which, even though did not recur with the high virulence of the Spanish flu pandemic, has shown an easy capability of spreading in favorable environmental conditions, such as those encountered in the military [201]. However, although influenza is considered a threat to the military, flu vaccination was only compulsory in the US military, on the basis of a WHO survey [7,201]. More recently, influenza vaccination has become present in the military vaccination program of 24/25 NATO countries that report the vaccination program for the military; however, in only nine countries for the whole military personnel, two of these nine countries uniquely recommend [36]. The relatively scarce use of immunization for influenza in the military is probably a consequence of the relatively poor effectiveness of the influenza vaccine in young adults [204], which is parallel to vaccine immunogenicity [140].

### 2.16. Adenovirus

Adenoviruses are a group of over 50 serotypes of a DNA virus, which may be transmitted by respiratory route, conjunctiva (in case of contact with contaminated hands), and fecal–oral route. They may induce acute respiratory disease, conjunctivitis, and gastrointestinal infections. Premises for epidemics are environmental conditions characterized by community life with overcrowding, a situation often encountered in the military, particularly the recruits, who are exposed especially in the first 3–5 weeks of training [205]. A new virus, later denominated adenovirus [206], was identified in the first half of the 1950s by Dr. Hilleman and Dr. Werner at WRAIR [207]. It was later recognized that adenovirus includes different serotypes and that types 4 and 7 were particularly implicated in acute respiratory disease in the military [124]. Adenoviruses were later recognized as the main etiological agent of acute respiratory disease in the military, with up to 80% of infected and 20% of hospitalized subjects [208]. Dr Hilleman developed a formalin-inactivated bivalent vaccine including serotypes 4 and 7, which was successfully tested for safety and efficacy, showing to be safe and over 90% effective, and was licensed in 1958. However, due to the risk of contamination by the oncogenic virus SV40, the license was retired in 1963 [29]. New live oral vaccines for serotypes 4 and 7 were developed in the 1960s by a group of military researchers led by Col. Edward Buescher and were tested in the military [29]. These vaccines proved to be safe, highly immunogenic, and protective [209,210] and were regularly administered to the US recruits on the first day of their arrival at the training camps starting in 1971 [29]. However, in 1996, this vaccination was interrupted, as the vaccines were not produced anymore by the unique manufacturer; thus, the US Department of Defense made a contract with another manufacturer [125], and in 2011, vaccination of the military was resumed [211], with a dramatic decline of febrile respiratory illness and of adenovirus respiratory infections, which decreased 100-fold [212]. This vaccine is licensed by the FDA for US military personnel, ages 17 through 50, who may be at higher risk for infection from these two adenovirus types [36]. Although the issue of adenovirus respiratory infection has been deeply studied by the US military, it has been reported in the military of other countries since the 1970s until now [213,214,215,216,217,218]. However, among the 25 NATO countries reporting vaccination schedules for the military, only one country reports that adenovirus vaccination is recommended for all recruits [36]. This is probably due to the adenovirus epidemiology in these countries, frequently involving serotypes for which vaccine is not available. Moreover, the relevance itself of the problem may be overlooked by the lack of pathognomonic symptomatology and the difficult access to molecular and/or serological diagnosis.

### 2.17. Coronavirus Disease 2019 (COVID-19)

Coronavirus disease 2019 (COVID-19) is a potentially lethal respiratory disease, first described in China at the end of 2019 and still ongoing, caused by an RNA coronavirus (SARS-CoV-2, because similar to the SARS-CoV described in China in 2003), with high contagiousness, so that in a few weeks from the first description, it was declared a pandemic by the WHO [219,220]. As of 23 May 2022, it has caused 525,618,514 total cases and 6,277,339 total deaths, thus showing an average global attack rate of 6.78% and case-fatality rate of 1.19% (https://coronavirus.jhu.edu/map.html (accessed on 21 July 2022)). From the same data bank, the post-vaccine average annual new cases and deaths have been calculated. The average values of new cases and deaths referred to a 28-day period occurring in the last year (2021–2022) and were multiplied by 13 to refer to the length of one year; the results were 195,044,798 annual new cases and 650,702 annual deaths (Table 1). Compared with the dreadful Spanish flu of more than a century ago, the attack rate and the case-fatality rate of COVID-19 are markedly lower, considering that in the Spanish flu, the estimated attack rate was as high as approximately 30% [221], and the estimated lethality 50 million deaths [222]. Nonetheless, the current pandemic is representing a great challenge for all the countries and the respective health services, which are overwhelmed by the high number of patients who are hospitalized, particularly in intensive care units, for the more severe cases, during the acute phases of the pandemic. The response to the pandemic by research was unprecedented and could develop and make available in less than one-year effective vaccines [223], monoclonal antibodies, and anti-viral agents, even though the great variability of the RNA virus has generated viral variants of concern, more aggressive and/or more transmissible, which may make the disease control uncertain. In the research for an effective vaccine, the Chinese military had an early and relevant role [224,225]. Although the pandemic is still ongoing and has not been eradicated nor transformed into an endemic disease, the vaccine’s effectiveness, especially against severe disease and its complications, including hospitalizations and death, is definitively demonstrated [226].

The military are exposed to the infection not only for their community life but even for the direct management of the pandemic for its control, which offers a variety of opportunities for exposure to the virus [227]. However, even though the military are particularly exposed to the virus and their rate of infection may significantly differ or not from the civilian population, they are expected to overcome the disease without complications, considering that they are generally young and in good health [228]. A comparison between the study of the COVID-19 outbreak in the aircraft carrier Theodore Roosevelt and the cruise ship Diamond Princess is a clear demonstration of the statement above (Table 6). Theodore Roosevelt is a ship with a crew of 4779 members, 1271 of whom have been found to be serologically confirmed COVID-19 infected (26.6%) and 60 had suspected COVID-19 for suggestive symptomatology, in the absence of positive serology. Out of these 1331 (27.85%) confirmed and suspected COVID-19 infected subjects, 23 (1.73%) have been hospitalized, 4 (0.3%) needed intensive care, and one died [229]. Diamond Princess is a cruise ship that started a cruise on 20 January 2020 with approximately 3700 passengers and crew members, during which an outbreak of 712 COVID-19 infected subjects occurred (19.24%, *p* < 0.0000001 vs. Theodore Roosevelt), with 36 (5%, *p* = 0.00003448 vs. Theodore Roosevelt) hospitalized, and 13 (1.83%, *p* = 0.00001793 vs. Theodore Roosevelt) deaths [230]. The significant difference in the attack rates, higher in the military ship, is probably related to the tighter available spaces for sleeping in the military ship, compared to the more comfortable cabins of the Diamond Princess, where social distancing and isolation are easier to reach, whereas the higher rates of hospitalization and death in the Diamond Princess is probably related to the military being young and in good health. Last year, another outbreak occurred on another US Navy ship, with a crew of approximately 350 members. The infected crew members were 22 (attack rate 6.3%), all were fully vaccinated and, although symptomatic, no severe cases were observed, none were hospitalized and no death occurred [231]. This observation is a testimony of the effectiveness of the vaccine on hospitalizations and deaths and of the limited protection against infection, in the presence of the aggressive viral Delta variant. A similar observation has been made on vaccinated British military personnel deployed to Western Africa. A total of 15 out of 26 soldiers had symptomatic, but not severe, COVID-19 infection, despite being fully (11) or partially (4) vaccinated [232]. Even the infection-induced protection is not absolute, as demonstrated in Marine Corps recruits, who are admitted to the basic training after a quarantine period and a baseline negative quantitative polymerase chain reaction (qPCR) and a serological test for specific antibodies. The risk of infection in the seropositive recruits was five-fold lower than that of the seronegative recruits, thus underlining marked, but not absolute, infection-induced protection [233]. The relevance of the community life to the infection spread has even been clearly demonstrated in non-embarked personnel, such as Marine Corps recruits before being admitted to basic training. They had to follow a 2-week quarantine period at home followed by 2 weeks on a college campus, during which the recruits were asked to wear masks and to adopt social distancing. At the end of this second 2-week period, approximately 2% of recruits were SARS-CoV-2 positive by qPCR, thus underlining the relevance of community life for the infection rate, despite the right and checked behavioral control measures [234]. Even in the Bolivian military, the rate of infection is higher than in the civilian population (2.5% vs. 1.26%, *p* < 0.0000001), whereas the rate of mortality is significantly lower (1.9% vs. 6.19%, *p* < 0.0000001) [235]. The rate of infection even in the Brazilian military is higher than in the civilian population [236], whereas the opposite is observed in the Korean military [237], thus confirming that the rate of infection may depend on many variables, including the coverage of the vaccination in the military compared with the general population. Moreover, despite the vaccine’s effectiveness against severe disease, the protection against infection seems to be quite limited, in particular for some types of viral variants of concern; thus, the research is actively engaged in developing more effective vaccines, possibly a “universal” vaccine [238], such as the one that is desirable to obtain even for influenza [239]. However, despite that no documents are yet available on the vaccination coverage of the military in all the countries of the world, it may be hypothesized that in all countries, the military have been considered a category to be primarily vaccinated, such as health care workers and vulnerable patients. The COVID-19 pandemic has the characteristic of profoundly interfering with societal functioning and stability, even for the relevant sequelae of the acute disease (so-called long COVID-19) that may be observed in over one-third of the subjects [240] and may markedly reduce fitness to work [241], thus fully justifying the marked interest of the military for COVID-19 and their involvement in the management of the pandemic, in the picture of close civil–military collaboration in several world countries [242].

### 2.18. Pneumococcus

*Streptococcus pneumoniae* is a Gram-positive diplococcus, whose discovery was independently described in the same year, 1881, by the US Major George Sternberg [243,244] and Louis Pasteur [245,246]. *S. pneumoniae* is potentially fatal, being able to induce, in addition to otitis media, sinusitis, and bronchitis, invasive pneumococcal disease (IPD), including pneumonia, meningitis, febrile bacteremia, and death. More than 90 different serotypes are known, based on the antigenic characteristics of the polysaccharide capsule, which induces neutralizing antibodies. This makes the search for a fully protective vaccine difficult, considering that the polysaccharide vaccines, either plain or conjugated to a protein matrix, are only protective for the included serotypes, and a vaccine including all the serotypes is impossible to realize [247]. However, the search for alternative vaccines, based on the inactivated whole cell or purified proteins, has demonstrated that they are safe and immunogenic, at cellular and humoral levels [248], but less effective than expected; thus, the only approved vaccines are plain or conjugated polysaccharide ones, which have been demonstrated to be able to reduce the nasopharyngeal carriage, a necessary step for reducing IDP [247].

*S. pneumoniae* is the main etiological agent of community-acquired pneumonia, responsible for nearly a quarter of them [249]. The military are sensitive to the problem of pneumococcal pneumonia, considering that in WWI they had to observe the dreadful and quite invariably fatal pneumonia complicating measles and Spanish influenza. The US military, therefore, tested in 1945 the first hexavalent pneumococcal polysaccharide vaccine and observed a reduced incidence of pneumonia and pneumococcal carrier rates [125]. Despite this successful experience, the pneumococcal vaccine was scarcely used and later withdrawn from the market [250], due to the higher confidence placed at that time in the newly available antibiotics compared to vaccines to deal with the pneumococcal disease issue [251]. More recently, the US military organized a large randomized, double-blind, placebo-controlled effectiveness study of the pneumococcal polysaccharide 23-valent vaccine for reducing pneumonia in healthy military trainees. However, the results of this large and well-performed study on more than 150,000 recruits did not show any protective effect of the polysaccharide vaccine, whose routine use in healthy military trainees was, therefore, not recommended [252]. Currently, only 8/25 NATO countries report that the respective vaccination programs for the military included the pneumococcal vaccine; in one country, it is intended for all the military personnel, whereas among the other seven, it is only recommended in four, and only considered for selected categories in the remaining three [36]. This confirms that the issue of pneumonia prevention is far from being completely resolved with the currently available vaccines.

### 2.19. Rabies

Rabies is an almost invariably fatal disease, caused by an RNA virus, which is largely present in many feral mammal animals, including dogs, cats, skunks, raccoons, and bats, and it is transmitted to humans by bites, scratches, and contact with skin lesions or mucosae. The virus, once transmitted, retrogradely proceeds along the peripheral nerves toward the medulla and the brain, where, after an incubation time generally ranging from 20 to 90 days, it induces encephalomyelitis, which manifests with severe symptomatology, characterized by difficulty in swallowing and hydrophobia, and either an encephalitic (furious) or paralytic (dumb) form, in 80% and 20% of cases, respectively [253]. The WHO estimates that globally there are at least 55,000 deaths each year from rabies, especially in Asia and Africa [254]. No effective therapy exists, but an effective inactivated vaccine and passive immunotherapy with human rabies immunoglobulins (RIGs) are available. In case of bite and suspected infection, post-exposure prophylaxis (PEP) may be administered as soon as possible, by cleansing the wound and inoculating human RIGs at 20 IU/kg around the wound [255], and by administering four vaccine doses intramuscularly in two weeks (0, 3, 7, and 14 days) for minor contacts [256].

Despite that the military are not actively engaged in rabies research, rabies is a disease of military interest, in particular for deployed active service members [257]. In the US Armed Forces in the period 2011–2018, 22,709 animal bites were reported, which is an average of eight animal bites per day [258]. Animal bites with consequent rabies have been observed during the Vietnam war [259]. After the Vietnam war, rabies was still a problem in the Philippines, where in 1984, 315 potential rabies exposures were managed and 79 of them received PEP [260]. The British Army had to manage 62 animal bites when deployed to Bosnia–Herzegovina in 1995–1996 [257]. The possible shortage of RIGs may heavily influence the outcome of an at-risk animal bite in deployed personnel; thus, pre-exposure prophylaxis by active immunization has been considered to avoid the need of administering RIGs in the PEP [256]. All 25 NATO countries reporting the military vaccinating program include rabies vaccine for selected military categories; however, in three countries, rabies vaccine is only recommended [36].

### 2.20. Yellow Fever

Yellow fever is a potentially lethal disease caused by an RNA flavivirus, which is transmitted by the bite of infected mosquitoes of the species *Aedes aegypti* and *Hemagogus*, endemic in Sub-Saharan Africa and tropical Central and South America [91]. The case-fatality rate of the disease is estimated at approximately 35%; modeling studies have estimated in 2013 the burden of yellow fever in 84,000–170,000 cases with 29,000–60,000 deaths [16] (Table 1). The incubation period is 3–6 days, and the disease may run asymptomatic or with a mild, not specific, symptomatology, with fever, myalgia, backache, headache, loss of appetite, nausea or vomiting, for 3–4 days. Most patients heal from the infection, whereas a few patients may enter a toxic phase one day after the end of symptomatology, with multi-organ failure, icterus and bleeding from the nose, mouth, eyes or stomach; half of these patients die within 7–10 days [16].

Yellow fever was endemic in Africa, and in the sixteenth century, it traveled to the Americas, following the slave trade, thus becoming endemic in the coastal areas of Central and South America and even in the southern and eastern coast of North America to Boston. From 1668 to 1893, over 135 epidemics of yellow fever occurred in the USA. In 1793, an epidemic of yellow fever killed 10% of the Philadelphia population and in 1878, another epidemic killed 20,000 people in the Mississippi valley [261]. At that time, nothing was known about the biology of the disease and the way it is transmitted. At the start of the Spanish–American war in 1898, the US troops were decimated by yellow fever in Cuba. Thus, the Surgeon General of the US Army, Gen. George Miller Sternberg, organized a Yellow Fever Commission, coordinated by Major Walter Reed, and composed of Majors James Carroll, Aristides Agramonte, and Jesse Lazear, with the duty of clarifying the way of transmission of the disease in order to prevent infection spreading [261]. The Commission went to Cuba to begin its activity in June 1900. It started by verifying the etiological hypothesis proposed by the Italian microbiologist Giuseppe Sanarelli, who in 1897, announced to have found the etiological agent of yellow fever, which was named *Bacillus icteroides*. The commission ruled out this hypothesis and then focused its activity on taking into account the work of Carlos Finlay, a Cuban physician who had suggested a transmission through the mosquito *Aedes aegypti* by performing specific experiments on human volunteers, which were unsuccessful. Finlay tried to expose healthy volunteers to the bite of mosquitoes 2–6 days after the mosquitoes had bitten a patient with yellow fever; however, he never succeeded in observing a clear case of infection transmission. The reason was clarified over 10 years later by the observations of the US physician Henry Rose Carter in 1898, relative to the “extrinsic incubation” of yellow fever in the mosquito, which was calculated in approximately 2 weeks. The Yellow Fever Commission thus repeated Finlay’s experiments, by taking into account the “extrinsic incubation” time of Carter and succeeded in demonstrating the transmissibility of the etiological agent by mosquitoes, thus providing scientific evidence to Finlay’s hypothesis. Considering that there is not an animal model for yellow fever, the commission used healthy human volunteers, including the same members of the commission and one of its members, Jesse Lazear, who died in 1900, at the age of 34 years [262]. The observations of the Yellow Fever Commission were published in 1901 (JAMA 1901;36: 431–40), and the Major physician US Army William C Gorgas, responsible for health in Cuba, received the disposition to free Havana of mosquitoes. His work was excellent because in 90 days, he transformed the epidemiological situation of Havana, in which one case of yellow fever per day was described on average from 1762 to 1901, whereas after mosquito disinfestation, it was free of the disease [261]. Thus, the fight against yellow fever was won in this phase by the US military.

The viral etiological agent was only isolated in 1927 from a sick man in Ghana. A live vaccine, attenuated by 200 subcultures of this virus, designated 17D strain, was developed in the 1930s by Theiler and Smith [263]; Theiler was awarded the Nobel Prize for Physiology or Medicine in 1951 [92]. The vaccine is generally safe, highly immunogenic and protective for long periods, considering that the presence of neutralizing antibodies has been found after 30–35 years from vaccination [264]. Currently, all 25 NATO countries reporting the respective military vaccination program include yellow fever vaccine, in 24 for selected categories, whereas in one country for all the military personnel. With the use of the vaccine, yellow fever does not represent a problem for the military at the global level anymore [36]. Yellow fever virus has been included among the possible biological agents, category C [57].

### 2.21. Japanese Encephalitis (JE)

Japanese encephalitis (JE) is a potentially lethal disease caused by an RNA flavivirus transmitted by the bite of infected *Culex* mosquitoes, in particular *Culex tritaeniorhynchus*; however, even other mosquito species may be vectors. The virus is endemic in large parts of South, South-East Asia and the Western Pacific, including an estimated population of over 3 billion people, particularly in rural areas, where the risk factor is living in proximity of rice fields and pig rearing [265]. It is estimated that the annual JE cases are 67,900, with 13,600–20,400 deaths (Table 1). The infection may run asymptomatic in most patients; in one case out of 250 infections, the disease is severe. After an incubation period of 4–14 days, symptomatology starts with high fever, chills, myalgias, headache, and mental confusion; however, opisthotonos and even acute flaccid paralysis may occur. The disease occurs preferentially in children <10 years, in whom it is generally more severe. The case-fatality rate of the severe disease is 20–30%, and approximately 30% of the survivors present permanent neurologic or psychologic disabilities [17].

The interest for the US military started during WWII; in 1942, a research team was established at WRAIR, with the duty of developing a vaccine for JE [257]. Even Major Albert Sabin received by the Commission on Neurotropic Virus Diseases of the Army Epidemiological Board the task to develop a JE vaccine [125]. The vaccine was a formalin-inactivated JE virus cultured in the brains of mice; it was used on 250,000 US soldiers during the war, starting in 1945, after an outbreak of JE in the US military stationed in Okinawa [125]. Albert Sabin had the opportunity to study and describe this outbreak and the use of the vaccine [266]. Even after the war, Albert Sabin and the US military collaborated with Japanese researchers for studying together the JE vaccine [267]. The US military suffered a relevant outbreak of 300 cases of JE in 1950 among the US troops stationed in Korea during the Korean War, although the US military were all vaccinated, and 16 lethal cases were observed [257]. Even during the Vietnam War, cases of JE in the US Air Force personnel were described [257], however, no reduction of US force fighting strength was observed [268]. Following the Korea outbreak, which had demonstrated the poor protection provided by the first used vaccine, the army interrupted the vaccination of the US military assigned to the Far East Command [125]. At the end of the 1950s, researchers of WRAIR working in Japan contributed to providing new knowledge on JE ecology [269]. The studies for the development of a new vaccine resumed in the 1980s, led by the CDC; however, the conclusive phase III studies were carried out in Thailand, under the leadership of Col. Charles Hoke, of the US Army Medical Component, Armed Forces Research Institute of Medical Sciences (AFRIMS) in Bangkok, Thailand, during which a monovalent (Nakayama strain) and bivalent (Nakayama and Beijing-1 strains) vaccine were studied in comparison with placebo. The results showed 91% of efficacy for both monovalent and bivalent vaccines [270,271]. This study could be carried out due to the previous research at AFRIMS of the military researcher Donald Scott Burke, who had set up a diagnostic test for anti-JE IgM in serum and liquor [272,273]. Another study was carried out by the WRAIR researchers on 538 US soldiers with monovalent JE vaccine, which confirmed the safety and the high immunogenicity of the vaccine and ruled out the possible interference with a previous yellow fever vaccination, another flavivirus [274]. In 2005, the production of the mouse brain-derived JE inactivated vaccine was discontinued by the manufacturing company because it was considered too reactogenic and poorly immunogenic [275]. Currently, in the USA, the only approved vaccine is IXIARO (JE-VC), which is a Vero cell-culture-derived inactivated vaccine [276]. However, even live and recombinant live vaccines are available [17,274]. Twenty-one out of the twenty-five NATO countries report that the military vaccination program includes the JE vaccine for selected categories of personnel [36]. The Italian military soldiers participating in the INTERFET (International Force to East Timor) mission in 1999 were vaccinated with the monovalent (Nakayama strain) mouse brain-derived JE vaccine, without side effects. However, some of them were infected by the dengue virus, and the previous vaccination with the JE vaccine has been considered partly cross-protective even for dengue [277].

### 2.22. Tick-Borne Encephalitis (TBE)

Tick-borne encephalitis (TBE) is a disease caused by an RNA flavivirus transmitted by the bite of ticks. The virus is present in many animals, such as wild rodents, deer, boar, dog, fox, sheep, cattle, and bat, and is transmitted to humans by ticks of the family *Ixodidae*, in particular *Ixodes Ricinus* and *Ixodes persulcatus*. Humans are a dead-end host and may even be infected by alimentary route, by eating raw contaminated dairy. Three subtypes of the virus are responsible for the respective diseases that are endemic in central, eastern and northern Europe (western subtype), eastern Europe, Russia and northern Asia (Siberian subtype), and eastern Russia as well as some parts of China and Japan (far eastern subtype) [278]. Although two-thirds of the cases are asymptomatic, the disease caused by the western subtype has a biphasic pattern, with a first phase characterized by nonspecific symptomatology (fever, myalgia, headache, fatigue, nausea) and a second phase, following a free interval, by meningoencephalitis, myelitis or paralysis, whereas the far eastern subtype is associated with a monophasic disease. Moreover, the European disease is milder (mortality 0.5–2%, and neurological sequelae up to 10%) than the disease caused by the far eastern subtype, which has a mortality of up to 20% and a higher prevalence of permanent neurological sequelae [279]. The diagnosis may be made by molecular or serological approaches. However, virus identification by molecular methods is poorly used because the viremia is present for a short time, and the reliability of serological tests is reduced for the possible cross-reaction among different flaviviruses. There is no available treatment, whereas an inactivated vaccine from a cell-cultured virus has been shown to be safe and protective in over 95% of recipients after three-dose administration [278]. The annual world cases of TBE are approximately 10,000–12,000 [280], whereas in Europe, over 3000 annual TBE cases are hospitalized [281]. The disease tends to be more frequent in males than in females in Europe and more severe in >50–60-year-old subjects, who are less responsive to the vaccine [278,282,283].

The interest for the military of TBE is linked to the country where the military live or are deployed to, whether TBE is endemic or not. The US military, which started to be interested in TBE in the mid-1980s [284], vaccinated the troops deployed to Bosnia in 1996 with an accelerated schedule (0, 7, and 28 days instead of 0, 1–3, and 9–12 months) of TBE vaccine in order to be readily protected; 80% of seroconversion rate was observed after the third vaccine dose, and the vaccine proved to be safe, with only 7/3981 (0.18%) vaccinees reporting self-limited symptoms. However, the infection risk was relatively low, considering that only 4/959 (0.42%) unvaccinated soldiers seroconverted [285]. Among the 25 NATO countries reporting the respective vaccination schedule for the military, twenty-two include the TBE vaccine, six of these countries (all European where TBE is endemic) provided to the whole military personnel, and in the other sixteen, only to selected categories [36]. The TBE virus has been included among biological agents, category C [57].

### 2.23. Human Papillomavirus (HPV)

Human papillomavirus (HPV) is a DNA virus, which infects epithelial basal cells, at cutaneous and mucosal levels, and may induce different cutaneous and mucosal lesions and even cancers. There are more than 100 serotypes, some of which may cause cervix, anal, penile, and oropharynx cancers, with serotypes 16 and 18 being the most frequently implicated in cancers. However, even serotypes 31, 33, 45, 52, and 58 may be considered high risk for cancer induction, whereas serotypes 6 and 11 are generally associated with anogenital warts, such as condyloma acuminatum, and are considered low-risk HPV [286]. The main way of HPV transmission is through sexual intercourse; thus, the military worldwide are at special risk [287], hence their interest in HPV, even considering that a safe and effective HPV vaccine is currently available. Three recombinant vaccines are available, the bivalent (16 and 18 serotypes), the tetravalent (6, 11, 16, and 18 serotypes), and the nine-valent (6, 11, 16, 18, 31, 33, 45, 52, and 58 serotypes). It may be calculated that with bivalent and tetravalent vaccines, the protection against cancer is approximately 70%, whereas it increases to approximately 90% with the nine-valent one [92]. Only one NATO country reports HPV mandatory vaccination in the military schedule, whereas in the other six countries, the vaccination is only recommended [36]. A longitudinal study in the US military showed that 14.6% of male recruits were HPV positive for serotypes 6, 11, 16 or 18 at entry and 34.2% of those originally negative for these serotypes seroconverted to one or more of them after 10 years [288]. However, more recently, an epidemiological survey on genital HPV infections developed during a 9-year long follow-up, between 1 January 2012 and 31 December 2020, has shown a significant reduction of infection for both genders, female service members from 261.2/10,000 to 163.1/10,000 person-years (37.5% of reduction) and male service members from 40.6 to 16.9/10,000 person-years (66.1% of reduction), a decrease that has been attributed, at least in part, to the introduction of a vaccine for females in 2006 and for males in 2010, which, even though it is not mandatory, is encouraged and offered to service members [289]. In a relatively low number of countries (76/195, 39%), the HPV vaccine has been introduced as mandatory and in most cases for young females, with a gradient of application ranging from 10% for low-income countries to 69% for high-income countries, thus clearly indicating the negative influence of poverty on the possibility of introducing this relatively expensive vaccine. The global HPV vaccination coverage is estimated to be as low as 12.2% [290]. Australia was the first country to organize, in 2007, an eradication program of the cancer of the cervix [291]. Despite that the fight against cervical cancer is a priority considering the high number of annual cases and deaths, especially in low-income countries [290], and for this reason, the vaccination campaign has mainly been addressed to young girls before starting sexual activity, the vaccination of the males should also be considered to prevent male cancers [292].

HPV vaccination in the military could contribute to the reduction of HPV-related cases of cancers if mandatory, considering that the military are at a higher risk of infection than the matched civilian population, and the simple recommendation of vaccination cannot reach critical coverage, considering the stigma linked to the sexually transmitted diseases [293,294]. Moreover, a cost-effectiveness estimate allows one to compare the care cost per case of anal cancer of USD 52,700 or 146,100 per case of oropharyngeal cancer versus USD 450 for HPV vaccination [295]. Thus, this hesitancy in making HPV vaccination mandatory for the military is quite surprising, and it diverges from the historical behavior of the military, that for many infectious diseases has generally anticipated the general population in vaccine research and application. Probably, this was the expression of a different time, in which infectious diseases could heavily influence the outcome of battles and war more than the combat capacity. Moreover, HPV is not acutely incapacitating, considering that it may induce deferred neoplastic disease. However, a larger vaccine use, especially in countries with compulsory conscription, may represent a relevant measure of public health.

### 2.24. Cholera

Cholera is a bacterial disease that can be transmitted through water or food contaminated with *Vibrio cholerae*, 01 and 0139 serogroups, endemic in 50 countries and able to induce epidemics. It is estimated that annually 1.3–4 million people become infected, resulting in 21,000–143,000 annual deaths [296] (Table 1). Seven pandemics since 1817 spread from Asia to all over the world. Right rehydration may lower the mortality from over 50% to 0.2% [21]. The prevention consists of water sterilization and sanitation. Cholera was first reported by the British military in 1770 [297]. Similar to all the infectious diarrheal syndromes linked to poor hygienic conditions, it has always been considered a threat by the military. In 1855, during the Crimean War, the Piedmont–Sardinia expeditionary force was deeply hit by cholera; 2728/18,000 military personnel fell ill with cholera, an attack rate of 15%, and 1230 died, a case-fatality rate of 45% [298]. A live vaccine against cholera was first developed by Jaime Ferran in Spain [299], but it was ultimately the vaccine developed by the German scientist Wilhelm Kolle in 1896, using heat-inactivated cholera bacilli, that came into general use and that served as a model for cholera vaccines for the next century [300]. As a military physician and hygienist during WWI, Kolle was highly successful in vaccination against cholera. This vaccine was widely used during WWI in the military, such as by the Italian Army when, in August 1915, cholera broke out in the Italian troops deployed along the Isonzo river. The anti-cholera mass vaccination of the military was then ordered and subsequently extended to civilians residing in closely affected areas. This approach allowed the containment of the epidemic, which remained almost confined to the military community and only marginally affected the civilian population. In 1915, the observed cases were 14,000, whereas they were reduced to only 170 in 1916 [301]. These data demonstrate that vaccination campaigns can be carried out safely even during the epidemic phases, helping to provide useful information to the scientific world to better understand the effectiveness of this vaccine. However, this vaccine was painful and did not give long-lasting immunity.

Furthermore, it is worth noting the contributions of US military investigators on the front lines of cholera research. The US Navy’s involvement with cholera began in Cairo, Egypt, during the 1947 cholera epidemic, when the commander of the Naval Medical Research Unit (NAMRU) 3, Robert A. Phillips, made some interesting observations. He established that the stools of patients with cholera were isotonic with their blood [302] and did not contain proteins; thus, allowing him to argue that no mucosal damage was present [297]. This observation allowed the rehydration of patients by infusion of isotonic electrolyte solutions to be possible, as even confirmed in a cholera outbreak in Bangkok Thailand, where Phillips applied his method [303]. This rehydration method allowed the mortality of cholera to be reduced from 20–30% to less than 1%; thus, saving a large number of lives. Later, in 1961 in Manila, Phillips discovered that isotonic electrolyte solutions containing glucose could be orally administered to rehydrate patients with cholera and other diarrheal diseases. This further observation made the rehydration method accessible even to developing countries for its lower cost, thus allowing millions of lives to be saved in the past several decades [297]. Richard Finkelstein, a civilian working at WRAIR, isolated the cholera exotoxin, which he called cholerogen, in 1963 [304].

Finally, the US military contributed to developing and testing improved cholera vaccines. Col. Jose Sanchez and colleagues, from WRAIR [305], and even in collaboration with the US Navy Medical Research Institute Detachment—Lima, Peru [306], studied a killed, whole-cell, vaccine plus recombinant cholera toxin B subunit (WC/rBS), and Col. David Taylor and other colleagues from WRAIR contributed to basic science research into a live attenuated O139 *Vibrio cholerae* vaccine prototype [307,308].

In addition, the US Department of Defense contributed to basic science research into a live attenuated cholera vaccine at the Armed Forces Research Institute of Medical Sciences in Bangkok [309] and at the Indonesian US NAMRU in Jakarta [310], respectively.

While considering the advances in the development of vaccines, also due to the contribution of the military, cholera is still a major global health problem in unsanitary conditions. Current cholera vaccines, represented by a two-dose killed whole cell monovalent (01) plus recombinant cholera B subunit of cholera toxin (WC-rBS), a two-dose killed whole cell bivalent (01 and 0139) (WC), and a single-dose live oral attenuated vaccine (CVD-103 HgR), are safe, feasible to use and represent a public health tool in the prevention of the disease, along with hygiene measures [311]. Currently, such as for typhoid fever, in the military of most NATO countries (21/25), including Italy, cholera vaccination is present in the vaccination schedule, but only for the troops deployed to at-risk epidemiological countries [36]. *V. cholerae* has been included among the possible biological agents, category B [57].

### 2.25. Leptospirosis

Leptospirosis is a potentially fatal bacterial disease caused by *Leptospira*, an aerobic bacterium containing in its structure a lipopolysaccharide similar to the one found in Gram-negative bacteria [312]. *Leptospira* is present in different wild and domestic animals; however, the main reservoir for human infections is *Rattus norvegicus* [313]. *Leptospira* is excreted in rat urine; thus, contaminating soil and water. Humans are accidental hosts, who may be infected through the trans-cutaneous or trans-mucosal passage, profiting from cuts or abrasions of the skin or conjunctival and/or oral mucosae [314]. Leptospirosis is therefore an occupational zoonosis; the most exposed worker categories are sewage workers, farmers in rainy areas and the military, particularly during exercises in marshy soils. The disease may be mild and self-limited; however, in some subjects and with some serovars, the disease may be severe, as in the case of Weil’s disease, caused by serovars of the *icterohaemorrhagiae* serogroup, in which the mortality is over 10%, or the severe pulmonary hemorrhage syndrome, which may have a case-fatality rate of over 50% [314]. Annually leptospirosis is estimated to be responsible for 1.03 million clinical cases with 58,900 deaths [25] (Table 1).

The etiological agent was discovered in 1915 by Japanese [315] and German [316,317] physicians, whereas the severe form of the disease had been described by Weil in 1886 [318]. However, the disease was present before and in the seventeenth century in New England, and in the eighteenth and nineteenth centuries in Europe, illnesses with the characteristics of leptospirosis had been described, particularly by military doctors [319]. Leptospirosis is a disease of interest for the military, because it is frequently associated with the military, both in wartime, considering the precarious hygienic conditions, particularly in humid trench warfare, and in peacetime, for training in standing water [320]. It was described during the second independence war in Italy in 1859 [319], during WWI and WWII, and the Vietnam War [308]. In the summer of 1942, there was an outbreak of febrile exanthem at Fort Bragg, which involved 40 US soldiers and recurred in the summer of 1943 and 1944, whose etiology was only clarified in 1952 by Major US Army William Gochenour and colleagues following isolation of *Leptospira autumnalis* [321]. In the same year, Major Gochenour and colleagues were able to diagnose as leptospiral meningitis an outbreak of “aseptic meningitis” occurring in 1949 in US soldiers serving in Okinawa [322]. In the 1980s, Dr. Ernest T. Takafuji from WRAIR and colleagues were able to successfully test the efficacy of chemoprophylaxis with doxycycline against leptospirosis on US soldiers training in field exercises in the Panama Canal [323]. Although an inactivated whole-cell vaccine has been available for more than a century, it is largely used in animals, whereas it is rarely used in humans, despite its effectiveness, due to its specific protection only against single serovars (Spirolept^®^, produced by Sanofi-Pasteur, only protects against *Leptospira icterohaemorrhagiae*), the quite heavy schedule, characterized by three subcutaneously administered doses followed by biannual boosters, and its reactogenicity [313,324]. Although information about the number of world countries adopting the leptospirosis vaccine for the military is lacking, among NATO countries, only two consider leptospirosis vaccination in occupationally exposed military personnel [36].

### 2.26. Dengue

Dengue is the most prevalent arthropod-borne viral disease [325], responsible for an estimated 390,000,000 annual infections, a quarter of which are symptomatic [18] (Table 1). The virus is an RNA flavivirus, of which four serotypes (1, 2, 3, and 4) are known, and is transmitted by the same vector of yellow fever virus, *Aedes aegypti*; however, in some geographical regions, other vectors, such as *Aedes albopictus* and *Aedes polynesiensis*, may even transmit the virus [326]. The infection may run completely asymptomatic, whereas in an estimated 25% of cases, it may induce non-specific fever, dengue fever, dengue hemorrhagic fever, and dengue shock syndrome. Dengue hemorrhagic fever is frequently observed in children and dengue shock syndrome, if severe, may be responsible for death in 9.3%, but up to 47%, of cases with profound shock [327]. Dengue is of interest to the military because it is highly prevalent at the global level and may heavily reduce the operational readiness of the soldiers, even though the annual mortality is estimated to be quite low, 12,000, mainly occurring among children [19] (Table 1). In a recent quantitative algorithm to quantify the burden of infectious diseases for the US military, dengue ranks third, after malaria and bacterial diarrhea [328]. Moreover, the prevention of dengue consists of the defense from the vector, considering that a satisfying vaccine registered in many world countries is still lacking. Although different types of vaccines are under study, including the live recombinant ones, inactivated, subunits with the envelope (E) protein alone or together with the precursor of the membrane (prM), only one tetravalent recombinant live on a YF17D backbone has been licensed in Mexico in December 2015, with the name Dengvaxia^®^ [329]. Afterward, other endemic countries registered this product with their respective regulatory authorities. This vaccine is administered according to a 0/6/12-month schedule and has the highest efficacy of 76.9% against serotype 4 and the lowest of 43% against serotype 2. However, the cumulative efficacy was substantially higher, 78.2%, in people already exposed to dengue compared to naïve [329]. In different projects for vaccine and monoclonal antibodies development, the US military, at WRAIR and Naval Medical Research Center (NMRC), are involved as further evidence of the interest in dengue for the military [329,330,331]. In addition to the diagnostic problem for the cross-reaction with other flaviviruses and the possible cross-protection between different flaviviruses [332,333], for dengue virus only the issue of antibody-dependent enhancement (ADE) has been described, which is the facilitated antibody-mediated viral entry into the cells through the FcγR [326]. ADE has been considered as the main reason for the waning, after approximately 2 years from an infection with a dengue serotype, of the cross-protection against the other three serotypes (heterotypic protection), whereas the homotypic protection is lifelong [329], in line with the long persistence of protective antibodies [334]. After the waning of heterotypic protection, people are more exposed to severe forms of dengue by heterologous serotypes [335]. This peculiar behavior of humoral anti-dengue immunity has to be taken into account when developing a dengue vaccine.

The US military have reviewed the burden of dengue from the American–Spanish War, through the Philippines, where they could observe that the disease more easily occurred in urban than in rural areas and that reinfection was not rare. During WWII, dengue occurred in many war theaters, particularly in the South Pacific, New Guinea and the Philippines; in the Vietnam War, the diagnosis moved from clinically to laboratory made, and finally in the Philippines again, Somalia, and Haiti [336]. During this long period, the engagement of the US military was continuous, mainly in the etiology and diagnosis, with a relevant contribution of the former Major Albert Sabin, and prevention through indirect measures, whereas the involvement in the research for an effective vaccine is more recent, probably for historical underestimation of the military significance of dengue [337,338]. Even the French military exert dengue surveillance for their overseas departments and territories endemic for dengue, where annually, 25,000 French soldiers are present, thus replacing the lack of a local epidemiological surveillance system [339]. However, in addition to the US, French and British military, who have a long historical tradition of being present at the global level in endemic areas, the military of all the world’s countries may be challenged with the dengue problem during peace-keeping operations in endemic areas [277]. Currently, US military researchers are still actively engaged in the search for a safer and more effective vaccine than Dengvaxia^®^, which has not been licensed by the FDA [339]. Dengvaxia^®^ may induce severe dengue in seronegative recipients of any age >9 years [340]. Moreover, the efficacy against serotypes 3 and 4 is good, whereas it is moderate to serotype 1 and marginal to serotype 2 [340]. In Table 7, the military relevance for and the military contribution to vaccine-preventable diseases is summarized.

## 3. Non-Vaccine-Preventable Infectious Diseases

### 3.1. Epidemic Typhus

Epidemic typhus is a bacterial disease historically associated with poverty, dirty environment, and overcrowding, all conditions that are found during war; thus, epidemic typhus has accompanied and sometimes has heavily influenced conflicts, mainly in the nineteenth and in the first half of the twentieth century. It is caused by *Rickettsia prowazekii*, a Gram-negative bacillus belonging to the order of *Rickettsiales*, which is transmitted to humans by the body louse (*Pediculus humanus corporis*), as discovered by Charles Nicolle in 1909 (who, for this discovery, was granted the Nobel Prize in Physiology or Medicine in 1928), at that time Director of the Pasteur Institute in Tunis [341]. The lice do not transmit *R. prowazekii* through the bite, rather they are infected after biting an infected human or animal; once infected, the lice eliminate a large number of microorganisms with feces as a powder, which remain viable for up to 100 days and may enter the body through skin abrasions. In addition, inhalation of aerosolized dry powder containing viable *Rickettsiae* is another effective way of infection; this type of infection may create some theoretical concerns for the possible use of *R. prowazekii* as a biological weapon, of category B [57,342]. In subjects who have suffered epidemic typhus even decades before, the disease may recur as a consequence of immunosuppression or wartime stress (recrudescent epidemic typhus or Brill–Zinsser disease) and spread in a lousy-naïve population, thus allowing this disease to be maintained by this human reservoir [341]. However, even an animal reservoir has been identified [343].

Historically, the first test for diagnosis was the Weil–Felix, based on the cross-reaction with *Proteus*; more recently, specific immunological and molecular methods have been set up for diagnosis [344]. Without therapy, typhus is a severe disease, characterized by easy spreading and relatively high mortality, ranging from 13% to 30% [344]. After an incubation time of 10–14 days, the disease suddenly bursts with non-specific symptoms, such as high fever, headache, myalgias, and rash of the trunk and limbs. Moreover, nausea and vomiting, pneumonia, petechial rash, central nervous system involvement with seizures, and mental confusion (hence the name “typhus”), may be present [345]. Antibiotic therapy with tetracycline or chloramphenicol promptly resolves. However, if the environmental situation allows, a bath and new clean dress and underwear are enough for eliminating lice; thus, the disease may easily spread in emergency and war conditions when these simple operations become impossible to be realized; it has been stated that epidemic typhus has caused more deaths than all the wars in history [344,346]. The trial of developing a vaccine has met a series of difficulties, including that *Rickettsia* is obligated to be intracellular, thus the need of growing it with cells. The pioneering study of Weigl, who prepared an inactivated vaccine from the homogenates of infected-lice intestines [347], which proved to be effective, did not solve the problem, because it was not fit for mass production [348]. However, the work of Cox, who demonstrated that *Rickettsiae* could effectively be grown on the yolk-sac membrane of a developing chick embryo [349], paved the way for the mass production of an effective vaccine [350]. The trial of developing a live vaccine was instead a failure [348].

Epidemic typhus has for a long time accompanied wars. In 1776, the war of the American Revolution might have been prolonged as a consequence of epidemic typhus in up to one-third of the American Army before one battle with the British Army [351]. During Napoleon’s Russia campaign, epidemic typhus heavily contributed to decimating the Grande Armée (Great Army), which was reduced 100-fold from 500,000 men to 5000 [1]. A detailed computation aiming at precisely calculating the deaths due to combat versus those due to disease attributes approximately 41,000 deaths to combat and 200,000 to infectious diseases, represented by typhus, dysentery, and diphtheria [352]. Whichever the real data, these figures underline that the invisible enemy was much more effective in decimating troops and influencing the outcome of the war than the visible one. Even during and after the WWI epidemic, typhus was a protagonist. In Serbia, which had approximately 3 million inhabitants at the start of the war, after 6 months, 500,000 people had epidemic typhus, and 40% of them, which is approximately 200,000 humans, died, including 70,000 Serbian troops and 30,000 Austrian prisoners [353]. At the worst point of the outbreak, 10,000 new cases per day were observed, and the mortality rate increased from 20% to 60–70% [353]. In the US troops, instead, the burden of typhus was limited, with only 47 cases and three deaths, probably due to the effective delousing organization [342]. Epidemic typhus was present and particularly active during the Russian Revolution in the period 1918–1922. It was estimated that in the period 1918–1920, there were 573,000 cases in the Red Army, with at least 100,000 deaths. The rate of infection was 204 per 1000 in 1919 and 315 per 1000 in 1920. No information is available for the White Army [354]. It was estimated that the total number of cases in the period 1918–1922 ranged from 15–16 to 25 million, whereas the number of deaths was estimated at approximately 2.5 million Russians [354]. WWI represents a sort of watershed for the relationships between infectious diseases and the war, considering that in all the previous wars, the ratio of infectious diseases/combat as the cause of death during the war was positive, with a marked advantage of infectious diseases in combat [338], whereas in WWI, this ratio was lower than 1. The British troops lost to disease were 113,000 versus 585,000 for combat and in the German Army 155,013 soldiers died from disease versus 1,531,048, who were killed in battle [355]. Regarding epidemic typhus, despite the hygienic conditions being similar on the Western and the Eastern Front, the disease was absent on the former and only present on the Eastern Front [355]. The first developments of bacteriology, which may have contributed to reducing the influence of infectious diseases in WWI [355], were operating even more in WWII, in which vaccination for epidemic typhus and the use of dichloro-diphenyl-trichloroethylene (DDT) for delousing, under the directive of the Joint US Typhus Commission, contributed to reducing the burden of epidemic typhus. The US troops had 104 cases and no deaths [342]. During the Korean War, 32,000 cases and 6000 deaths were observed in the South Korean military and civilian population, whereas a single case was observed in the US military [342]. No cases were observed during the Vietnam War, whereas since 1993, tens of thousands of cases of epidemic typhus were observed in different African and Latin American countries [342]. In particular, the civil war in Burundi was associated with an outbreak of epidemic typhus in the refugee camps [356]. However, in Russia, epidemic typhus was still present at the end of the last century [357]. Despite the interest in epidemic typhus for the military, even in consideration of its possible use as a biological weapon, no effective vaccine is currently available, considering that the Cox vaccine used in the US military during WWII did not comply with the modern recommendations on vaccine safety. The recent developments in the knowledge about protective immunity and immunological determinants for protection form the basis for developing innovative and effective vaccines [358].

### 3.2. Scrub Typhus

Scrub typhus is an acute febrile disease caused by a type of rickettsia, *Orientia tsutsugamushi*, transmitted by the bite of infected *Leptotrombidium spp*. *mites*. The disease is characterized by fever and maculopapular rash, with headache, lymphadenopathy, and frequent involvement of the central nervous system. Without therapy, the disease may have a mortality as high as 60% [342]. Antibiotic treatment with tetracycline and, more rarely, chloramphenicol is generally resolutive, even though recent antibiotic resistance has been observed. Scrub typhus is endemic in a large triangle including South and Southeast Asia and Australia. During WWII in the Pacific war theater, the Allied military had 16,000 cases of scrub typhus, 7300 of whom were in the US troops, with 331 deaths, whereas the Japanese had 20,000 cases [342]. During the Korean war, only eight cases were observed in the US military, whereas during the Vietnam War, it was estimated that 20–30% of the fever of unknown origin, once excluded malaria, was due to scrub typhus. The interest of the military in scrub typhus is witnessed by the studies of US military researchers on antibiotic therapy, a fieldable diagnostic test development, the first DNA sequence publication of the *Orientia*, the antibiotic resistance and the patented recombinant rickettsia protein (56 kd, useful for vaccine and diagnosis).

### 3.3. Trench Fever

Trench fever is a typical disease for the military, which was first described in 1915 by a British military physician, Major John Graham, who reported that a soldier had suffered from fever for three days, headache, dizziness, lumbago and shin pain. After a few days, the symptomatology appeared again and disappeared three days later, leaving fatigue. Other soldiers with the same clinical characteristics were observed by Major Graham, who described for the first time a disease closely associated with the military, highly incapacitating, even though no deaths were reported [359,360]. In 1917, the British War Office and the US Red Cross set up two independent commissions for trench fever, the first chaired by David Bruce, the commander of the Royal Army Medical College and discoverer of the etiologic agents of brucellosis and African trypanosomiasis, and the second by the US Major Richard P. Strong, who had directed the International Commission for limiting the epidemic typhus in Serbia in 1915. The two commissions ended their activity after a series of studies and experiments, in 1918, by setting some acute, relevant, and original points, regarding the incubation time, the symptomatology, the transmission by body lice, and the suggestion of a *Rickettsia* as a probable etiologic agent [361,362]. The rickettsia-like bodies were identified in the lice after biting a man with trench fever by a German bacteriologist in 1916 [363] and confirmed in 1918 by a British researcher [364]. There are no records on the number of cases of trench fever during WWI, but 800,000 soldiers on the Western Front may be estimated [360]. Only in 1961 was the etiologic agent of trench fever cultivated by J William Vinson of Harvard University and Henry S Fuller of WRAIR [365] and was first defined as *Rickettsia quintana* (after the fever duration), then *Rochalimaea* and lastly *Bartonella quintana* [360]. Through paleomicrobiological studies, it was established that *Bartonella quintana* has been found in human bodies over the span of 5 millennia [366]. It has been found in the teeth of Napoleon’s soldiers retreating from Russia, with a percentage of 20% for *B. quintana* and 8.6% for *R. prowazekii*, however, a comparison between the civilian and military population in the seventeenth and eighteenth centuries did not find any significant difference [366]. Currently, *B. quintana* has been found in refugee camps [356], homeless and HIV infection, with specific syndromes other than trench fever, such as endocarditis, chronic bacteremia and bacillary angiomatosis [367].

### 3.4. Leishmaniasis

Leishmaniasis is a protozoan parasitic disease transmitted by the bite of sandflies, which may manifest as cutaneous, mucocutaneous, and visceral leishmaniasis, the last invariably fatal by 2 years, if not treated. The first description of leishmaniasis is an almost exclusively military story, considering that in 1885, the British Major David Douglas Cunningham described parasites in the cutaneous lesions, followed in 1898 by the Russian Army physician Peter Fokitsch Borovsky, who made further descriptions. Moreover, in 1903, two British military physicians, Major William Boog Leishman and Captain Charles Donovan described the parasite in the spleen of a soldier who died of visceral leishmaniasis in India and in the spleen of a living soldier collected by puncture, respectively [368]. Neither Leishman nor Donovan identified the parasite as a new one, never described before. It was Major Ronald Ross, who had been a military physician in India and had left the military service in 1899 to return to England at the School of Tropical Diseases in Liverpool, to whom Donovan had sent some slides for an opinion, who rightly interpreted the parasite as a new one genus, proposing to denominate it *Leishmania donovani*, after Leishman and Donovan [369,370]. The disease is transmitted by the bite of some sandflies, generally the *Phlebotomus* in Asia, Africa, and Europe (old world) and *Lutzomyia* in the Americas (new world). The infestation may run asymptomatic; however, in some cases, the disease appears after an incubation period of weeks to months [368]. The different *Leishmania spp*. may induce different types of disease, with *Leishmania major* and *tropica* especially responsible for cutaneous leishmaniasis (the first of which has rodents as the reservoir and is a rural problem, the second has dogs as the reservoir and is an urban problem), whereas *Leishmania donovani* is mainly responsible for the more severe visceral leishmaniasis; finally, *Leishmania aethiopica* may induce cutaneous, diffuse cutaneous, disseminated cutaneous and mucocutaneous leishmaniasis, all in the old world. In the new world, the main species of *Leishmania* is *Leishmania braziliensis*, which, together with *Leishmania mexicana*, *amazonensis* and *guyanensis* are responsible for the induction of cutaneous, mucocutaneous, diffuse and disseminated cutaneous leishmaniasis in the new world, whereas *Leishmania infantum* is responsible for the induction of visceral and cutaneous leishmaniasis both, in the old world and new world [371]. The WHO estimates that 350 million people are at risk of leishmaniasis in 88 world countries; the annual number of cutaneous leishmaniasis cases is 1.0–1.5 million, and 500,000 is the number of annual cases of visceral leishmaniasis [372]. Ninety percent of cutaneous leishmaniasis cases occur in seven countries: Afghanistan, Algeria, Brazil, Iran, Peru, Saudi Arabia, and Syria, whereas 90% of visceral leishmaniasis cases occur in only five countries, Bangladesh, India, Nepal, Sudan, and Brazil [372]. The diagnosis may be made by identifying amastigotes, the form of the parasite in the spleen, bone marrow and lymph nodes, whereas in the gut of the sand-fly it appears as promastigote, which is injected into humans, and then transformed inside macrophages to amastigote. There is an effective therapy that is principally based on sodium stibogluconate (pentavalent antimonials), which may intralesionally be administered in cutaneous leishmaniasis, and intramuscularly or intravenously in visceral leishmaniasis; however, this therapy, similar to the other available ones, is not exempt from toxicity; thus, the need of a safer and effective therapy is felt [373]. No preventive vaccine for human use has been approved by the regulatory authority yet, while one inactivated *Leishmania amazonensis* human vaccine has been licensed for immunotherapy in Brazil and another live *Leishmania major* in Uzbekistan [374]. Considering that no preventive vaccine for human use, nor preventive chemotherapy, is currently available, prevention, in case of travel to at-risk areas, may only be carried out with individual protective measures, including long-sleeve permethrin-impregnated clothes, able to reduce 75% the occurrence of leishmaniasis during a 6-week period [375], a lotion of *N*,*N*-diethyl-m-toluamide (DEET) as a repellent, and the use of pyrethrin-treated bed nets [376], in addition to vector and reservoir control [368].

The interest for the military is linked to missions in at-risk areas, the type of activity, and the compliance with personal protective measures. In addition to the contribution of British military physicians in the first description of the parasite, the US military researchers provided relevant original contributions in the field of personal protective and vector control measures [377] and on the best conditions for the use of diagnostic and therapeutic tools [368]. The incidence rate of leishmaniasis among the US military was 7.2 cases per 100,000 person-years between 2001 and 2016, with the majority of cases being cutaneous leishmaniasis [378]. However, the retrospective analysis of leishmaniasis in the US military shows that the recorded cases during WWII were only 361. No cases have been reported in the Korean and Vietnam wars, whereas 19 cases of cutaneous and 12 of visceral leishmaniasis have been reported in the Operation Desert Storm in Iraq in 1990–1991. In 2003, during Operation Iraqi Freedom, a war in which the trend observed in WWI of lower influence of infectious diseases compared to the battle injuries on the number of deaths was inverted, considering that the aeromedical evacuations for disease and nonbattle injuries were six times more common than for battle injuries [379]; 0.23% of all the deployed US ground military had cutaneous leishmaniasis (*Leishmania major*) and 2.1% in a survey on 15,549 US military deployed to one or more operations. An undetermined number of cases had cutaneous leishmaniasis (*Leishmania tropica*, *Leishmania infantum-donovani*), and at least nine cases had visceral leishmaniasis (*Leishmania infantum-donovani*) [380]. The refugees are relevant vectors of disease, as demonstrated by the net increase in *Leishmania* cases in Lebanon from the period 2000–2012, when the annual number of cases ranged between 0 and 6, to 2013, when 1033 cases were reported, 998 (96.6%) of which in the Syrian refugees as a consequence of the Syrian war, and the remaining 3.4% due to Lebanese nationals and Palestinian refugees [381]. Even military training activity in at-risk areas is burdened by a high rate of leishmaniasis incidence, as demonstrated in the French military exercising in French Guyana [382], in the Peruvian military making training activity in the Amazon Basin, in which an incidence rate of cutaneous leishmaniasis of 25% was observed [383], in two cohorts of Dutch military troops exercising in Belize, in which an attack rate of 25.2% and 17.5%, respectively, was observed [384], in the British military following jungle training in Belize [385], and in the Colombian military [386]. Despite that the number of cases of leishmaniasis is relatively low, the vulnerability of the military to this sand-fly-borne disease makes the search for a safe and effective preventive vaccine for human use a high priority for the military.

### 3.5. Malaria

Malaria is a mosquito-borne protozoan parasitic disease that is highly widespread and potentially fatal. The parasite is *Plasmodium*, four species of which are responsible for almost all human cases, including *P. falciparum*, which is responsible for the most severe clinical form, *P. vivax* and *P. ovale*, which induce resting stages (hypnozoites) able to reactivate the disease many months or years after the initial event, and *P. malariae*, and is transmitted by the bite of a female *Anopheline* mosquito [387]. The complex cell cycle of the parasite is partly developed inside the mosquito and partly inside the human host, where the parasite enters the blood, and goes to the liver, in which *P. vivax* and *P. ovale* may stay inactive (hypnozoites) for a long time, and finally, it completes its cycle inside the erythrocytes, thus being able to induce severe anemias [387]. The estimated global cases in 2020 were 241 million cases, with 627,000 deaths, in prevalence <5-year-old children [15] (Table 1). The disease is endemic in a large part of Latin America, Africa, the Arabic Peninsula, and South-Southeast Asia and may manifest as periodic fevers; however, particularly in children and in the infections by *P. falciparum*, the clinical course may suddenly be complicated by cerebral malaria, generally due to parasite sequestration, and severe anemia, acidosis, and respiratory failure [387]. Even though febrile disease in a person living in or coming from an endemic malaria region should always induce suspicion for malaria, lack of pathognomonic symptomatology makes the laboratory diagnostic confirmation an absolute need. Traditional laboratory diagnosis is carried out by microscopy on thick and thin blood smears. Moreover, recently rapid diagnostic tests have even been set up, which allow for a quick, specific, and ultrasensitive diagnosis [388].

Malaria has traditionally represented a relevant threat for the military, hence their interest in malaria prevention and treatment. The therapy was initially represented by quinine, successfully administered in high doses to US soldiers with intermittent or remittent fevers in Florida in the period 1838–1842, during the second Seminole War, by the US chief medical officer of the deployed force Benjamin Franklin Harney [389]. During WWII, the shortage of quinine represented a worry for the US military; thus, a Malaria Drug Development Program was set up, from which new anti-malarial drugs, such as chloroquine, amodiaquine, primaquine, proguanil, and pyrimethamine were made available [389]. Under the pressure of the Korean War, in which many malaria cases were caused by *P. vivax*, insensitive to chloroquine in the stage of hypnozoite, the US Army promoted studies of the effectiveness of primaquine, which resulted in success [389]. Finally, during the Vietnam War, the growing observed malaria drug resistance pushed the Division of Experimental Therapeutics of WRAIR to coordinate a great collaborative effort for developing new effective drugs. In the 1960s and 1970s, mefloquine and halofantrine were developed at WRAIR and approved by the FDA [389]. Further evidence of the interest of the military in malaria are the discoveries of the etiology itself. The mosquito vector was discovered in 1897 by Surgeon-Major Ronald Ross, of the Indian Medical Service, who could identify the parasite in the gastric wall of a mosquito after it had previously bitten an infected man [390], a discovery for which he was awarded the Nobel Prize for Physiology or Medicine in 1902. *Plasmodium* had been discovered, instead, in 1880 by the French military physician Charles Louis Alphonse Laveran in Algeria, in the blood smear of a man who had recently died of malaria; for this discovery, Laveran was awarded the Nobel Prize for Physiology or Medicine in 1907 [391]. The interest of the military in malaria is further witnessed by the efforts of WRAIR in developing a malaria vaccine. The need for an effective vaccine is highly felt in the military, because the disease is present in large world areas with an estimated 3 billion people at risk in 2013 [392], and it has heavily hit the military in the different wars [380], by resulting in high incapacitance and influencing the battle outcome. Despite the complexity of developing a vaccine for parasitic diseases and the different trials for an effective anti-malaria vaccine, the only approved vaccine for malaria is a recombinant bivalent vaccine, expressing HBsAg together with the B-immunogenic repeat polypeptidic part of the circumsporozoite protein and T epitopes (RTS-S) of *P. falciparum*, whose study started in 1984 from a collaboration between WRAIR and SmithKlineBecham, currently GlaxoSmithKline, and the vaccine was developed in 1987 [393,394]. Although this vaccine, Mosquirix™, has only shown partial protection against the infection in children 5–17 months, it has been approved by the WHO, which in October 2021, has recommended its widespread use, with a four-administration schedule, after a trial on over 800,000 children in Africa showed a substantial safety and partial protection [395]. It is the first approved anti-parasite vaccine. This represents a milestone, even though prevention for the military continues to be based on chemoprophylaxis and protective personal measures, including insecticide-impregnated uniforms, sleeping under bed-nets, and applying insect repellents on the skin [396]. The researchers of WRAIR are involved even in the research on monoclonal antibodies. In fact, they have recently been coauthors of a small trial on a low-dose subcutaneous or intravenous monoclonal antibody for preventing malaria, which resulted to be safe and protective [397]. Malaria at the start of the twentieth century in 1900 was present in the Panama Canal, representing a great threat to the workers. The Surgeon General of the US Army sent Col. William Crawford Gorgas there. Col. Gorgas had worked in Habana with Carlos Finlay and Walter Reed by successfully contrasting another mosquito-borne disease, such as yellow fever, as a chief health officer. The fight against the vector carried out by Gorgas, by drawing swamps and spraying oil, putting a screen on the windows and providing prophylactic quinine at a dose of 150 mg twice daily was highly successful. In 3 years, the cases of malaria went from 800 per 1000 workers to 16 [389]. Vector control was the strategy followed in all the countries, as may be seen in Figure 1. During WWI, malaria represented an unexpected, invisible enemy capable of inducing at least 1,500,000 cases in all the war theaters and in the different Armies, with a case-fatality rate ranging from 0.2% to 5%, depending on the war theater [398]. A peculiar event was the simultaneous outbreak of malaria from *P. falciparum* and Spanish influenza in the Egyptian Expeditionary Force in Palestine, a condition that led to a halt of military operations and created even relevant diagnostic problems, which were only clarified by the performed autopsies [192]. 

During WWII in the US military only, the cases of malaria were at 572,950, ranking second after diarrhea [399], a figure which consents to understand the dimensions of the problem and the great interest of the military for malaria. In the US military during the Korean War, the admissions for malaria were 34,864, whereas in the Vietnam War, they were 65,053 [400]. Even the French military paid a heavy toll on malaria in the nineteenth century. In 1895, during the conquest of Madagascar, approximately 6000 deaths for malaria occurred versus less than 30 killed in action [401]. In Macedonia, in 1916, 50% of the French military had malaria with 600 deaths [402]. Each year approximately 40,000 French military personnel are deployed to or travel through over a dozen malaria-endemic areas [403]. From 1986 to 2011, 13,543 malaria cases were observed in the French Armed Forces, 2.2% were serious cases with eleven deaths, five of which occurred in Gabon or upon return from this country. The *P. falciparum* species was responsible for 78.1% of attacks and *P. vivax* for 16.4% [403]. It should not be forgotten that the military may even become involuntary malaria vectors by going to endemic areas and returning home. This may more easily happen with *P. vivax*, considering that the disease may manifest for months or years after the infection. This was observed in the USA during the Korean War [404,405] and the Vietnam War [406], while the Russians imported thousands of malaria cases from *P. falciparum* during the war in Afghanistan [407]. The military contribution to malaria is in the etiology and vector identification as well as in some therapies and vaccine development.

### 3.6. Lymphatic Filariasis

Lymphatic filariasis is a highly incapacitating mosquito-borne disease, the second most common after malaria among vector-borne diseases [408]. The vectors may be *Anopheline*, *Culex* and *Mansonia* mosquitoes, which bite at night, whereas in the Pacific, the vector is the *Aedes* mosquito, which bites during the day [409]. The etiological agents are three species of nematode parasites, *Wuchereria bancrofti*, responsible for 90% of cases, *Brugia malayi*, present in East-Southeast Asia, and *Brugia timori*, present in the Timor area [408]. The worms have a complex cycle, are transmitted by the bite of the vector and localize in the lymphatic vessels. The adult worms generate microfilariae, which migrate into lymph and blood channels. The disease may be acute or chronic, and its main manifestations are fever, lymphangitis, lymphadenopathy with pain, and in males, scrotal edema of the acute form, whereas the chronic form is highly invalidating with elephantiasis, lymphedema and hydrocele [409]. Nearly one billion people in tropical areas are at risk of lymphatic filariasis, and it is estimated that approximately 36 million people are incapacitated as a consequence of chronic lymphatic filariasis [409]. There are three effective drugs, albendazole, ivermectin, and diethylcarbamazine citrate, which may prophylactically be administered to at-risk populations, as recommended by the WHO, with annual mass drug administration (MDA), a strategy that aims to eradicate filariasis [410].

Filariasis was an unexpected invisible enemy for the US troops during WWII in the Pacific war theater, with many thousands of hit men, more than 10,000 among Navy and Marine Corps, and an unspecified number of illnesses among the Army troops, with a global amount estimated at 14,000–16,000 men having clinical evidence of infection and some battalions with 70% of their contingent out of service [368,408]. This acute form of lymphatic filariasis showed a trend of spontaneous clinical resolution once the disease was recognized and the patients were moved away from the endemic countries [409]. The contributions of the US military researchers to the fight against filariasis ranged from diagnosis to treatment, where the evidence that eradication strategy was possible through a safe and effective treatment was provided [368].

### 3.7. Schistosomiasis

Schistosomiasis is an infectious disease caused by a trematode parasite, which may penetrate the human skin in contaminated freshwater. Humans are the main definitive host, whereas intermediate hosts are snails living in freshwater. The cercariae are released from the snails and penetrate the human skin and, after 5–7 weeks, mature into adult worms, capable of generating eggs, which are retained in the body if not excreted by feces or urine. The excreted eggs become miracidia, which infect the snails, thus maintaining the cycle [411]. Three main species of schistosomes infect humans, *Schistosoma mansoni, Schistosoma haematobium*, and *Schistosoma japonicum*. The first two are found in Africa and the Middle East, *S. mansoni* is also found in the Americas, while *S. japonicum* is found in Asia, mainly in China and the Philippines [411]. Eggs that remain in the body are responsible for the host inflammatory response, which may create granulomatous lesions in the host tissues, intestine and liver in the case of *S. mansoni* and *japonicum*, and in the bladder and urogenital tract in the case of *S. haematobium*. The infection is more frequent in adolescents and tends to decrease in adults and older adults; it may be acute, with sudden onset of fever, malaise, abdominal pain, myalgia, headache, eosinophilia (Katayama syndrome), and it may be chronic, with symptomatology that may be hepatosplenic or urinary, depending on the type of *Schistosoma* [411]. It is estimated that nearly 800 million people are at risk, and over 250 million are infected, 90% of whom live in Sub-Saharan Africa [412]. Even for schistosomiasis, there is the possibility of safe and effective chemotherapy with praziquantel, which may even work for prevention; thus, its annual administration is recommended by the WHO with the objective of eradicating the disease, in association with the fight against the intermediate host [411]. Vaccines for human use are not available yet [412].

Schistosomiasis has accompanied the military in war since the period of Napoleonic wars. In WWI, several hundreds of British and Australian troops were infected in Egypt and the Middle East, whereas in WWII, over 1500 British and African troops became infected in Nigeria. Hundreds of US soldiers were infected during the liberation of the Philippines. Although schistosomiasis in the military is far from the number of cases of malaria and filariasis, it may represent a further obstacle to military readiness during operations in endemic areas. Thus, the US military provided relevant contributions to the diagnosis, in the demonstration that praziquantel therapy could revert advanced hepatosplenomegaly and obstructive uropathy, and in the indications of the right conditions of salinity and chlorination for inactivating cercariae [368].

### 3.8. Trypanosomiasis

There are two types of trypanosomiases, the African and the American trypanosomiasis. African trypanosomiasis is caused by the parasites *Trypanosoma brucei gambiense* or *rhodesiense* transmitted by the bite of the tsetse fly, *Glossina spp*. *T. brucei* is a flagellated protozoan parasite whose transmission by the bite of the tsetse fly was discovered in 1895 in Zululand by the British military physician David Bruce [413]. Trypanosomes were first found in the blood of a European man in the Gambia by Dr. Robert Michael Forde [414] and Dr. Joseph Everett Dutton [415], whereas trypanosomes in cerebrospinal fluid were first observed, described and put in connection with sleeping sickness by Dr. Aldo Castellani, later on, full Professor of Tropical Medicine at the University of Rome and General of the Italian Navy [416]. *T. brucei gambiense* is responsible for 97% of the human African trypanosomiasis and is present in western and central Africa, whereas *T. brucei rhodesiense* is responsible for under 3% of African trypanosomiasis and is present in eastern and southern Africa [417]. *T. brucei rhodesiense* and *gambiense* induce an invariably fatal disease if not treated; however, the first has a quicker clinical course of approximately one year, whereas the other has a more chronic course of a few years. During the twentieth century, three severe epidemics of sleeping sickness were registered, the first in 1896 until 1906, mainly in Uganda and Congo, the second in 1920, and the last in 1970 through the late 1990s [417]. Considering the high number of deaths in the order of hundreds of thousands, the most renowned scientists were engaged in the search for effective therapies, including Nobel Prize winners, such as Charles Laveran and Robert Koch. Moreover, in the second outbreak of the 1920s, a winning move was suggested by the French military physician Eugène Jamot, represented by the institution of mobile teams for capillary detection and treatment of sleeping sickness cases; this strategy, organized in Cameroon, was highly successful, considering that the sleeping sickness prevalence decreased from 60% in 1919 to 0.2–4.1% in 1930 [418]. The progress in therapy, the organization of the mobile teams as well as the vector control achieved good results, as witnessed by the prevalence in 2009, for the first time under 10,000 cases, and in 2019 and 2020, when 992 and 663 cases, respectively, had been reported [417].

American trypanosomiasis is caused by *T. cruzi* and is transmitted by the feces of blood-sucking bugs belonging to the subfamily of *Triatominae*, as discovered by the Brazilian hygienist Carlos Chagas in 1909 [419]. The infection, named Chagas disease, has two clinical phases, the first is clinically nonspecific and is 4–8 weeks long, during which parasitemia is present, whereas the second phase is clinically symptomatic in 15–30% of the infected people, and presents organ damage usually 10–25 years after the first clinical phase, at cardiac level, with potentially fatal cardiomyopathy, and at the digestive level, with megaesophagus and megacolon [419]. Currently, 7–8 million people are estimated to be infected by *T. cruzi* in Latin America, and 25 million are at risk. In 2008, more than 10,000 people died from Chagas disease [419]. Moreover, due to migration, it is estimated that 400,000 people infected by *T. cruzi* are present in the world outside Latin America, three-quarters of whom are in the USA [419]. As for African trypanosomiasis, no vaccines are available, but only drugs for therapy. Despite that trypanosomiasis has rarely represented a problem for the military, the intervention of the military in the study of the disease and in the means for fighting it is demonstrated by the relevant discoveries made by military physicians, including David Bruce, Charles Laveran and Eugène Jamot, as well as by the studies performed by the military researchers at the US Army Medical Research Unit (USAMRU) in Kenia on epidemiological and clinical aspects of African trypanosomiasis and at WRAIR in the immunology, but mainly therapy, of the two types of trypanosomiasis [368].

### 3.9. Other Parasitic Diseases

Other parasitic diseases able to heavily influence the operational readiness of the military are intestinal parasites, such as *Entamoeba histolytica*, which is a leading cause of diarrhea worldwide [420]. It is estimated that over 500 million people are infected and 40,000–100,000 individuals annually die from amebiasis worldwide [22] (Table 1). The military may be exposed to infection if deployed to endemic areas [420,421]. The US military physician Charles F Craig in the first half of the last century provided relevant contributions to the serological diagnosis as well as to clinical and pathological observations [422,423,424], whereas the military physician E. Vedder could demonstrate the amebicidal power of emetine in vitro, thus paving the way to its application in the treatment of amebiasis [425].

Hookworm is a soil-transmitted helminth disease caused by *Ancylostoma duodenale* or *Necator americanus.* It is estimated that worldwide 740 million people are infested, mainly in China, Sub-Saharan Africa, East Asia and the Pacific Islands [426]. These worms reach the digestive mucosa where they feed themselves by inducing chronic hyposideremic anemia. At the start of the last century, the US military physician Bailey K Ashford deeply investigated the hookworm from *Necator americanus* in Puerto Rico, which was responsible for the death of almost 12,000 people per year [426,427]. Although in general this parasitosis does not represent a threat for the military deployed to endemic areas, nonetheless, it was reported in some USA military soldiers deployed to Grenada [428] and Vietnam [368] and in the Singaporean military deployed in jungle training activities in Brunei [429].

### 3.10. Human Immunodeficiency Virus (HIV)

Human immunodeficiency virus (HIV) is a retrovirus discovered in 1983 by Françoise Barré-Sinoussi and Luc Montagnier at Pasteur Institute in Paris [430], for which they were awarded the Nobel Prize for Physiology or Medicine in 2008. HIV is the etiological agent of the acquired immunodeficiency syndrome (AIDS), first described in 1981 in the USA in homosexual men [431]. The virus was independently isolated and described a few months later by Robert Gallo, then working at the National Cancer Institute, in Maryland, USA, who provided further evidence of the association between the virus, which was called human T-lymphotropic virus (HTLV)-III, considering that Gallo had previously described HTLV-I and II, and AIDS [432]. A long controversy ensued over which of the two scientists had discovered the virus, which ended in 1987 with an agreement reached by USA President Ronald Reagan and French Prime Minister Jacques Chirac, with a division of the royalties of the diagnostic test between the two countries [433]. It was only in 1986 that the virus was named HIV. HIV enters the host following sexual intercourse, blood/blood derivatives administration, or from mother to fetus and through its envelope glycoprotein 120 links to the CD4 helper T-lymphocytes, which are slowly but progressively destroyed, thus reaching low values, <200/µL, which are incompatible with maintaining good health, thus paving the way to opportunistic infections and cancers, until patient death [434]. From its first description in 1981 until 1996, when an effective therapy was made available [435], AIDS was invariably fatal and present at a global level, particularly in Sub-Saharan Africa, where HIV infection still represents more than 70% of the estimated global infections, which were 36,848,000 in 2019, 0.5% of the world population, showing a slow general decline. Except for in the Americas and Europe, where the rate of infection increases each year; the mortality is instead in global decline [436]. One million and a half new HIV infections were reported in 2020 and 680,000 AIDS-related deaths [13] (Table 1). Currently, therapy is not resolutive; however, it allows a good quality of life with only one pill per day. Instead, no effective vaccine is yet licensed, although therapeutic and prophylactic vaccines have been developed and tested, in one case until phase 3.

HIV infection has been considered of high interest for the military since the first description, considering that it is a sexually transmitted disease, a type of disease that is particularly widespread among the military. It is severe and invariably fatal and it may be transmitted by blood, thus representing a risk for the soldier as a “walking blood bank” [437]. However, in the US military, this interest increased and became an organic scientific HIV program after the death of a recruit for generalized vaccinia after smallpox vaccination due to an unrecognized HIV infection [438]. This dramatic event prompted the US military authorities to introduce the screening for HIV dating from October 1985 to all applicants for military service as well as to all active-duty forces, in order to protect the recruits from the side effects of the living vaccines in case of HIV positivity and to protect the “walking blood bank” [437]. The HIV-positive applicants were considered unfit for military service, whereas the infected military members, even if retired, were offered a program of periodical checks, which represented an opportunity for helping infected patients and obtaining information on this new disease [437]. This was even more relevant because it occurred during a period of poor knowledge of the disease, due to the fact that generalized HIV screening was difficult, rather nearly impossible, to be carried out for the strong resistance of the associations of patients, who had fear of social stigma, considering that the disease was first described in homosexual men and drug addicts [439]. This popular resistance represented a strong brake everywhere to the possibility of knowing at least the epidemiology of the infection by generalized screening, which could not be performed. Thus, the strategy of the US military of introducing compulsory HIV screening for the applicants to military service allowed for one to collect information on the spread of the infection among adolescents and young adults, and it was estimated that the rate of positivity in the military could be lower than in the corresponding general USA population [437,440,441]. Meanwhile, even relevant pathogenetic and clinical observations were collected, including the defective regulation of the Epstein–Barr virus (EBV), which could expose AIDS patients to the risk of developing EBV-containing lymphomas [442], the defective response to vaccinations in early stage HIV-infected patients [443], the systemic inflammation and immune activation as hallmarks of the disease, as witnessed by the high interleukin-6 levels [444], and finally the Walter–Reed staging classification system, which has represented a guide for the judgements of fitness to the military service and for clinicians during their clinical activity [445]. Moreover, the experience of the US military researchers allowed one to calculate the rate of decline of CD4+ lymphocytes dependent on the stage of the disease [446]. Furthermore, the Walter–Reed Retrovirus Research Group made relevant observations even in the field of malignancies in the course of AIDS, neurocognitive impairments, and anti-retroviral therapy [437]. The WRAIR was even the lead agency to test in phase 3 an HIV-1 vaccine in Thailand, which showed only a modest benefit but was useful for future research [447,448]. Despite great advancements having been achieved in a relatively short time, HIV infection remains a public health global threat. However, the US military researchers at WRAIR allowed for gaining early precious insights into the knowledge of the disease, which have represented a reference point everywhere.

### 3.11. Hepatitis C

Hepatitis C virus (HCV) is an RNA flavivirus for the first time identified in April 1989 [449,450], which is estimated to have infected 58 million people in all six WHO regions, with an annual incidence of 1.5 million new cases and 290,000 deaths per year, as a consequence of cirrhosis and hepatocellular carcinoma [14] (Table 1). HCV may induce an acute, usually asymptomatic and spontaneously clearing, disease in approximately 30% of cases, whereas in the remaining 70%, the disease becomes chronic, with 15–30% of these chronic cases evolving to cirrhosis in approximately 20 years [14]. HCV is first transmitted by exposure to blood or blood derivatives, more rarely by sexual intercourse, and the incubation time ranges from 2 weeks to 6 months; however, acute infection is asymptomatic in approximately 80% of subjects; thus, it is generally undetected. The diagnosis is based on a serological approach for the research of specific antibodies and a confirmation test in case of positivity with a molecular approach to identify the viral RNA. No vaccine is still available; the direct-acting antivirals (DAAs) are highly effective, able to clear over 95% of the chronically infected patients in a period of 12–24 weeks, depending on the presence or not of cirrhosis [14]. However, currently, access to DAAs is still too limited, even in affluent countries [451], and it is estimated that less than 10 million patients have completed the treatment with DAAs [14].

The interest of the military in HCV is mainly due to the need of protecting the “walking blood bank” and, despite that no compulsory HCV screening at enrollment in the military is generally required, the percentage of HCV positivity is usually low, around 0.5% or even less in the USA, where it is lower than in the civilian corresponding population [452] or in the few European countries for which data are available [453,454]. Even in Morocco [455], India [456], and Brazil [457], HCV prevalence in the military is quite low, 0.245%, 0.45%, and 0.7%, respectively. Conversely, in the civilian blood donors of Sierra Leone, HCV positivity rate was 1.2% [458], and in the patients of a military hospital in Rwanda, the rate of HCV-infected patients was 9.6% [459]. The highest world prevalence of HCV viremia at 6.3% is in Egypt [460], as a consequence of the anti-schistosomiasis campaign carried out with unsafe injections [461]. However, a massive effort to voluntarily screen all the adult Egyptian population in order to offer the DAAs by the government allows for foreseeing that in a short time Egypt may achieve an HCV viremic prevalence of 0.5%, which is similar to the nearby regions [462]. In the US military, a cost-effectiveness analysis of introducing compulsory HCV screening at enrollment versus the cost of treatment for the military for those who may be HCV positive, has established that screening is cost-effective [463]. Moreover, a study carried out on the HCV prevalence in 10,000 soldiers returning from international missions to Iraq and Afghanistan allowed one to observe that 23 subjects were HCV positive; the majority of them (18/23) were positive at enrollment, with only five cases of positivity being service associated, thus even more underlying the economic advantage of introducing compulsory screening [463]. With an effective vaccine still unavailable, the only policy for eradicating the disease is to actively search for positive subjects and to offer them the DAAs. This policy may even help the military to protect the “walking blood bank”.

### 3.12. Hepatitis E

Hepatitis E virus (HEV) has been first suggested to be the cause of an outbreak of hepatitis in Kashmir (India) at the end of the 1970s and reported in 1980 [464]. In 1991, the virus E was cloned [465], a relevant step not only for the precise viral description but even for its use in diagnosis and vaccine development. HEV is responsible for a generally acute hepatitis, transmitted by fecal–oral route, through contaminated water in hyperendemic and endemic countries. However, HEV is even present in the developed world as a zoonosis, for ingestion of undercooked meat from infected animals, such as swine, boar and deer. Four viral genotypes exist, the first two mainly occurring in hyperendemic and endemic countries (Africa, Asia, Mexico and Brazil) and the 3–4 especially occurring as sporadic cases in developed countries [466]. Despite that HEV generally induces acute hepatitis, chronic hepatitis may be observed in immunocompromised people, such as transplanted patients. Pregnant women present high mortality, ranging from 10% to 50% [467]. It is estimated that globally each year, 20 million people are HEV infected, 3.3 million of whom are symptomatic, with 44,000 deaths [23] (Table 1). Although the fecal–oral route is the main way of transmission, the persistent viremic phase does not exclude that HEV may even be transmitted by blood [466]. Two recombinant vaccines for HEV were developed, the first of which has been studied by the US military researchers of WRAIR together with GlaxoSmithKline in phase 2 [468], but it did not further progress, possibly for lack of commercial value [467]; the other seems to be safe and effective, and it has been licensed in China, but it is not recommended by the WHO for extensive use due to insufficient information about safety and efficacy in different categories of patients [469]. Thus, HEV has been included in this review among the non-vaccine-preventable infectious diseases.

The interest of the military for HEV is witnessed by having found different outbreaks in deployed troops in hyperendemic countries, the first of which was among Russian troops in Afghanistan in 1981 [470], and others were described by the US military [471,472,473,474,475,476]. A complete resolution of the HEV issue for the military may only come from the availability of a safe and effective vaccine that protects against all viral genotypes. Meanwhile, careful control of water and food is the better prevention for deployed troops. Furthermore, doubt on possible blood transmission is relevant for the military in consideration of the principle of the “walking blood bank”.

### 3.13. Chikungunya

Chikungunya is an RNA viral disease transmitted by *Aedes aegypti* and *Aedes albopictus* mosquitoes, characterized by fever, arthralgia, and skin rash [477]. The disease is generally self-limiting, with mortality way lower than 1%. The arthralgia is incapacitating and in nearly two-thirds of patients, may last for more than one year after the infection [478]. The disease was described for the first time in Tanzania in 1952 and is present in Africa and South Asia, where relevant outbreaks have been observed. However, the disease has even been described in temperate climates, considering that an outbreak of more than 200 patients has been reported in Italy in 2007 [479]. No licensed vaccine is available, although the US military at WRAIR has worked on a project for developing an inactivated and subsequently a live attenuated vaccine, approved as an investigational new drug (IND) [480]. The US military considers Chikungunya a relevant threat even though currently no higher risk for the military compared to the civilian population has been observed [480]. When Chikungunya cases were considered among military personnel deployed to endemic areas, low prevalence was observed [481]. Conversely, the situation is different when observed among the military stationed in areas with outbreaks. The most accurate study on the military was on the French policemen during the outbreak on Reunion Island, where 266,000 out of the 775,000 inhabitants presented symptoms of the disease [477]. Out of 662 policemen, 128 (19.3%) were serologically positive for the Chikungunya virus, and 3.2% were asymptomatic. Chronic arthralgia was reported in over 90%, acute fever in nearly 90%, skin rash in over 50% and tiredness in all [482]. During this outbreak, the French health authorities asked the US military about the possibility of using the IND live vaccine [480]. Among the US military, 78 cases of Chikungunya were reported in Puerto Rico between 2010 and 2016 and 118 in 2014–2015 [481].

### 3.14. Zika

Zika is an RNA Flavivirus transmitted by the bite of *Aedes* mosquitoes, first isolated in 1947 in Uganda from a macaque monkey and in 1954 in Nigeria from human cases. Since then, sporadic cases were reported, whereas it reemerged in this century, with the first outbreak occurring in 2007 in Yap, Federated States of Micronesia, followed by another outbreak in 2013 in French Polynesia [339,483]. In May 2015, Zika virus reached Brazil, where an estimated 440,000–1,300,000 persons were infected [483]. The WHO declared the Zika virus a public health emergence based on the net increase in microcephaly in Brazil (20/10,000 live births vs. 0.5/10,000 live births in the previous years) [483]. Another dreadful complication is Guillain–Barré Syndrome. Despite that no vaccine is still available, many candidate vaccines are studied, seven of which are in phase 1. WRAIR is studying an inactivated vaccine in collaboration with the Beth Israel Deaconess Medical Center [338].

### 3.15. Crimean–Congo Hemorrhagic Fever

Crimean–Congo Hemorrhagic Fever (CCHF) is the most widespread tick-borne disease in the world and the second (after dengue) viral hemorrhagic fever in the world [484]. The disease is caused by an RNA virus transmitted by the bite of a hard tick of the species *Hyalomma.* However, even contact with infected animals may transmit the infection. Moreover, inter-human transmission has also been reported. The infection may run asymptomatic in a large percentage of cases, up to 90% in hyperendemic areas [485]; however, symptomatic cases may be severe with high mortality of up to 40% [484]. First described during WWII in Soviet military personnel in Crimea [486] and in 1956 in Congo (hence the name), CCHF is now present in Asia, Africa, and Eastern and Southern Europe, including Spain [484,487]. There is no therapy nor available vaccine, even though in Bulgaria, an inactivated vaccine from suckling mouse brain was licensed and used for at-risk categories of workers, including the military. However, less reactogenic and more effective are considered the inactivated vaccines from cell cultures [484]. The Turkish military has reported the preparation and successful use in patients with high-level CCHFV viremia of hyper-immunoglobulins. They were prepared by the sera of 22 convalescent subjects and only one administration of 400 Kubar Units of hyperimmune immunoglobulins to 15 patients with a high level of viral copies (≥108/mL), allowing 13 of them (86.6%) to be cleared from the virus, whereas two died [488]. Passive immunotherapy may be a promising tool in CCHF, in consideration of the severity of the disease and the current lack of effective therapy or vaccines. Based on the high mortality of the infection, the possibility of dissemination by aerosols and the current lack of therapy or vaccine prevention, CCHFV has been included in the high biohazard agents, to be handled only in biosafety levels 3–4 [489] and among biological agents of category C by the CDC [57,484].

### 3.16. Hantaviruses

Hantaviruses are RNA viruses of the family *Bunyaviridae*, of which three main types are known, one present in Finland and Scandinavia, called Puumala virus, able to induce a milder disease, the nephropathia epidemica (NE), the Hantaan (HTNV) old world type, which causes the hemorrhagic fever with renal syndrome (HFRS), a severe clinical condition with a mortality of approximately 12%, and the new world type, able to induce the most severe hantavirus cardiopulmonary syndrome (HCPS), burdened with a mortality of approximately 40% [489]. The first outbreak of over 10,000 cases of HFRS was reported in WWII in Finland, among Finnish and German troops [490], and during the Korean War, over 3000 United Nations troops developed HFRS [491]. However, the etiologic agent was only identified in the 1980s, whereas the new world type was only identified in 1993 [489]. The viruses are present in rodents and are transmitted to humans by aerosol of contaminated biological fluids or feces, a condition which may induce consideration of hantaviruses as potential biological weapons, for easy dissemination, high clinical severity and lethality, lack of therapy and vaccine [487,492]. The military seems to be particularly exposed, due to the easier contact with rodents, which is associated with the disruption of the rodent habitat linked to the war and with the operational training in the field [493]. During military training activity in Germany in 1990, a hantavirus outbreak from the Puumala virus was observed with an attack rate of 8.5% [493]. In the former Yugoslavia, particularly in Bosnia–Herzegovina and Croatia, many outbreaks of HFRS, some of which were large, have been reported between the 1950s and the 1990s, thus representing a likely threat for NATO forces deployed there, considering that over 50% of soldiers in the field are exposed to possible HFRS risk factors [491]. Moreover, in the military personnel in Europe in war, activity or maneuver-theater overlapping symptomatology of HFRS and HCPS has been reported, irrespective of the isolated virus, but coherent with the possibility that two genetically related viruses, infecting both through the same respiratory way, both inducing a sort of “cytokine storm”, may determine damages in two target organs, lung and kidney, which are generally addressed separately [494]. An inactivated vaccine, Hantavax^®^, has been developed and licensed in Korea and has been used in the Korean army; however, a study carried out in the period 2009–2017, for evaluating its effectiveness in the progression of HFRS, failed to demonstrate significant protection against the progression of HFRS [495]. US military researchers are working on a bivalent DNA vaccine against the Puumala virus and HTNV and have successfully completed a phase 2a study, by demonstrating the immunogenicity of this tentative vaccine [496]. Hantaviruses have been included among the biological agents, category C [57].

### 3.17. Other Arboviral Diseases

West Nile virus was first observed in Uganda in 1937. It is a flavivirus primarily transmitted by the bite of *Culex* mosquitoes. The infection in 80% of cases is asymptomatic, and in symptomatic cases the clinical course is generally not severe, characterized by fever, rash and lymphadenopathy. More recently, at the end of the last century, in the outbreaks registered in Romania and in New York, the trend to target the CNS with meningoencephalitis was more pronounced, as well as gastrointestinal symptomatology. There is no specific approved therapy or effective human vaccines. Thus, for the military, the only prevention consists of a vector control strategy [497]. West Nile virus has been considered a potential biological agent, category C [498].

Rift Valley virus was first described in the Rift Valley in Kenia in 1931. The RNA virus is mainly transmitted by mosquitoes, and the infection is generally benign or paucisymptomatic. However, it may be complicated by encephalitis, hepatitis, ocular disorders, nephritis, and hemorrhages; the mortality is generally low, but percentages as high as 22–28% have even been reported. In this case, no specific therapy nor effective vaccine is available [499]. The Rift Valley virus should be considered when observing fevers of unknown origin in the military [500].

### 3.18. Acute Respiratory Syndrome

An acute respiratory syndrome is frequently observed in the military, particularly in the first weeks of recruits in the training phase. It is calculated that 25,000–80,000 US recruits suffer acute respiratory disease (ARD) each year and that 200,000–600,000 US service members had ARD each year during the influenza seasons of the year 2012 through 2014 [501]. The socio-environmental conditions favoring ARD for trainees are overcrowding, psychological and physical stress, sleep deprivation, exposure to dust, smoke and extremes of temperature [501]. ARD presents with a syndrome of common cold or pneumonia depending on whether the higher or lower airways are interested by infection-induced inflammation, irrespective of the etiological agent. Even though the common cold is primarily determined by rhinoviruses (approximately in 50% of cases), even coronaviruses, adenoviruses, influenza and parainfluenza viruses, respiratory syncytial virus (RSV), human metapneumovirus, and group A streptococcus (*Streptococcus pyogenes*) may be found. In analogy, pneumonia is primarily induced by *Streptococcus pneumoniae*; however, adenoviruses and influenza virus may even be responsible. The intermediate clinical picture of bronchitis with cough may be induced by adenoviruses, influenza virus, *Mycoplasma pneumoniae*, *Chlamydophila pneumoniae*, and *Bordetella pertussis* [501]. There is a large overlap of induced syndromes and etiological agents; thus, a precision diagnosis cannot be made based on the clinical picture only, but it has to be laboratory driven, a need that may not always be satisfied in operational activities or, even more, in wartime. Despite that several etiologic agents are potentially implicated in ARD induction, those that are more frequently observed in trainees are adenoviruses, influenza virus, and *S. pyogenes* [502]. For adenoviruses 4 and 7, a safe and effective, FDA-approved, live, oral vaccine is available, and is administered to US recruits as the only military in the world [501]. Influenza virus, similar to adenoviruses, has been treated among the vaccine-preventable infectious diseases, whereas *S. pyogenes* has not been treated yet. It has represented a threat for the military during WWII, associated with scarlet fever, and more recently in the period 1990–2011 in the US military, when at least 17 outbreaks of *S. pyogenes* have been observed, all treated with antibiotic chemoprophylaxis, generally consisting of benzathine-penicillin G; however, even erythromycin and azithromycin may be used. *S. pyogenes* generally induces pharyngitis, which resolves without complications. Less commonly, *S. pyogenes* may induce suppurative and invasive infections, including meningitis, brain abscesses, pneumonia, and necrotizing fasciitis. The mortality of uncomplicated pharyngitis is <1%, whereas in the complicated forms, it ranges between 15% and 25%. Finally, *S. pyogenes* in some subjects may even induce autoimmune diseases, such as acute rheumatic disease, which may manifest with either endocarditis or glomerulonephritis [501]. *Streptococcus pneumoniae* has already been treated among the vaccine-preventable infections as well as for *Bordetella pertussis* and coronavirus (as far as SARS-CoV-2 is concerned), whereas rhinoviruses, para-influenza viruses, RSV and human metapneumovirus, as well as *Mycoplasma pneumoniae* and *Chlamydophila pneumoniae*, have not been treated yet. Paradigmatic of the difficulty of differential diagnosis in ARD and the need to set up an integrated surveillance system based on a network of well-equipped laboratories and skilled researchers, as in the US military, to respond to the new diagnostic challenges posed by emerging infectious diseases, is the request for help of the Jordanian Ministry of Health in April 2012 addressed to the NAMRU-3 in Cairo, Egypt, to clarify the cause of 11 cases hospitalized for ARD, two of whom died soon after hospitalization. These cases were carefully analyzed for influenza, parainfluenza types 1 and 3, human metapneumovirus, human coronaviruses (including SARS-CoV), and adenovirus, but all the results were negative. In September 2012, Middle-East respiratory syndrome due to coronavirus (MERS-CoV) became widely known, and NAMRU-3 received from the CDC the biological material for making diagnoses. The cases were investigated again, in agreement with the Jordanian Ministry of Health, and the two cases deceased were clearly positive for all three genes of MERS-CoV [501].

RSV is a worldwide respiratory virus, present in over 90% of 5-year-old children. The immunity against RSV is short; thus, it is not difficult to be reinfected even by the same strain. The relevance for the military was first observed in the US military in 1959, thereafter confirmed in the military trainees and even during the Vietnam war. In some studies, in US and UK military trainees, it was present in 11% and 14% of the trainees with respiratory symptoms, respectively, behind adenovirus, which was prevalent in both, with 48% and 35%, respectively, and influenza virus, which was 11% in the US and 19% in the UK trainees [503,504]. Conversely, in a large study on the Dutch military, RSV was found in only 3% of radiologically confirmed pneumonia cases [505].

*Mycoplasma pneumoniae* was first isolated in 1942 from a recruit with primary atypical pneumonia, and afterward, it was found in recruits with pneumonia with variable, but always rather high, percentages, ranging from 6–10% to >50%. In the 1960s, primary atypical pneumonia from *M. pneumoniae* was differentiated from that due to adenovirus, and a favorable response to di-methyl-chlortetracycline was observed [501].

*Chlamydophila pneumoniae* has been recently identified as a relevant etiologic agent for ARD/pneumonia. Recent studies by US Navy researchers have observed *C. pneumoniae* as the etiologic agent of 10–15% of all pneumonia cases in military recruits [501].

In conclusion, acute respiratory syndrome is one type of pathology of great interest for the military, especially for recruits and trainees, probably for environmental living conditions. Acute respiratory syndrome may be due to a series of etiologic agents, for a minority of which preventive vaccination is available. However, even in these cases, vaccine-induced protection is not absolute, considering that not all the possible pathogenic strains are included in the vaccine, as for *S. pneumoniae*, or the reaction of immune response is not completely protective against the infection, as in the case of influenza and SARS-CoV-2, due to the type of antigen and the high variability of these RNA viral agents. Finally, for adenovirus, as already commented in the dedicated paragraph, it is not understandable why the vaccine is only administered to the US military, although the epidemiological problem is present in the military of other countries.

### 3.19. Acute Diarrheal Syndrome

Diarrheal diseases have accompanied the military for a long time, particularly when deployed and in wartime. Depending on the place of deployment, the etiology may change, with cholera being a dreadful threat, mainly in Southeast Asia and in the nineteenth century. Currently, the main threats are represented by *Shigella*, *Salmonella*, *Campylobacter jejuni*, enterotoxigenic *Escherichia coli*, and the Norwalk virus. Cholera had a case-fatality rate of over 60% without therapy, reduced to 20–30% with therapy before 1960. The studies of the US military physician Robert A. Phillips started at the end of the 1940s and continued into the 1950s, by introducing the intravenous rehydration treatment, which could reduce the death rate from 30% to 0.6% [297]. Currently, oral dehydration is an affordable treatment even in developing countries and in diarrheal syndromes not caused by *Vibrio cholerae*. The study of Phillips and his collaborators saved the life of millions of people. As cholera is a vaccine-preventable infection, in this work, it has already been treated in the section dedicated to vaccine-preventable infections. Even typhoid fever, in the past a relevant cause of morbidity and mortality for the military, particularly in wartime, has already been treated among the vaccine-preventable infections. Despite that diarrheal syndrome was already present during the American Revolutionary War and the American Civil War, during the Spanish–American War, a board led by Walter Reed was able to identify typhoid as the etiologic agent of diarrhea, and water sterilization, together with improved sanitation, could significantly reduce the morbidity from 85/1000 in 1898 to 6/1000 in 1900 [297]. During Operation Desert Shield (September–December 1990, Saudi Arabia), a study on soldiers with diarrheal syndrome had shown that 57% of the US troops reported at least one episode of diarrhea, and 20% were temporarily unable to perform their duties because of it. The cause of diarrhea was a bacterial pathogen in 49.5%, with the most frequently isolated enterotoxigenic *E. coli*, followed by *Shigella sonnei.* A few cases of vomiting in addition to diarrhea were due to the Norwalk virus [506]. More recently, in the US troops deployed to Iraq and Afghanistan in 2003, 76% had at least one episode of diarrhea, 45% were incapacitated for a median of 3 days, and 17% were confined to bed for a median of 2 days [507]. Most cases were watery diarrhea, typical of enterotoxigenic *E. coli*, which represents the most frequent cause of travelers’ diarrhea, whereas 12% of cases, who reported fever, and 3% of cases, who reported bloody stool, were probably caused by either *Campylobacter* or *Shigella* [507]. Despite the efforts of the US military researchers of WRAIR for developing effective vaccines for *Shigella*, *Campylobacter*, and enterotoxigenic *E. coli*, no vaccine is still available, and acute diarrheal syndrome continues to be a relevant problem for deployed troops, and a damage reduction may only come from indirect hygienic measures [505]. Some of these infectious agents have been included among the biological threats of category B [57]. In Table 8, the military relevance for and the military contribution to non-vaccine-preventable diseases is summarized.

## 4. Biological Agents for Bio-Warfare/Bioterrorism Category A–B

Biological weapons (BW) achieve their target effects through infectious agents causing disease. Bioterrorism is defined as the deliberate release of viruses, bacteria, or other biologic agents used to cause illness or death in humans, animals or plants [508].

In 1999, biological agents that can be deliberately released as BW were classified by the CDC into three categories, A, B, and C, based on a series of five criteria evaluated for each agent, such as: (a) impact on public health due to the ability to produce cases of illness and death; (b) possibility of affecting large masses of the population by highly stable microorganisms, easily obtainable in large quantities, easily transported and disseminated; (c) direct person-to-person transmission capability; (d) ability to arouse alarm and panic reactions in the population, as they are perceived as highly dangerous; (e) need to take special public health measures for their control [509,510].

Category A includes the most dangerous agents, those that have the greatest potential to create public health and national security problems because they: (a) can be easily disseminated or transmitted from person to person; (b) cause high mortality with potential for major public health impact; (c) have the potential to cause panic in the population and social disruption; (d) require the adoption of special measures for public health preparedness.

Category B agents are moderately easy to disseminate; they have moderate morbidity and low mortality rates. However, they require enhancement of central diagnostic capacity and disease surveillance. Finally, Category C includes emerging microorganisms that could be engineered for mass dissemination in the future because of availability, ease of production and dissemination, and potential for high morbidity and mortality rates and major health impact [509]. The complete list of biological agents of categories A, B, and C is reported in Table 9.

Despite that history has long been disseminated of episodes highly suggestive of deliberate trials of spreading dreadful diseases through contaminated objects or cadavers of plague victims, only more recently, with the birth of bacteriology at the end of the nineteenth century, and particularly following WWI, during which Germany was suspected, but not later confirmed, of trying to disseminate anthrax, glanders, cholera and plague in the different war theaters, the attention of different countries was addressed to the issue of BW. For the first time, it was tried to contain the risk of BW use by a diplomatic initiative, such as the Geneva Protocol, in 1925. Moreover, in the same period, different countries started their scientific engagements in offensive and/or defensive BW programs [511]. During WWII, BW programs were active in Japan, France, Germany, the Soviet Union, the USA, the UK, and Canada, whereas, following WWII, only the ones in the Soviet Union, the USA, Canada, France, and the UK survived.

During the Korean War, the Soviet Union, China, and North Korea accused the USA of using BW against North Korea; however, this accusation has never been formally proven, and other episodes of assumed deliberate use of BW were registered [511]. However, in 1969 for microorganisms and 1970 for toxins, the USA officially retired from the offensive BW programs, the WHO edited in 1970 the report *Health Aspects of Chemical and Biological Weapons*, in which a comparative estimation of the different lethal and incapacitating capabilities of various BW was shown, with anthrax being the most dangerous among the tested agents, followed by tularemia. Meanwhile, the UK proposed to the UN Committee on Disarmament an evaluation of the need of prohibiting the development, production and stockpiling of BW [511]. From this initiative, the Biological Weapons Convention (BWC) was born in 1972, which entered into force in 1975; however, it exerted a weak action of containment, considering that inspections were not allowed. Its poor control activity was witnessed by the fact that the Soviet Union, one of the three co-depositary countries, represented by the Soviet Union, the USA and the UK, admitted in 1992, through the President of the Russian Federation Boris Yeltsin, that the former Soviet Union had an offensive, still active, BW program from the birth of the BWC [511]. Moreover, in 1995, the United Nations Special Commission (UNSCOM) on Iraq obtained from Iraq, after a long period of reticence, the admission that, despite being a signatory of the BWC, it had developed biological and toxin weapons after the Gulf War in 1991 [512]. Recent events in the international scenario have reminded the world of the danger of organized terrorist attacks and the possibility that some biological agents may be deliberately released to be used as weapons. Following the disaster of 11 September 2001 in the USA, the possible use of biological agents as aggressive was no longer considered a problem only for the military, but a strategy for bioterrorism that represents a risk for the entire civil community. Biotechnologies represent a means to improve the implementation of faster diagnostic methods and more effective tools of protection and therapy. However, the strong development of biotechnologies in the last three decades and the internet-based diffusion of methods also make the production and development of biological agents more accessible to non-bio-terroristic groups. The development and production of genetically modified organisms, such as antibiotic-resistant bacterial strains, represent a concern in the field of BW. A special category of human-made outbreaks of disease is the manipulation and dissemination of pathogens with the intention of disrupting societies. This may be part of government policy in biological warfare, but it is also a means used by terrorist groups or criminals [511,513]. In this field, the military has always been heavily involved both in the development of offensive programs and in the search for countermeasures to deal with these diseases. Licensed vaccines are currently available for a few threats, such as anthrax, Ebola virus, and smallpox; the last one has already been treated among the vaccine-preventable diseases, and research is underway to develop and produce vaccines for other threats, such as tularemia, botulism, plague, and Marburg virus. However, while smallpox was eradicated by a vaccine, vaccines are lacking or unsatisfactory, regarding safety and effectiveness, for most biological agents. The biological agents of category A have all been treated, whereas only some biological agents of category B and C have been treated, according to their major military contribution.

### 4.1. Anthrax

Anthrax, a potentially lethal zoonosis caused by *Bacillus anthracis*, a Gram-positive, rod-shaped, spore-forming and toxigenic bacterium, is one of the many infectious agents identified as a potential bioterrorist weapon [514], due to the stability of spores, which can persist for decades in the environment. Anthrax spores can be easily disseminated, as seen in the multiple releases via mailed letters or packages in the fall of 2001 in the USA [52,515], and it is thus considered a biological agent, category A [57]. The disease may be cutaneous, inhalational, and gastro-intestinal, with the first being milder and the other two, particularly the inhalational one, more severe. The estimation of the WHO about the consequences of air dissemination of 50 kg of anthrax spores over a densely populated city was 125,000 infections and 95,000 deaths [516].

During WWI, German scientists planned to infect livestock with anthrax [516,517]. During the interwar period, in different world countries, including France, the UK, the former Soviet Union, Canada and Japan, offensive biological weapon programs were launched. In those years, Japan, with the 731 unit led by the military physician General Shiro Ishii, was active in testing biological weapons in Manchuria. In particular, human experimentation on anthrax, which caused the death of thousands of Han Chinese people, was performed [516,518,519]. In 1943, Ishii used anthrax spores in villages south of Shanghai in retaliation for the assistance to US soldiers. In 1942 and 1943, the USA, UK, and Canada evaluated the potential use of airborne spores of *B. anthracis* as a biological weapon contaminating Gruinard Island, in Scotland. Most trials were conducted through missiles and aerials, and the effectiveness of the weapons was judged by exposing sheep, tethered downwind of the detonation point, to the airborne cloud of spores [520,521]. Only in 1986 was Gruinard Island decontaminated, and this episode represents now a well-known part of the history of BW development [522]. During the Cold War, the USA and the former Soviet Union maintained active offensive biological weapon programs [523]. However, in 1969, the USA unilaterally stopped their program [524], even if the former Soviet Union went on until 1992 with a covert offensive biological weapons program named Biopreparat, with a series of laboratories officially working for vaccine preparation, but actually on BW [525]. Anthrax spore production for military use was one of the main programs [526]. A demonstration of this research program on *B. anthracis* is testified by the tragic consequences of anthrax spore inhalation by humans in Sverdlovsk in 1979 [527]. In this episode, the accidental release of a contaminated aerosol from a facility of the former Soviet Union for the production of weaponized anthrax spores caused the death of at least 68/77 (88%) exposed people who lived along the direction of airflow carrying the contaminated aerosol. Fear of the possible use of anthrax as a BW continued in the following years, so much so that in 1991, during the Gulf War, the USA military was vaccinated [528]. Finally, weaponized anthrax spores, deliberately released by mail, were used in 2001 in the USA, inducing eleven cases (five of which were lethal) of inhalational and eleven cases of cutaneous anthrax. The analysis of these cases has enabled insights into better management of inhalational anthrax [515,516].

Along with research and experimentation on anthrax as BW, and in the same way, the search for countermeasures has been active in the military environment, both in the use of antibiotics and in the search for protective antibodies but, above all, in the development of vaccines.

During the 1930s, in the military laboratories of the Sanitary Technical Institute (STI) in Kirov, former Soviet Union, two avirulent, non-capsulated ST-1 and ST-3 strains derived from a virulent anthrax strain were isolated [529]. The protective efficacy was tested in guinea pigs [530]; however, after WWII, these strains of suspended live spores were used for vaccine development in humans as well and were recommended for being administered by scarification or subcutaneously [531]. In the period 1943–1950, the live spore vaccine was tested on 3500 volunteers; despite the adverse events not specified, it was defined as safe and immunogenic, and the protective potency of the vaccine was not specifically determined [529]. In 1951 and 1952, the live spore vaccine was administered to over 140,000 subjects living in endemic areas for anthrax and as control, a population of over 400,000 non-vaccinated matched subjects was analyzed. Fever in the first two days was observed in no more than 0.3% after both scarification and subcutaneous administration, whereas only in this last case even local symptoms were reported. Moreover, only 2.1/100,000 vaccinated subjects had anthrax versus 11.3/100,000 non-vaccinated ones, a significant difference [530]. The live spore vaccine received a license for scarification in 1953 and in 1959 for subcutaneous administration. In 1960, in the former Soviet Union, two million persons were yearly vaccinated and boostered with another dose after one year [529]. In those years, Russian military researchers tested the human anthrax live spore vaccine for aerogenic administration [532], whereas, in the period 1973–1975, repeated studies on the effectiveness of the human ST-1 vaccine were carried out using subcutaneous administration, and even in this study, the difference between vaccinated and control group was highly significant [529].

Wright and other USA military researchers at Fort Detrick, Maryland, developed early vaccines against anthrax [533,534,535]. An aluminum-adsorbed anthrax vaccine was tested in a human field trial in the 1950s, demonstrating a 92.5% reduction in disease incidence (cutaneous and inhalation cases combined), and was licensed in 1970 as AVA (anthrax vaccine adsorbed) [536,537]. Moreover, the military researchers from Fort Detrick challenged vaccinated nonhuman primates with lethal doses of anthrax spores, thus demonstrating that AVA was protective in more than 90% of cases [320,538]. It consists of an acellular vaccine containing anthrax toxin antigens and results in protective immunity after three to six doses [529]. In 1991, AVA was for the first time administered on a large scale to US military personnel deployed to Iraq during the Persian Gulf War. Afterward, in 1998, the US military started the Anthrax Vaccine Immunization Program to protect US military active duty and reserve members [539]. In seven years, from 1998 through 2005, about 5.6 million doses of AVA were administered to 1.5 million US military personnel [52]. Despite this large experience and the evidence for protection, a certain level of reactogenicity and the unusually long vaccination schedule were accompanied by a general feeling of lack of confidence in the safety of AVA. Thus, many studies have been carried out not only by the US military researchers, in order to clarify this crucial point. After evaluation of all the accumulated scientific data, the National Academy of Sciences concluded that the anthrax vaccine has an adverse-reaction profile similar to that of other adult vaccines [540]. The US military researchers even provided precious information on the need for a combined post-exposure prophylaxis of anti-microbials and vaccination, considering that anti-microbial prophylaxis alone for 1 month allowed 10–30% of cases to be infected [320], thus indicating the need of a combined treatment or alternatively 60 days of anti-microbial treatment, as carried out in the at-risk subjects in the period of the anthrax letters in the USA [517].

In 1979 in Great Britain, an anthrax vaccine precipitated (AVP) using an avirulent toxigenic, non-capsulating (pXO1+/pXO2−) 34F2 strain of *Bacillus anthracis*, originally isolated by Sterne in 1937, was licensed. It contained protective antigen (PA) and trace amounts of lethal factor (LF) and edema factor (EF), the three toxin components, thus inducing better protection than AVA, which only contains PA [541]. It was developed in Porton Down, Salisbury, a Centre for Microbiological Research dependent on the Ministry of Defense until 1979, the year in which the center was split into two separately controlled locations (one military and the other civilian). The research program for the use of airborne spores of *B. anthracis* as a BW in Gruinard Island, mentioned above, was conceived in the Centre of Porton Down.

At present, new second-generation vaccines in current research programs include recombinant live vaccines and recombinant sub-unit vaccines. Even in the development of these innovative vaccines, military laboratories are engaged [320].

Anti-anthrax hyperimmune serum was obtained independently by immunizing animals in France by the military physician Émile Marchoux and in Italy by Achille Sclavo at the end of the nineteenth century [516], a few years after Emil von Behring and Shibasaburo Kitasato in Germany developed anti-tetanus and anti-diphtheria hyper-immune sera, thus opening the era of passive immunotherapy. This serum was used in animals and even in humans, representing the only protection for a long period. After the dramatic event of the letters contaminated with anthrax spores in 2001, the US government prepared hyperimmune intravenous immunoglobulins by collecting sera of immunized at-risk workers and military personnel [255], which, although not approved by the FDA and to be used under IND status, was protective, alone and even more together with anti-microbials [528]. In 2012, a human anti-PA monoclonal antibody, raxibacumab, was approved by the FDA as prophylaxis and therapy of inhalational anthrax; in the same experimental conditions, it resulted as more protective than the intravenous polyclonal immunoglobulins [529]. In 2016, another monoclonal antibody, obiltoxaximab, was approved by FDA. Currently, further monoclonal antibodies are under study, based on the assumption that targeting different anthrax antigens may be more effective [542].

In addition to the study of vaccines, the contribution of the military is currently at the forefront of the molecular genotyping of *B. anthracis.* For some years, the Italian Army Medical Research Center has been engaged in studies on differentiating and identifying strains from different geographic areas. This could be crucial for tracing strains deliberately released in a bioterrorism attack [516,543]. In a crisis of suspected bioterrorism, standardization, speed and accuracy, together with the availability of reference typing data, are important issues, as illustrated by the 2001 anthrax letters event. Along this line, several research studies have described methods able to investigate genetic diversity between epidemic strains through single nucleotide repeat (SNR) analysis, in a fast and widely accessible way and particularly useful under field conditions [543,544,545,546].

In conclusion, pre-exposure anthrax vaccination has always been a target for military research; however, its use has been considered fundamental for selected population groups for which a calculable risk factor can be assessed. Only four of the NATO countries have included the anthrax vaccine in the vaccination schedule of selected categories of military personnel [36]. The risk, now evident, for the possible use of anthrax as BW against the civilian population highlights the importance of the contribution of military medical research in recent decades to obtain safer and more effective new anthrax vaccines [320].

### 4.2. Botulism

Botulism is a severe neurologic disease caused by neurotoxins produced by the bacterium *Clostridium botulinum*. Botulinum neurotoxins (BoNTs) are the most potent naturally occurring toxins, with as little as 50 ng of neurotoxin sufficient to cause human botulism and to represent a significant biowarfare and bioterrorism threat [547]. BoNTs are currently represented by at least seven serotypes and more than 40 subtypes. Four of the seven serotypes (A, B, C1, D, E, F, G) are pathogens for humans, in particular A, B, E, and F. These toxins may enter the body through inhalation, ingestion and wounds. Due to its extreme toxicity, easy production and dissemination, BoNT has been classified as a category A biothreat agent by the CDC [548]. New clostridial strains that produce novel neurotoxin variants are being identified with increasing frequency, which presents challenges when organizing the nomenclature surrounding these neurotoxins [549].

As a prophylactic countermeasure, from 1959 until 2011, pentavalent (ABCDE) botulinum toxoid (PBT) was available as IND. However, in 2011, the CDC stopped vaccine production considering its limited potency and high reactogenicity. At the present time, the most advanced candidates are recombinant nontoxic proteins. However, no licensed vaccines for prophylactic protection against botulism are currently available. In addition, vaccine development has been greatly complicated by the therapeutic use of BoNTs for a growing number of indications including movement disorders, hemifacial spasm, essential tremor, tics, writer’s cramp, cervical dystonia, cerebral palsy, vascular cerebral stroke, and, more recently, chronic pain, migraine, headache, and overactive bladder [550].

Since 2013, in Europe and the USA, a heptavalent F(ab)^2^ equine (ABCDEFG) botulism antitoxin was approved to treat individuals with symptoms of botulism after exposure or suspected exposure to botulinum neurotoxin. Even a trivalent (ABE) equine IgG is available, as well as a tetravalent (ABEF) equine, but only in Japan. Finally, human IgG for intravenous use (baby-BIG) is available for infant botulism. However, all these tools are in limited supply, and their administration is finely regulated [550].

Military researchers also gave their contributions for this disease. Since 1941, BoNTs have become a military issue when an American Military Attaché in Berne, Switzerland, reported that “German experts and French collaborators in the Koch Foundation laboratories near Paris” were developing “botulinum toxin in an inert carrier for dissemination by air-burst bombs” [551]. This episode convinced the US Department of Defense to focus on medical countermeasures, including vaccines, to immunize and protect personnel in the laboratories at Camp Detrick against accidental exposure to biological warfare agents [552]. The research at Camp Detrick, from 1943 to 1956, provided the foundation for the use of BoNTs as a tool for studying the trophic regulation of skeletal muscle within motor neuron terminals and, more recently, for elucidation of the intricate details of neurotransmitter release at the molecular level [552]. Indirectly, Camp Detrick researchers also played a significant role in studies that led to the use of BoNTs as a pharmaceutical product that has been approved by the FDA for treating movement disorders, autonomic dysfunctions, and other conditions. It was also identified as a critical nutritional component for improved growth of *C. botulinum* and increased production of BoNT serotype A. The purification processes developed at Camp Detrick represent the base for the production of crystalline material required for the manufacture of the toxoid vaccine. Based on the research by Camp Detrick investigators, a toxoid supply of over 1 million units was available to vaccinate about 300,000 soldiers before the operations of D-Day in Normandy [552]. In 1956, Camp Detrick was renamed Fort Detrick, and the US Army Medical Unit was established. Afterward, in 1972, it was designated the US Army Medical Research Institute of Infectious Diseases (USAMRIID), the lead military laboratory in the medical defense against biological warfare threats [552]. Recently, US military researchers from USAMRIID reported on the production of recombinant BoNT toxin domain subunits as vaccine candidates against multiple serotypes. This ciBoNT HP (catalytically inactive holoprotein vaccines) vaccine elicited a more robust neutralizing antibody response, providing better protection against a challenge with parental toxins or with dissimilar subtypes [553].

In addition, a relevant contribution must be recognized to several studies of the Italian Army Medical and Veterinary Research Center, together with USAMRIID researchers in Fort Detrick, in the field of epidemiology and genotyping of BoNT, with several studies focused on better understanding genetic variability among all of the *C. botulinum* serotypes coming from various geographic origins [549,554,555,556]. All these studies provide guidelines for botulinum neurotoxin subtype nomenclature.

### 4.3. Plague

Plague is caused by the facultative intracellular, Gram-negative, bacterial pathogen, *Yersinia pestis*. Plague is a severe, potentially lethal, disease that may manifest in three forms, bubonic, pneumonic and septicemic plague. It is a zoonosis, with rodents as reservoirs and fleas as vectors. Three great plague pandemics, resulting in nearly 200 million deaths in human history, have made *Y. pestis* one of the most virulent human pathogens. Even if the plague is often classified as a problem of the past, it remains a current threat in many parts of the world, particularly in Africa [557,558]. In addition, the possible use of *Y. pestis* as a bioterrorist weapon is a serious threat due to its pathogenicity, easy dissemination, and human-to-human transmission. For this reason, it has been classified by the CDC as Category A biological agent, also considering that different strains of *Y. pestis* showing resistance to the currently available antibiotics have been identified in Madagascar [559]. The earliest recorded use of *Y. pestis* as a biological weapon dates back to the 14th century, when a Tatar army, in the attempt to conquer Caffa, in Crimea, reportedly catapulted victims of plague over the city walls [560]. The Black Death, as the plague was named, which swept through Europe, the Near East, and North Africa in the mid-14th century, was probably the greatest public health disaster in recorded history and one of the most dramatic examples ever of provoked pandemic. Caffa should be recognized as the site of the most spectacular incident of biological warfare ever, with the Black Death as its disastrous consequence [561].

The first plague vaccine, developed at the end of the 19th century, consisted of killed, whole-cell, *Y. pestis* [562]. An immunogenic and somewhat less-reactogenic licensed vaccine (USP) containing a formalin-killed, highly virulent 195/P strain of *Y. pestis*, was effective in preventing or ameliorating bubonic disease, as seen by the low incidence of plague cases in US military personnel serving in Vietnam [563,564]. However, in vivo data on experimental animals suggested that this vaccine did not offer optimal protection against pneumonic plague [562]. Live attenuated *Y. pestis* vaccines, such as the EV strain (a virulent *Y. pestis*, derived from a patient identified as EV, and attenuated in the 1920s by serial passages), have been used in several countries for decades [565]. It was considered more immunogenic in the animals but more reactogenic than the inactivated one. The reactogenicity of the inactivated plague vaccine was evaluated to be lower than that of the whole-cell inactivated typhoid vaccine that was used in 1940 [320]. The live vaccines seemed to be able to protect against pneumonic and bubonic plague and induced high antibody titers, but unfortunately, they could have severe side effects and only induced short-lived protection that required annual boosters [566,567].

In analogy with all potential BWs, the military has invested resources in plague research. At the end of the last century, the USAMRIID developed a recombinant fusion protein, F1-V, comprising full-length capsular fraction 1 (F1) and low calcium response virulence protein (V) antigens [568,569,570,571]. This vaccine has long shown promise as a vaccine candidate against both pneumonic and bubonic plague in rodents [568,569,570]. It was recently shown that F1-V adjuvanted with aluminum hydroxide (alum) using an IM/SC prime-boost regimen, provided complete protection against intranasal challenge with virulent *Y. pestis* CO92 in mice, guinea pigs, and macaques [572]. That F1 and V may be key molecules for active and passive immunization is also proven by the protective effect elicited not only by a recombinant fusion protein F1-V acting as a vaccine but even by anti-F1 and V human monoclonal antibodies able to protect mice from challenge with *Y. pestis* [573]. However, currently, no FDA-licensed vaccine [565], nor polyclonal/monoclonal antibodies for human use against plague [574] are available. In addition, a relevant contribution of the Italian Army Medical Research Center was in the field of strain differentiation of *Y. pestis*, a difficult challenge for these microorganisms considering their high intraspecies genome homogeneity [545,575]. Fast strain identification and comparison with known genotypes may be crucial for naturally occurring outbreaks versus bioterrorist events discrimination. In this regard, these studies were focused on assessing the inter-laboratory reproducibility of in-house developed real-time PCR assays for the identification of *Y. pestis* [576].

### 4.4. Tularemia

Tularemia is a zoonosis caused by *Francisella* (formerly *Pasteurella*) *tularensis*, a facultative intracellular, Gram-negative, coccobacillus, which may manifest in different clinical forms, including the glandular and ulcero-glandular, oculo-glandular, oropharyngeal, typhoidal, and pneumonic. *F. tularensis* is present in some animals, in particular rabbits, and may be transmitted to humans by drinking water contaminated by infected animals, drinking juice from contaminated fruits, or by aerosols and dust, or finally by the bite of insects, such as ticks and mosquitoes. It has been included among biological agents, category A [57]. There are two types of *F. tularensis*, one more severe and contagious (10 cells represent the LD50 for rabbits and 50% infectious dose for humans), present in North America, and the other less virulent and contagious (10 million cells represent the LD50 for rabbits), present in Europe and the former Soviet Union [320,577]. A great outbreak of tularemia, involving 100,000 Soviet troops, occurred in Stalingrad during WWII. It started among the Nazi military and spread to the Soviet military. The large majority (over 95%) were pneumonic forms of tularemia. This unusual clinical presentation raised suspicions that this outbreak was unnatural and deliberate. Whichever the origin of the Stalingrad outbreak, it remains a military interest for tularemia, as even witnessed in the cases observed in Kosovo during the recent Balkan war [320].

The causative bacterium of tularemia, *F. tularensis*, was isolated in 1919 [578]. After the first few attempts with killed whole-cell vaccines, known as Foshay vaccines [579], which were not effective [580], a vaccine against tularemia was developed in the Soviet Union in the 1940s. It was a live attenuated vaccine, and millions of people living in endemic areas underwent immunization [581]. Afterward, in 1956, a mixture of two attenuated strains (155 and 15) of *F. tularensis* was brought from the Russian Institute of Epidemiology and Microbiology (Gamaleia Institute, Moscow, Russia) to Fort Detrick, USA. From an ampoule of this product, a vaccine was produced, which resulted in protection for mice and guinea pigs [582].

Studies at Fort Detrick continued into the 1960s; the strain was tested for safety and efficacy in human volunteers and introduced as the *F. tularensis* live vaccine strain (LVS) [583,584,585,586]. This vaccine has been used since the mid-1960s and is associated with a significant decline in the rate of laboratory-acquired infections at Fort Detrick [587]. However, considering the incomplete knowledge of the mutations in LVS and its residual virulence, the vaccine remains under IND status and is administered only under protocol and with written informed consent.

Currently, in addition to military researchers, there are several groups involved in developing new-generation vaccines against tularemia, both by applying new technologies and by using different routes of administration [588]. In particular, attempts are focused on creating new live attenuated mutants [589,590], novel subunit vaccines [591] or glycoconjugate of lipopolysaccharide O antigen of *F. tularensis* [592]. No tularemia vaccine is licensed yet, nor polyclonal/monoclonal antibodies for prophylaxis/therapy, despite that polyclonal antibodies have been demonstrated to be protective in mice [593], whereas monoclonal antibodies only result in partial protection [594].

### 4.5. Filoviruses

The *Filoviridae* family is composed of enveloped RNA viruses with non-segmented, negative-sense genomes. Filoviruses are divided into three serologically distinct genera: *Ebolavirus*, *Marburgvirus*, and *Cuevavirus* [595]. They are among the most dangerous pathogens in the world, cause viral hemorrhagic fever, with case-fatality rates of up to 90% and are included in Category A of biological agents that could be used as weapons.

Before the 2014 West African Ebola virus outbreak, several filovirus vaccines had been tested in small rodent models, but few candidates had moved into advanced development [596]. Even in this, case military research has always been on the front line, and in 1980, the first candidate vaccine on the basis of a heat- or formalin-inactivated Ebola virus was tested at USAMRIID, Fort Detrick, on guinea pigs [597]. In the following years, Warfield and other military researchers worked on virus inactivation with preservation of antigenic and structural integrity by a photoinducible alkylating agent [598]. The inactivated Ebola virus vaccine protected 80% of vaccinated mice from lethal disease. Similar levels of protection were measured with inactivated Marburg virus studies. Virus-like particles (VLPs) of Marburg virus vaccination completely protected guinea pigs [599]. Despite the high efficacy in these animal models, the vaccine did not provide the proper level of protection to primates from lethal infection, both with Ebola and Marburg viruses [600,601].

More recently, several studies conducted always at USAMRIID investigated the possibility of using replicon-containing VLP filovirus vaccines [602,603,604,605,606,607,608]. Replicon-containing VLPs are generated by using a Venezuelan equine encephalitis virus (VEEV) vector to produce replication-incompetent particles capable of entering a host cell. For both Ebola and Marburg viruses, the antigen encoded is typically glycoprotein (GP), due to its capacity of inducing protective antibody responses [604,606]. Even though it is a promising topic, the protective efficacy of these vectored vaccines remains to be tested.

Another attractive strategy was the use of recombinant adenovirus vectors. Replication-deficient adenovirus vectors are highly immunogenic and can generate robust B and T cell responses to viral antigens [609]. Among the studies in this area, a pan-filovirus vaccine was tested, among others, by Swenson et al. at USAMRIID by using multiple adenovirus constructs to express genes encoding Ebola GP, Ebola nucleoproteins, Marburg nucleoproteins, and three Marburg GP combined into a single vaccine [610]. This approach is believed to provide widespread protection from multiple filovirus species [611].

DNA vaccine platforms have seen significant contributions from military researchers, such as the studies of Grant-Klein et al. on the evaluation of the ability of codon-optimized DNA vaccines against the GP of Ebola and Marburg delivered by electroporation, individually or as a mixture [612]. Further, they have also investigated different routes of vaccine delivery and various DNA doses to optimize protection and to elicit the most robust antibody responses. Multiple clinical trials have been subsequently conducted to evaluate the safety and immunogenicity of filovirus DNA vaccines, also with the contribution of US military researchers of WRAIR, as in the case of the first Ebola or Marburg vaccine trials performed in Africa, showing that, given separately or together, both vaccines were well tolerated, safe, and elicited antigen-specific humoral and cellular immune responses, suggesting limited immune interference [613].

Afterward, another relevant contribution of military researchers has been provided with the approach by Filovirus VLPs, non-replicating vaccines generated by co-expression of the GP and structural matrix protein VP40 in mammalian cells or insect cells [614,615]. VLPs containing filovirus GP have successfully been used to vaccinate rodents, even in the absence of adjuvants [616]. Moreover, it was demonstrated that VLPs activated cellular immunity and that CD8+ T-cells are required for protection [617]. The protective efficacy of VLPs seen in rodent models has also been observed in non-human primates by Warfield and colleagues [618]. These studies showed encouraging results versus both Ebola and Marburg viruses [619,620]. The multitude of advantages afforded by VLPs make them a promising filovirus vaccine platform.

USAMRIID researchers also investigated the possibility of using cytomegalovirus as a vaccine vector due to its strict species specificity and continuous replication within the host [621]. Likewise, rabies virus vectors have been explored as a vaccine platform against both Ebola and Marburg viruses by several groups of investigators, including the military [622].

Finally, the most encouraging candidate for a filovirus vaccine is currently represented by recombinant vesicular stomatitis virus (rVSV). One of the proteins encoded by rVSV is the G protein that, expressed on the surface of the virion, enables viral entry [623]. At the beginning of the 2000s, Garbutt et al. produced the first rVSV-expressing Ebola GP, Marburg GP o Lassa GP [624]. Afterward, pre-clinical studies have established that rVSV filovirus vaccines can rapidly induce protective immune responses in nonhuman primates [625,626]. After the 2014 Ebola outbreak, there has been a notable acceleration in clinical trials in which there was a relevant role by USAMRIID researchers. Even military researchers in the Russian Federation provided an important contribution to this vaccine approach [627]. The 2014–2016 Ebola outbreak in West Africa is a paradigmatic example of military management of the epidemic both at the local level and on the part of the deployed military by different western countries [242].

In all these studies, numerous partners from the public and private sectors have combined efforts and resources to develop a Zaire ebolavirus (EBOV) vaccine candidate (rVSVΔG-ZEBOV-GP) such that the rVSVΔG-ZEBOV-GP vaccine was approved as ERVEBO^®^ by the European Medicines Authority (EMA) and by the FDA in December 2019 after five years of development [628]. No licensed vaccine for the Marburg virus is available yet.

Ebola virus disease has been faced with plasma from convalescent people. In 1995, during the Ebola outbreak in Kikwit, Democratic Republic of Congo, this treatment seemed promising, considering that seven out of the eight patients who received convalescent plasma survived, a mortality of 12.5% versus 80% of the not treated patients [629]. A non-randomized study did not confirm this result and did not find a significant improvement in survival in patients who received convalescent plasma, whose anti-Ebola neutralizing titer was unknown [630]. However, when the anti-Ebola antibodies were measured, it was observed that the infusion of plasma with a high antibody titer was accompanied by viral load reduction [631], thus confirming the usefulness of passive immunotherapy in Ebola disease. The military researchers of USAMRIID, who have collaborated in this study, even participated in another study on the preparation of hyperimmune intravenous immunoglobulins, which showed 5–6-fold increased potency compared to the pool of convalescent plasma and increased survivability of infected mice, when administered concurrently or 2 days after infection, thus hypothesizing that it may become a relevant tool for post-exposure prophylaxis [632]. Recently, at the end of 2020, two monoclonal antibodies, INMAZEB^®^ (consisting of three anti-Ebola glycoprotein monoclonal antibodies) [633] and EBANGATM (obtained by a convalescent patient and directed against the Ebola virus glycoprotein [634]) have been licensed by the FDA and, more recently, by the EMA.

### 4.6. Arenaviruses

Arenaviruses are divided into old world and new world viruses. All the members belonging to the Arenavirus genus are linked to the progenitor lymphocytic choriomeningitis virus (LCMV), which was discovered in 1933 and in 1964, and the similarities between the chronic disease caused in mice from LCMV and the one caused in hamsters from Machupo virus, a member of new world viruses, were noted [635]. They are named Arenavirus from the Latin arena, meaning sand [636], the most important member of the old world is the Lassa virus, discovered in Nigeria in 1969, which is the most prevalent rodent-borne arenavirus circulating in West Africa. It is estimated that it is responsible for 300,000 infections per year and 5000 deaths [636]. The virus is present in some rodents and may be transmitted to humans by aerosols of the excreta. The mortality may be as high as 30%, and the same severity, and even more, is observed in the new world Arenaviruses, such as Junin, Machupo, and Sabia. For these characteristics of easy dissemination and high severity, Arenaviruses are also included among Category A biological agents. The contribution of military researchers must be identified, as we have seen thus far, in the studies at USAMRIID in Fort Detrick, especially on the pathogenetic mechanisms of the infection. From this point of view, a relevant work, born from the collaboration with other European scientists, has allowed for clarification of the mechanisms of entry of the virus into infected cells. Specifically, when the Lassa virus latches on to a receptor on the cell surface, it is first transported to a lysosome inside the cell, from which it can escape by hooking onto a receptor called LAMP1. This work, performed at USAMRIID using authentic Lassa viruses, was critical for validating the role of LAMP1 in Lassa virus infection, opening the door for the development of directed therapies [637]. Currently, no licensed vaccine for the Lassa virus is available nor monoclonal antibodies for human use. Only for the Junin virus is there a vaccine that has been registered in Argentina, where the virus is endemic.

### 4.7. Brucellosis

Brucellosis is a zoonosis caused by small Gram-negative coccobacilli of the genus *Brucella*. There are seven species of *Brucella*, of which four (*B. abortus*, *B. melitensis*, *B. suis*, and *B. canis*) are pathogenic to humans, with approximately 500,000 new human cases annually reported [638]. *Brucella* mainly infects cattle, swine, goats, sheep, and dogs, and the disease may be transmitted to humans by eating or drinking unpasteurized contaminated cheese or milk or by inhaling airborne agents. Person-to-person transmission is rare. Considering its relative prolonged capacity for incapacitation, even in the presence of a low mortality rate (5% of untreated cases), and given its significant contagiousness by inhalation (10–100 bacteria sufficient to determine human disease), *Brucella* is considered a potential biological weapon, so much so that it is included in Category B of CDC classification [320]. Brucellosis is found globally and manifests with flu-like symptoms, including fever, weakness, malaise, and weight loss. In this case, the contribution of the military arises from the name itself of the disease. In 1887, the British military physician David Bruce was able to isolate the germ, by him denominated *Micrococcus melitensis*, but later re-denominated *Brucella melitensis* in his honor, from the spleens of patients who died in the Malta isle due to a febrile illness, also known as Malta, Mediterranean or Undulant Fever [639]. Due to the major role played by the Royal Army Medical Corps in clarifying the nature of the disease and its way of transmission, through the contaminated milk of the goats, thus leading to its prevention, the disease was also nicknamed the “Corps Disease” [640]. However, brucellosis remained a constant threat in more recent times, in the Mediterranean region and the Middle East during World War II and in the Middle East today. In addition, considering its potential relevance as a weapon, in the 1950s, brucellosis has been included in various biological offensive programs in the USA, in South Africa, and in the former Soviet Union [524,641,642].

The contribution of military research also concerns vaccine development. The excellent results obtained in the veterinary field allowed for dramatically reduced cases of human brucellosis. Current veterinary vaccines are based on live Rev1 *B. melitensis* strain and attenuated *B. abortus* strain 19. From the latter derived the preparations utilized for human vaccines used in the past in some nations [643]. However, these human preparations showed a high incidence of cases of clinical disease and adverse reactions, as reported both in the USA and in the former Soviet Union, where this vaccination is still widely used [644]. More recently, as part of the Brucella Vaccine Development program, WRAIR military researchers have evaluated the effect as an adjuvant of a *Neisseria Meningitidis* outer membrane protein for an intranasal *B. melitensis* immunization in mice and guinea pigs [645]. In China, military research developed a rapid and highly efficient method for the identification of candidate antigens, using a combination of immunoproteomics with immunization and bacterial challenge [646]. At present, no licensed human vaccine against brucellosis is available in the western world.

### 4.8. Q Fever

Q fever is a zoonotic disease due to Gram-negative coccobacillus *Coxiella burnetii*, a rickettsia-like organism capable of prolonged survival under harsh environmental conditions. Disease in humans is essentially caused by inhalation of dust from infected animals such as cattle, goats and sheep. The most frequent symptoms are flu-like syndrome, pneumonia, and hepatitis [647]. Q fever has military relevance for both the risk of natural infection in military deployed abroad and for its potential use as a bioterrorism agent, considering that it is included in Category B of CDC classification. Thousands of cases of Q fever have been seen in military personnel since the disease was first reported in 1937 [648,649]. However, it is believed that during the American Civil War, 1,765,000 cases of pneumonia and 45,000 deaths among the unionist troops might have occurred, including cases of Q fever, and the data are probably underestimated. This phenomenon was certainly much more serious among the Confederates, but there are no reliable data on the matter [650]. After being identified, it was first believed to be limited to Australia, hence the definition of Queensland fever, but afterward, it was found that the distribution was worldwide, and the retrospective analysis of military researchers established that many British and US soldiers were affected by this disease during WWII in Greece and Italy [651,652]. After WWII, distinct outbreaks among US troops have been reported in Libya, in the USA (California and Texas) in the 1950s and, more recently, during Operation Desert Storm, Restore Hope, Enduring Freedom, and Iraqi Freedom [653,654,655,656]. In addition, an outbreak was reported among the Czech military in Bosnia in 1997 [657].

The military contribution was relevant after the identification of the germ, both in better defining the clinical aspects and the development of dedicated vaccines. Particularly noteworthy, were the studies of US Col. AS Benenson concerning the development of the first effective Q fever vaccine [658,659]. This vaccine, based on whole-cell, formaldehyde-inactivated, ether-extracted *C. burnetii* with 10% egg yolk sac, was generally well tolerated, even though nonnegligible severe adverse reactions could be observed [660]. However, it proved useful for protection versus laboratory-acquired infections or for military personnel deployed to high-risk areas. Thereafter, this vaccine had been available at the US Army Medical Research and Materiel Command, Fort Detrick, under IND protocol, but due to its adverse side effects, it was (and is) not commercially licensed in the USA. Currently, the only licensed vaccine has been developed in Australia, Q-Vax^®^ (Seqirus UK Limited, Maidenhead, UK) in 1989, and is used only in Australia in high-risk groups [661]. Even in this case, the frequent hypersensitivity reactions to the vaccine have hindered the license of Q-Vax^®^ beyond Australia. Consequently, the research for a less reactogenic vaccine is still in progress in military institutions, specifically at the USAMRIID [662,663].

### 4.9. New World Viral Encephalitis

Viral encephalitis poses a significant threat to military personnel employed in areas at risk for these infectious agents. In addition, many of these agents are classified as Category B (*Alphaviruses*) and Category C (tick-borne encephalitis, West Nile fever) biological threat agents by the CDC and have reportedly been developed as biological weapons in the past [664].

*Alphavirus* genera belonging to the *Togaviridae* family are positive-sense RNA viruses, usually transmitted by a mosquito vector. Infection may run asymptomatic but is usually severely debilitating, with fever, malaise, headache, with risk of encephalitis and is occasionally fatal, as in the case of the new world Alphaviruses Venezuelan, Eastern and Western equine encephalitis viruses (VEEv, EEEv, WEEv) [665].

VEEv in humans may run asymptomatic or cause mild symptoms, including fever, headache, and, occasionally, convulsions and disorientation [257]. It was first isolated in Venezuela, in 1938, and occurs mainly in South and Central America. In addition to mosquitoes, the virus is also transmitted in laboratories via aerosol [666].

There are currently no FDA-approved vaccines for VEEv, but two IND vaccines are available for at-risk laboratory personnel, TC-83 (live attenuated virus developed in 1961) and C-84 (a formalin-inactivated version of the TC-83 strain); however, neither vaccine could protect experimental animals against inhalation infection [667,668,669,670]. Other engineered live attenuated vaccines have been investigated in mice and non-human primates. A candidate vaccine is the V3526, even if it protects against only a few genotypes of VEEv [671]. For these reasons, the research has always continued over the years with a different approach, as DNA vaccines have proven particularly effective at eliciting protective immune responses in animal models [672,673]. Laboratory workers at USAMRIID are currently immunized with live (TC-83) and killed (C-84) vaccines. Moreover, military studies have increased knowledge on attenuation of VEEv, with a novel antigen expression system based on VEEv genes, usable for other viruses [674]. Finally, military research is also at the forefront of the study on the use of monoclonal antibodies, currently on animal models, as prophylactic and therapeutic approaches [257]. Particularly, at USAMRIID, it has been demonstrated that post-exposure administration of monoclonal antibodies protected macaques from the development of severe VEE disease even when administered 48 h following aerosol exposure [675].

EEEv is rare; however, it is a serious disease, with 50% to 75% mortality and severe neurological sequelae in survivors [676]. Most of the cases occur in North, Central, and South America, and in the Caribbean. The military had paid attention to this virus as well, and in the 1970s, they investigated and produced a formalin-inactivated vaccine to protect laboratory personnel working with EEEv [677,678]. This vaccine (TSI-GSD 104) is currently available as a US Army IND and has been administered to nearly 1000 workers [679]. Moreover, the military research originated a chimeric vaccine containing genes of EEEv and WEEv [676] as well as a field-deployable RT-PCR assay [680].

WEEv is another alphavirus, and its basic description is the same as that for VEEv and EEEv. It has been isolated for the first time in California in 1930, and outbreaks still occur in the western USA [257]. In humans, it can present with fever, typical encephalitis and abnormal mental status, including focal neurological signs [681]. A formalin-inactivated vaccine is available as IND and is administered to the laboratory workers who are exposed to the virus. This vaccine is well tolerated, considering that <5% of recipients report minor symptoms [257]. Even in this case, military research contributed to various aspects, from pathogenetic mechanisms to the effects of the virus on different animal models [682,683,684,685]. In addition, a repertoire of mouse monoclonal antibodies against WEEv has been characterized and shows promise for immunodetection and immunotherapy [686]. In the early 1990s, a formalin-inactivated vaccine (CM 4884), obtained from an attenuated strain, was developed by the US Army and made available under IND status [687]. The vaccine was administered to laboratory workers at risk of contracting WEEv.

Finally, other military studies have investigated killed alphavirus vaccines and genetically engineered, live attenuated virus vaccines [688,689,690]. In Table 10, the military interest for and the military contribution to the biological agents, category A, and some biological agents, category B, are reported.

## 5. Aeromedical Evacuation of Patients with Highly Contagious, Severe Infectious Diseases

A particular mention must be reserved for aeromedical evacuation of patients with highly contagious, severe infectious diseases. Evacuation of such patients is relevant to military contingency operations because troops may be placed at risk for hemorrhagic fevers and other infections during deployment to tropical environments or by adversaries’ use of biological warfare agents [691]. Historically, aeromedical evacuations with high-level biocontainment systems have been conducted by a limited number of military organizations. Among these are the Deployable Air Isolator Team (DAIT) of the British Royal Air Force [692], the Aeromedical Isolation Team of the USAMRIID [693,694], and the Aeromedical Isolation Team of the Italian Air Force [695]. The Ebola Virus Disease (EVD) outbreak of 2014–2016 provided the largest experience for the aeromedical evacuation of patients with viral hemorrhagic fevers [696]. Several healthcare workers who acquired the infection while caring for patients in West Africa were transported to EVD treatment facilities in the USA and Europe. While several of these patients were transported by military services, as in the case of the Italian Air Force [695,697], civilian aircraft were contracted to provide evacuation services by government agencies [698]; at least 10 nations conducted aeromedical evacuations with high-level containment systems for at least 33 patients with EVD.

The aeromedical evacuation of COVID-19 patients during the pandemic was a new and unpredictable scenario. The new tasks were to move patients with acute respiratory distress syndrome that caused an overflow of intensive care units regionally and to transport home, in a timely fashion, civilians or military personnel with suspected or confirmed COVID-19 cases, from other countries or in deployed military units. In all these circumstances, the availability of military organizations with a background in the transport of high infectious patients in the early stages enabled effective response to the many tasks received.

Aeromedical transport solutions for highly infectious patients to prevent transmission are available as open and closed systems. In open systems, the medical crew, wearing full personal protection equipment (PPE), and the patients are in a negative pressure isolation unit (container) with an airlock. Usually, open systems allow for easier access to the patient and thus enable safer care for critically ill patients; however, the prohibitively high costs of open systems due to the need for a larger airframe are a limiting factor. Since the EVD epidemic, multiple organizations have developed isolated systems enabling simultaneous isolation and care of multiple patients; these include the US Department of State Containerized Bio-Containment System [699] and the US Department of Defense Transport Isolation System [700]. In closed systems, the patient is isolated in a smaller negative pressure isolation unit, with the medical crew outside that chamber. Patients are treated by facilities that are fixed to several holes in the wall of the isolation unit. The main limitations of closed systems are limited access to the patient and reduced manual dexterity when delivering care through porthole gloves [698]. The Italian Air Force Isolation Unit employed closed systems during the pandemic, such as the Aircraft Transit Isolator (ATI) and the IsoArk N-36 on their fixed and rotary wing assets (C-130J, Boeing KC-767A, C-27J, HH-101A) (Figure 2). In other countries, private air ambulance providers have carried out many transfers of moderately to severely ill patients, by other cost-effective closed isolation system solutions [701,702].

Unfortunately, these open and closed isolators are limited in both number and capability and require specially trained teams of medical personnel. To overcome these limitations, another approach is the restricted flight of a cohort with specific communicable diseases to reduce the risk of patient-to-patient transmission. The cabin of the aircraft is divided into a clean area for the crew and a dirty area for the aeromedical crew and patients, adequately separated with plastic partitions, neutral zones and pressure gradients. The Italian Air Force applied these procedures several times in February 2020 to repatriate Italian citizens from China, the UK and Japan and in the following months for the repatriation of military personnel from deployed units abroad. In March 2020, the UK government transferred a significant number of patients suffering from COVID-19 in multiple missions using a hybrid military–civilian model [703]. The same procedures were applied in France and Germany on military aircraft with Patient Transport Units for intensive care, used for strategic evacuation to transport critically ill patients across Europe [704].

Isolation of infectious patients during aeromedical evacuation is a complex process with numerous requirements, involving highly trained personnel, as well as specialized equipment and validated infection control processes, and requires a careful and individualized risk−benefit analysis for each patient before transport. The pandemic experience is adding a large amount of information about the processes, procedures, and equipment available and should promote the development of standards and consensus guidelines to transfer such capabilities to other organizations that might have a current need and to respond to future crises.

## 6. Discussion

History shows that infectious diseases influenced the outcome of battles and wars more heavily than weapons. From sieges of the walled cities in antiquities to the bombardments of cities in the modern era, the involvement of civilians in battles has become increasingly frequent as well as their susceptibility to infections. Paradigmatic in this regard is the situation of infectious diseases reported in Ukraine after 2014, when Russian special forces occupied part of the southeast area of Donbas and annexed the Crimea peninsula, thus activating a state of local conflict on the southeastern border of Ukraine. From 2014 to 2017, an outbreak of poliomyelitis was observed, after Ukraine was certified polio-free in 2002. In the period 1 January–5 November 2019, 56,802 cases of measles were reported to the WHO, tuberculosis was rising, and COVID-19 showed a case-fatality rate of 2% [705]. Thus, the military has always been forced to face not only the visible enemy but also the invisible agents of infectious diseases, which have frequently influenced the outcome of battles and wars more heavily than the strategic capacity. The possibility to neutralize the consequences of the spreading of microorganisms became a military strategic capacity particularly important after the birth of bacteriology—in Germany with Robert Koch and in France with Louis Pasteur. Military physicians from Germany, the UK, and France were the winners of the first, the second, and the seventh Nobel Prize for Physiology or Medicine and allowed passive immunotherapy to be a reality for defense against dreadful diseases such as tetanus and diphtheria and the discovery of etiology and the vector of an ancient plague for humankind, such as malaria, allowing for its containment. Other military researchers developed an effective typhoid vaccine and identified new pathogens, such as leishmaniasis. Meanwhile in the USA, Major Walter Reed contributed to the identification of the yellow fever vector, thus paving the way for its containment, with the effective preventive measures designed by Major William Gorgas. The military from the countries with a strong attitude to be present in different war theaters worldwide had the need to cultivate and develop the then-rising bacteriology to protect the armies, and their research and observations have represented useful conquests for all humankind. The two WWs and the other wars of the twentieth century have represented the opportunity to test in the field the scientific acquisitions in infectious diseases and to speed their application [706], with a growing role of the US military medical researchers, considering that the USA has organized a stable military network for the study of infectious diseases, with prestigious institutes in the USA and overseas; one of the research Institutes is entitled to Walter Reed, the WRAIR, which was created in 1893 and without interruptions has continued its scientific activity, by launching specific organic programs, such as the HIV/AIDS program for the military, which has obtained many different and useful observations for the advancement of science in HIV, even for the general population [448]. Moreover, the financial resources made available to the US Department of Defense for medical research is considerable; recently, it has been USD 2 billion per year [706]. The military has historically been engaged in the search and early application of preventive tools, such as vaccines. This has led to relevant epidemiological results, as in the case of typhoid fever, meningococcal meningitis, tetanus, measles, and rubella. In other conditions, instead, the adoption (or lack of adoption) of the available new vaccinations has not been in line with the traditional timely military intervention, as in the case of meningococcal meningitis B, HPV, pneumococcus, adenovirus and influenza. In the last three cases, a responsibility may be attributed to the vaccines themselves, which may not be considered either fully protective or easily obtainable (this is the case for the adenovirus vaccine, which has only been approved by the FDA for the US military, and even for leptospirosis and dengue vaccines), whereas for meningitis B and HPV, the reason may be due to scarce awareness of the risk of infection.

However, the situation of infectious diseases is highly moving and is influenced by the variability of socio-environmental conditions, including climate change [707], urbanization and deforestation, which induced the emergence of new pathogens for humans and the re-emergence of pathogens that seemed to have disappeared. Between the last decades of the last century and the beginning of this century, at least 30 new, potentially lethal, pathogens have been described [222,708]. Such a moving situation induced the WHO, in the first half of the 1990s, to launch a project on the emerging and re-emerging infectious diseases and to propose an organic and stable collaboration with the global military health services for surveillance and control of infectious diseases, by creating inside the WHO, a new formal position for a military physician liaison officer. This position was first covered, in the period 1995–2000, by one of the authors of this review (R.D.), and afterward, by US military physicians up to 2012 [709]. The first military liaison officer could demonstrate, by a global survey to which 76 world countries replied, that the collaboration between civilian and military health services was already a precious and productive reality in some places, and that in some developing countries, military health infrastructures could replace the lacking civilian ones [7]. The subsequent activity of the US military physicians consolidated this civil–military collaboration, by strengthening collaborative projects with developing countries, through the network of the US military research laboratories in developing countries, such as in Peru and Thailand [709]. However, this collaboration is becoming closer and closer, making the fields of collaboration larger, from research and development to disasters and health emergencies management [710]. In this context, aeromedical evacuation of patients with highly infectious diseases, a medical, technical, and organizational challenge with significant risks for the safety of the patient, the crew, and the population, was an almost exclusively military activity until the Ebola outbreak of 2014–2016 [696,711,712,713]. The heritage of engagement of the military in the fight against infectious diseases, even before the birth of bacteriology [714], in particular for some dreadful diseases, such as influenza [190], and currently with the COVID-19 pandemic [242], legitimizes a sort of “militarization” of the management of infectious health emergencies, which have the potential for deeply undermining societal functioning and stability [708], as observed during the recent Ebola outbreak in West Africa in 2014–2016, the Zika virus outbreak in Brazil, or the COVID-19 pandemic worldwide [242]. This sort of “militarization” of the fight against infectious diseases, which is based on historical military involvement and a series of characteristics the military has, including quick mobilization, logistic organization and the possibility to impose lockdowns by law enforcement, needed during the pandemic, is not agreed upon by everybody [190,242]. However, this international engagement of the military in humanitarian health initiatives has been favored by different governments in the context of global health diplomacy, a sort of soft power able to achieve relevant international security results with lesser expenses compared with the agencies officially dedicated to military–military or civilian–military international collaborations. Military physicians should specifically be formed for this innovative duty, and models of training have already been proposed [715,716]. Moreover, terrorism and bioterrorism are further dramatic phenomena that stimulate a closer and closer military–civilian collaboration, in order to set up an effective response [717].

Another aspect that should be here underlined is that the research and development of new effective tools for the prevention and control of infectious diseases representing a threat to the military may be assimilated into the research and development of weapons and armaments. In this context, the research and development of vaccines and/or polyclonal/monoclonal antibodies against neglected infectious diseases, which is considered not cost-effective by the civilian industry for lack of market, may be carried out or financially supported by the military, which is engaged in this activity for strategic defensive objectives, regardless of market logic. However, the costs for developing a vaccine are, with today’s rules, prohibitive even for the military of affluent countries, such as for the USA [513], thus inducing to think that such efforts may only be faced through international collaboration among the militaries of allied countries, and a strengthened civil–military collaboration. In conclusion, a close civil–military collaboration for health promotion is of reciprocal interest, particularly in the field of infectious diseases, irrespective of immediate or delayed danger caused by the disease. An example of this is, among others, HPV vaccination, which may prevent a non-acute disease, such as cancer; however, the extension of vaccination to male subjects, as observed in the military, may be a relevant measure of public health.

## 7. Conclusions

In conclusion, war and infectious diseases have a reciprocal influence, because war creates socio-environmental conditions favoring the spreading of infectious diseases, which conversely may heavily influence the outcome of wars, by drastically reducing the operational readiness of the military. Paradigmatic of the influence of conflicts and social disruption on infectious diseases is the complex and difficult story of polio eradication. Conflicts are frequently accompanied by a re-emergence of the eradicated polio, as in Ukraine, which was declared polio-free in 2002, and a polio outbreak was registered following the conflict with the Russian Federation in 2014 [705]. The same situation was observed in Syria in 2013, when polio, which had been eradicated 18 years before, re-emerged as a consequence of the civil war [140,718]. Even in Afghanistan, Pakistan, Nigeria, Somalia and Kenia, the persistence or re-emergence of polio is generated by local conflicts and socio-environmental, economic and health disruption [719]. An excellent historical review documenting the direct relationship between conflicts, wars and social disruption with re-emerging infectious diseases has recently been published [720]. The military health services have historically been forced to effectively combat infectious diseases, and this specific expertise should be put in common with the civilian counterpart to maximize the efforts and the results in the control of infectious diseases. The quick mobilization, the logistic organization and the power of imposing restrictions by law enforcement are characteristics of the military worldwide, which have been recognized and used by the governments in recent outbreaks and pandemics, and which have the potential for deeply undermining societal functioning and stability, inside the country and in international military–military and civilian–military collaboration, in a sort of global health diplomacy.

## Figures and Tables

**Figure 1 biomedicines-10-02050-f001:**
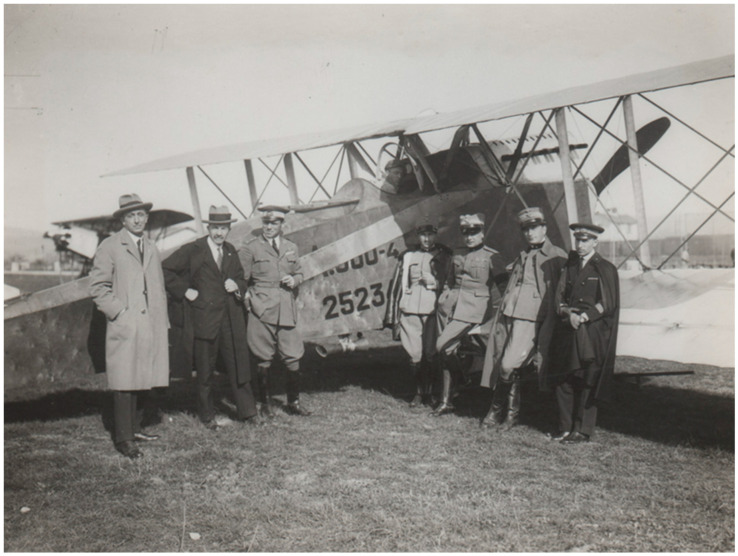
In the years between the 19th and 20th centuries, military and civilian health were collaborating side by side in the fight against the vector of malaria. In this picture, military and civilian Italian Health Authorities witness the diffusion by airplane of Paris green (the most widely used insecticide in that period) for malaria vector control in the countryside around Rome in 1928. (Courtesy of the Archive of the Istituto Superiore di Sanità, Roma, Italia https://arch.iss.it/, accessed on 21 July 2022).

**Figure 2 biomedicines-10-02050-f002:**
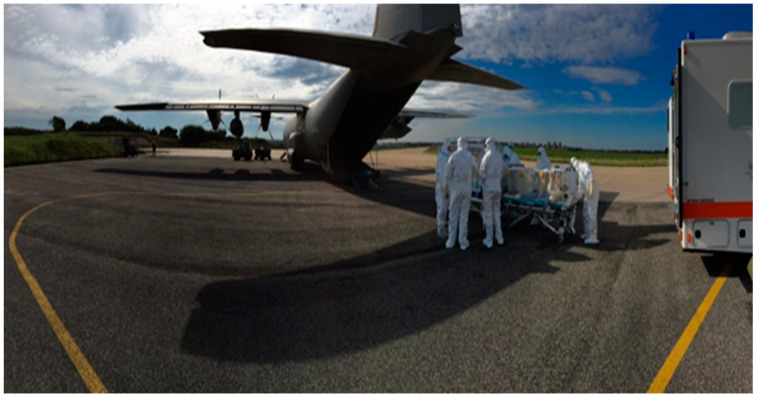
A high biocontainment aeromedical evacuation team engaged in the transfer of the Aircraft Transit Isolator (ATI) stretcher from the aircraft to the ambulance for subsequent transport of the patient to the hospital.

**Table 1 biomedicines-10-02050-t001:** Current estimated global infections and deaths of some infectious diseases according to transmission type.

Diseases According to Transmission Type	Estimated Global Infections	Estimated Global Deaths	Year Reference
**Air-borne transmitted **			
Tuberculosis	8,700,000	1,400,000	2011 [8]
COVID-19	195,044,798	650,702	2021–2022
Influenza	1,000,000,000	300,000–500,000	Typical epidemic year [9]
Meningococcal Meningitis	1,200,000	135,000	[10]
Measles	9,700,000	134,200	2015 [11]
**Blood-borne/sexually transmitted **			
Hepatitis B	1,500,000	820,000	2019 [12]
HIV infection	1,500,000	680,000	2020 [13]
Hepatitis C	1,500,000	290,000	2019 [14]
**Vector-borne transmitted **			
Malaria	241,000,000	627,000	2020 [15]
Yellow fever	84,000–170,000	29,000–60,000	2013 [16]
Japanese encephalitis	67,900	13,600–20,400	[17]
Dengue	390,000,000	12,000	2010 [18] 2002 [19]
**Fecally transmitted **			
Typhoid	11,000,000–20,000,000	128,000–161,000	2018 [20]
Cholera	1,300,000–4,000,000	21,000–143,000	2015 [21]
Amoebiasis	500,000,000	40,000–100,000	2000 [22]
Hepatitis E	20,000,000	44,000	2017 [23]
Hepatitis A	158,944,000	39,280	2019 [24]
**Water-related **			
Leptospirosis	1,030,000	58,900	2015 [25]

**Table 2 biomedicines-10-02050-t002:** Infectious diseases and available vaccines and antibodies for prophylaxis and therapy.

Vaccine-Preventable Infectious Diseases	Type of Vaccine	Type of Antibody
Smallpox	Live/recombinant	Specific human
Typhoid fever	Live/Polysaccharide Subunit/Conjugate	
Tetanus	Subunit	Specific human
Diphtheria	Subunit	Specific equine
Pertussis	Inactivated whole cell/recombinant	
Tuberculosis	Live	
Meningococcal meningitis	Polysaccharide Subunits/Conjugate	
Hepatitis A	Inactivated	Standard human
Hepatitis B	Subunit	Specific human
Poliomyelitis	Live/Inactivated	
Measles	Live	Standard human
Mumps	Live	
Rubella	Live	Standard human
Varicella	Live	Specific human
Influenza	Subunits/Live	
Adenovirus	Live	
COVID-19	RNA	Monoclonals
Pneumococcus	Polysaccharide Subunits/Conjugate	
Rabies	Inactivated	Specific human/equine
Yellow fever	Live	
Japanese encephalitis	Inactivated	
Tick-borne encephalitis	Inactivated	
Human papillomavirus	Recombinant	
Cholera	Inactivated whole cell/Recombinant/Live oral	
Leptospirosis	Inactivated whole-cell	
Dengue	Recombinant live	
** Non-Vaccine-Preventable** **Infectious Diseases**		
Epidemic typhus	The inactivated vaccine in World War II	
Scrub typhus		
Trench fever		
Leishmaniasis	Vaccine Brazil immunotherapy/Uzbekistan live	
Malaria	Recombinant, licensed for pediatric use	
Lymphatic filariasis		
Schistosomiasis		
Trypanosomiasis		
Other parasitic diseases		
Human Immunodeficiency Virus		
Hepatitis C		
Hepatitis E	Recombinant vaccine licensed in China	
Chikungunya virus	Live attenuated vaccine (IND°)	
Zika virus		
Crimean-Congo hemorrhagic fever	Inactivated vaccine licensed in Bulgaria	
Hantaviruses	Inactivated vaccine licensed in Korea	
West Nile and Rift Valley viruses		
Acute respiratory syndrome		
Acute diarrheal syndrome		
** Biological Agents for** **Bio-Warfare/Bioterrorism Category A–B**		
Anthrax	Inactivated	Polyclonal/Monoclonal
Botulism	Subunit (IND°)	Equine/human
Plague	Subunit (IND°)	
Tularemia	Live (IND°)	
Viral hemorrhagic fevers (filovirus/arenavirus)	Viral vectored (Ebola)	Monoclonal (Ebola)
Brucellosis		
Q fever	Inactivated vaccine licensed in Australia	
New World Viral Encephalitis	Live/Inactivated (IND°)	

°IND, investigational new drug.

**Table 3 biomedicines-10-02050-t003:** Effectiveness of typhoid vaccine in the British Army (Anglo-Boer War, 1899 and World War I, 1915).

Anglo-Boer War	Immunized	Unimmunized	*p*
British Army	14,626 (4.46%)	313,618 (95.54%)	
Disease	1417 (9.7%)	48,754 (15.5%)	<0.0000001
Case-fatality rate	163 (11.5%)	6991 (14.34%)	0.002965
** World War I **			
British Army	604,420 (94%)	38,580 (6%)	
Disease	570 (0.094%)	295 (0.764%)	<0.0000001
Case-fatality rate	34 (5.96%)	89 (30.2%)	<0.0000001

From references [43,44] slightly modified.

**Table 4 biomedicines-10-02050-t004:** Effectiveness of prophylactic hyper-immune anti-tetanus serum in the British Army in World War I (WWI) and anti-tetanus vaccine in the US Army in World War II (WWII).

British Army		*p*	US Army		*p*
Tetanus incidence September 1914	9/1000	0.04018	Tetanus incidence WWI	13.4/100,000	0.001305
Tetanus incidence December 1914	1.4/1000		Tetanus incidence WWII	0.44/100,000	
Pre-serum average case-fatality rate	85%	<0.0000001			
Post-serum average case-fatality rate	47%				

From references [63,66] modified.

**Table 5 biomedicines-10-02050-t005:** Mean annual incidence of some vaccine-preventable infectious diseases in the Italian military in two different periods as well as relative reduction.

Disease	Mean Annual Incidence 1986–1997	Mean Annual Incidence 2008–2018	Reduction
Pulmonary TB	10.4/100,000	0.675/100,000	15.4-fold
Hepatitis A	17.5/100,000	0.5/100,000	35-fold
Hepatitis B	19/100,000	0.44/100,000	43-fold
Measles	671/100,000	1.31/100,000	512-fold
Mumps	45.5/100,000	0.32/100,000	142-fold
Rubella	936/100,000	1.825/100,000	512-fold
Varicella	1300/100,000	7.29/100,000	178-fold

**Table 6 biomedicines-10-02050-t006:** COVID-19 infections in the aircraft carrier Theodore Roosevelt and in the cruise ship Diamond Princess.

Ships	Theodore Roosevelt	Diamond Princess	*p*
Crew/passengers	4779	3700	
Infected	1331 (27.85%)	712 (19.24%)	<0.0000001
Hospitalized	23 (1.73%)	36 (5%)	0.00003448
Deaths	1 (0.075%)	13 (1.83%)	0.00001793

From [229,230] modified.

**Table 7 biomedicines-10-02050-t007:** Relevance for the military of vaccine-preventable infectious diseases and military contribution to their control.

Disease	Military Relevance	Military Contribution
Smallpox	It may heavily influence the outcome of a battle/war—biological weapon category A	First variolization of an army—early vaccine uses in the military worldwide may have contributed to disease eradication
Typhoid fever	Outbreaks in deployed troops to endemic areas and wartime—biological agent category B	Vaccine development and use—dramatic typhoid reduction, particularly in WWI
Tetanus	Frequent contaminated wounds in the military	Passive immunization—collaboration in vaccine development
Diphtheria	Recently observed in adults	Vaccination as a public health measure—military and civilian surveillance systems should be interconnected
Pertussis	Recently observed in adults	Vaccination as a public health measure
Tuberculosis	Higher prevalence in the military than in the general population up to WWI	Discovery of infectious nature. Vaccine development. Epidemiology in wartime
Meningococcal meningitis	High morbidity and mortality in the military	Identification of immune protection—polysaccharide vaccine development
Hepatitis A	Widespread in the military— “camp jaundice”	Demonstration of protection by human Immunoglobulin—vaccine development
Hepatitis B	The military are exposed to sexually transmitted diseases—soldiers as a “walking blood bank”	Demonstration of protection by antibodies
Poliomyelitis	During WWII, polio was highly incapacitating	Vaccination as a public health measure
Measles	Highly contagious, severe disease	Vaccination as a public health measure
Mumps	Highly contagious, incapacitating disease	Vaccination as a public health measure
Rubella	Incapacitating disease—congenital rubella syndrome as a dramatic problem	First isolation of the virus—vaccine development
Varicella	Highly contagious, incapacitating	Vaccine use is quite limited
Influenza	Frequent cause of acute respiratory disease in the military	Support to first vaccine development—first isolation of “Asian” virus—identification of drifts and shifts—organization of surveillance system
Adenovirus	Frequent cause of acute respiratory disease in the military	First isolation of the virus—vaccine development
Coronavirus disease-2019	The military are exposed because they are engaged in pandemic containment	The military have been crucial for organizing diagnostic and vaccination campaigns
Pneumococcus	Responsible for severe acute respiratory disease	Discovery of microorganism—first hexavalent polysaccharide vaccine
Rabies	Severe threat to deployed service members	Preventive vaccination
Yellow fever	Endemic in Cuba—threat to the US military deployed there—biological agent category C	Demonstration of mosquito-transmission Disease control through vector eradication
Japanese encephalitis	Possible threat for the military deployed to Asia	Vaccination WWII—epidemiology—field trial inactivated vaccine in Thailand
Tick-borne encephalitis	Possible threat for the military deployed to endemic countries—biological agent category C	Vaccine has demonstrated to be safe and immunogenic
Human papillomavirus infection	The military are exposed to sexually transmitted diseases	HPV vaccine inclusion in the military vaccination schedule may be a relevant measure of public health
Cholera	Severe disease frequently present in wars—biological agent category B	Rehydration therapy—vaccine development
Leptospirosis	The military may be infected in field exercise training and wartime	Chemoprophylaxis by doxycycline
Dengue	Incapacitating threat for the military deployed to endemic areas	Vaccine development

WWI, First World War; WWII, Second World War; HPV, human papillomavirus.

**Table 8 biomedicines-10-02050-t008:** Relevance for the military of non-vaccine-preventable infectious diseases and military contribution to their control.

Disease	Military Relevance	Military Contribution
Epidemic typhus	Present in many wars—biological agent category B	USA troops received Cox’s vaccine in WWII
Scrub typhus	The military deployed to endemic areas are at risk	Patented recombinant rickettsia protein
Trench fever	The name itself witnesses military relevance	First description—etiology
Leishmaniasis	The military deployed to endemic areas are at risk	First description—personal protection—vector control
Malaria	The military deployed to endemic areas are at risk	Etiology—drugs, monoclonal antibody, and vaccine development
Lymphatic filariasis	The military deployed to endemic areas are at risk	Demonstration of eradicating treatment
Schistosomiasis	The military deployed to endemic areas are at risk	Diagnosis—treatment—environ. prevention
Trypanosomiasis	The military deployed to endemic areas are at risk	Etiology—treatment, mobile teams
Other parasitic diseases	The military deployed to endemic areas are at risk	Treatment
HIV infection	The military is at risk of sexually transmitted diseases—soldiers as “walking blood bank”	Epidemiology—disease biology—vaccine development
Hepatitis C	Soldiers as “walking blood bank”	Screening—monitoring pre/post-risk mission
Hepatitis E	It is a risk for the military deployed to endemic areas	Vaccine development
Chikungunia	The military deployed to endemic areas are at risk	Vaccine development
Zika	The military deployed to endemic areas are at risk	Vaccine development
Crimean–Congo	Biological agent category C	Passive immunotherapy
Hantaviruses	The military deployed to endemic areas are at risk—biological agent category C	Vaccine development
Acute respiratory syndrome (influenza, rhinoviruses, para-influenza viruses, respiratory syncytial virus, adenoviruses, coronaviruses, human metapneumovirus *Streptococcus pyogenes*, *Streptococcus pneumoniae*, *Bordetella pertussis*, *Mycoplasma pneumoniae*, *C. pneumoniae*)	It is one type of pathology of great interest for the military, especially recruited trainees, probably for environmental live conditions. It may be due to a series of etiologic agents, for a minority of which preventive vaccination is available. However, even in these cases, the vaccine-induced protection is not absolute, such as for *S. pneumoniae*, influenza and SARS-CoV-2, in the last two cases because of the high variability of these RNA viral agents. Finally, for adenovirus, the vaccine is only administered to the US military, although the epidemiological problem is present in the military of other countries	Support to first flu vaccine development—first isolation of “Asian” virus—identification of drifts and shifts—organization flu surveillance systems—first adenovirus identification and vaccine development—co-discovery of *Streptococcus pneumoniae*—testing the first hexavalent polysaccharide vaccine—COVID-19 vaccine development—US military have organized a network of worldwide laboratories for providing advanced diagnostic capabilities, as proven with MERS-CoV in Jordan in 2012
Acute diarrheal syndrome (cholera, *Salmonella*, *Shigella*, enterotoxigenic *E. coli*, *C. jejuni*, Norwalk virus)	This is a condition of great concern for the military. Cholera and typhoid fever are not a problem anymore. Some of these agents are considered biological threats category B	Vaccine development—WRAIR is working to develop effective vaccines for *Shigella*, *Campylobacter*, and enterotoxigenic *E. coli*; however, no vaccines are available yet

**Table 9 biomedicines-10-02050-t009:** Biological agents category A, B, and C, CDC classification (https://emergency.cdc.gov/agent/agentlist-category.asp; (accessed 27 July 2022) and [511], slightly modified).

**Biological Agents, Category A**
variola major (smallpox); *Bacillus anthracis* (anthrax); *Clostridium botulinum* toxin (botulism); *Yersinia pestis* (plague); *Francisella tularensis* (tularemia); Viral hemorrhagic fevers; ◦ filoviruses; ▪ Ebola hemorrhagic fever; ▪ Marburg hemorrhagic fever; ◦ arenaviruses; ▪ Lassa (Lassa fever); ▪ Junin (Argentine hemorrhagic fever) and related viruses.
**Biological Agents, Category B**
*Brucella species* (brucellosis); epsilon toxin of *Clostridium perfringens*; Food safety threats (*Salmonella* species, *Escherichia coli* O157:H7, *Shigella*); *Burkholderia mallei* (glanders); *Burkholderia pseudomallei* (melioidosis); *Chlamydia psittaci* (psittacosis); *Coxiella burnetii* (Q fever); ricin toxin from *Ricinus communis* (*castor beans*); *Staphylococcus* enterotoxin B; *Rickettsia prowazekii* (typhus fever); Alphaviruses, such as eastern equine encephalitis, Venezuelan equine encephalitis, and western equine encephalitis (viral encephalitis); Water safety threats (*Vibrio cholerae*, *Cryptosporidium parvum*).
**Biological Agents, Category C**
Nipah virus; hantaviruses; tickborne hemorrhagic fever viruses; tickborne encephalitis viruses; yellow fever; multidrug-resistant tuberculosis.

**Table 10 biomedicines-10-02050-t010:** Relevance for the military of possible biological weapons and military contribution to their control.

Category A	Military Interest	Military Contribution
Smallpox	Possible biological weapon	Large vaccine use
Anthrax	Possible biological weapon	Vaccine development—epidemiology—genotyping
Botulism	Possible biological weapon	Vaccine development—epidemiology—genotyping
Plague	Possible biological weapon	Vaccine development—epidemiology—genotyping
Tularemia	Possible biological weapon	Vaccine development
Filovirus	Possible biological weapon	Vaccine development—polyclonal human Immunoglobulin
Arenavirus	Possible biological weapon	Pathogenesis
**Category B**		
Brucellosis	Possible biological weapon	Etiology—vaccine development
Q fever	Possible biological weapon	Vaccine development
Viral Encephalitis	Possible biological weapons	Vaccines and mAbs development—fieldable diagnosis

mAbs, monoclonal antibodies.

## Data Availability

Not applicable.

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
