# Peer review of "A Historical Review of Military Medical Strategies for Fighting Infectious Diseases: From Battlefields to Global Health"

_biomedicines, 2022, doi:10.3390/biomedicines10082050_

Round 1

Reviewer 1 Report

I congratulate the authors for this extraordinary work. It is a work of great value with high quality and will be very well received in the scientific community. I have few suggestions for the authors. Only that they improve the quality of the figures and their description.

In my opinion, this review is very interesting. Currently, there is no revision of this magnitude in Europe. This review is of great importance, and will be cited both in Europe and in other countries. The authors have done a good job and after minor changes the manuscript can be published.

Another important point is the use of English grammar. Please, the authors should improve the use of English grammar.

I congratulate the authors again.

Author Response

We would like to thank the reviewer for his/her very kind evaluation of our review. The description of the figures has been modified in order to make them more understandable and the use of English grammar has been improved.

Reviewer 2 Report

In this manuscript titled “A historical review of military medical strategies for fighting infectious diseases: from battlefields to global health”, the authors did a thorough review on infectious diseases and its relationship with military mission. Overall, this is an interesting and comprehensive review. However, I do have several minor questions.

Minor issues:

1. Line 268-270, “At half of seventies of the last century a new living oral typhoid vaccine was developed [36] and in the second half of eighties the Vi polysaccharide-protein conjugate was also developed [37]”. It is a little vague here, more details are probably needed here.

2. Line 290, grammar issue. Chang “elaborates a neurotoxin toxin” to “produces a neurotoxin”?

3. Line 335, “by exposing the tetanus and diphtheria toxins to little amount of formaldehyde”, it is probably better to specify the amount?

4. In the section “2.10. Poliomyelitis”, “In some subjects, the virus, which has a marked neurotropism, localizes at spinal level, most frequently in the anterior horn cells of the cord”, is there any explanation of why it targets the anterior horn cells of the cord?

5. In the section “2.14. Varicella”, “In conclusion, despite the availability of a safe and effective tool for varicella prevention, it appears that the vaccine is not so largely used in the military and, even when it is used, the policy to limit vaccine administration to those lacking documentation of infant vaccination or disease”, is there any specific reason why its usage is limited even though this vaccine is safe and effective?

6. Line 1327-1328, “all the 25 NATO countries reporting the military vaccinating program include rabies vaccine for selected military categories”, any data on the efficacy and safety?

7. Line 2069-2071, please check the grammar.

8. Line 33550-3356, “such effort may only be faced through an international collaboration among the militaries of allied countries, such as NATO”, I would suggest to delete “such as NATO”. This review has emphasized data from NATO for numerous times, I don’t think it is necessary to show this review as a NATO-specific review at the discussion and conclusion part.

9. In the “conclusions” section, “In conclusion, war and infectious diseases have a reciprocal influence”. It seems that the authors mainly talked about the impact of eliminating infectious diseases on military missions in this review. The authors didn’t mention much about how limiting military conflicts will affect infectious diseases, so I don’t really see a “a reciprocal influence” here. In the conclusion part, it may be nice to discuss a little about how limiting military conflict or escalated military actions will affect infectious diseases. Resources from United Nation about military and disease may help to draft this part.

Author Response

We thank the reviewer for his/her kind evaluation of our review and for the suggestions which allowed the manuscript to be improved.
